# Optimal Domain-Aware Privacy Mechanisms for Synthetic Data Generation

**Sajani Vithana**[1] **Sangwon Jung**[2] **Haoyang Hu**[3] **Viveck R. Cadambe**[* 3] **Flavio P. Calmon**[* 1] **Haewon Jeong**[* 4 5]

## Abstract

Differential privacy (DP) imposes fundamental trade-offs between privacy and statistical fidelity in synthetic data generation. While access to public data has been shown to improve these trade-offs empirically, existing approaches use public data only indirectly, through pre-processing (e.g., using pre-trained generative models) or post-processing steps (e.g., matching target statistics estimated from public datasets), while relying on domain-agnostic DP mechanisms. In this work, we lay the theoretical framework to study the principled incorporation of public data into DP mechanisms themselves. We consider normalized histograms as distribution estimators and characterize the asymptotically optimal domain-aware privacy mechanism within a specific class of DP mechanisms. We introduce PUBMIX, a public-data-aware DP mechanism that can be used in histogram-based data synthesis pipelines. Our experiments demonstrate that PUBMIX significantly improves synthetic data generation quality compared to domain-agnostic privacy mechanisms.

## 1. Introduction

Modern data-driven systems require access to large, diverse datasets in order to train, evaluate, and validate models. In many of these settings, the underlying data consists of sensitive records such as patient histories or financial transactions that cannot be directly shared or reused due to privacy regulations. Differentially private (DP) synthetic data generation offers a solution to this challenge by constructing a synthetic

dataset that preserves the statistical properties of the original private data with formal DP guarantees (Ponomareva et al., 2025). The resulting synthetic dataset can be freely shared and reused in downstream tasks with no privacy leakage.

To improve utility of DP synthetic data generation, several works have explored leveraging auxiliary public data from the same domain, typically incorporating it through pre-processing or post-processing steps. Examples include pre-training generative models on public data (Ghalebikesabi et al., 2023), using public data to inform which statistics to measure (Fuentes et al., 2024; Liu et al., 2021a), and post-processing synthetic data to align with public data constraints (Slavkovic & Reimherr, 2020; Wang et al., 2023). In all these approaches, however, the underlying DP mechanism (how randomness is injected to ensure privacy) remains domain-agnostic, typically based on Gaussian or Laplace noise (Dwork et al., 2006).

We propose a fundamentally different paradigm for incorporating public data: using it directly to design how randomness is injected to achieve privacy in data synthesis. By constructing domain-aware noise distributions informed by public data, we obtain DP mechanisms that achieve improved privacy–utility trade-offs compared to domain-agnostic privacy mechanisms.

DP synthetic data generation typically involves estimating the distribution of a private dataset, perturbing the estimate to ensure privacy, and sampling synthetic data from the perturbed distribution (McKenna et al., 2021; Frigerio et al., 2019; Pfitzner & Arnrich, 2022; McKenna et al., 2022; 2019; Hardt et al., 2012; Zhang et al., 2017). Standard domain-agnostic DP mechanisms, such as Laplace or Gaussian noise, perturb the estimated private distribution in arbitrary directions on the probability simplex. In contrast, we show that public data can be leveraged to identify optimal directions along which the private distribution should be perturbed.

In this paper, we develop a theoretical framework for analyzing a linear mixing-based class of domain-aware DP mechanisms for private data synthesis, considering normalized histograms as private distribution estimators. In this canonical setting, we characterize the asymptotically optimal noise distribution, demonstrating that it takes the form of a *floor-raised* (Gallager, 1968) version of the public distribution. From a theoretical stand point, these insights serve

---

*Senior authors in alphabetical order. [1]School of Engineering and Applied Sciences, Harvard University, Allston, MA, USA. [2]Trillion Labs, Seoul, South Korea. [3]School of Electrical and Computer Engineering, Georgia Institute of Technology, Atlanta, GA, USA. [4]Department of Electrical and Computer Engineering, University of California, Santa Barbara, USA. [5]Flatiron Institute, New York, USA. Correspondence to: Sajani Vithana <sajani@seas.harvard.edu>.

*Proceedings of the 43rd International Conference on Machine Learning*, Seoul, South Korea. PMLR 306, 2026. Copyright 2026 by the author(s).

as a foundational step toward developing domain-aware DP mechanisms for complex distribution estimators used in synthetic data generation (see Sec. 3.2). From a practical perspective, our analysis directly informs the design of PUBMIX, a domain-aware DP mechanism for histogram sampling that can replace standard Gaussian or Laplace mechanisms in existing histogram-based synthesis pipelines (Lin et al., 2024; Xie et al., 2024; Abacha et al., 2025; Zhang et al., 2025) for improved utility (see Sec. 5.2). Our key contributions are:

- We introduce a principled framework for leveraging public data to construct domain-aware noise distributions that improve the privacy-utility trade-off in DP data synthesis compared to domain-agnostic privacy mechanisms.

- We provide an information-theoretic analysis of the privacy–utility trade-off in DP histogram sampling within a class of mechanisms and characterize the asymptotically optimal domain-aware privacy mechanism.

- We propose PUBMIX, a DP histogram sampling mechanism that incorporates domain-aware noise and serves as a drop-in privacy module for histogram-sampling-based synthetic data generation.

### 1.1. Related Work

**DP synthetic data generation** has been studied across modalities including tabular data, text, and images. Existing approaches include (i) statistical methods that privately release noisy summary statistics to fit distributional models (McKenna et al., 2022; Hardt et al., 2012; Cai et al., 2021; Zhang et al., 2017; McKenna et al., 2021), and (ii) training deep generative models with DP-SGD (Frigerio et al., 2019; Dockhorn et al., 2023; Yue et al., 2023). Complementary paradigms include PATE (Papernot et al., 2018; Long et al., 2021), which trains teachers on disjoint private partitions and aggregates their noisy predictions to label public data for synthesis; Private Evolution (Lin et al., 2024; Xie et al., 2024; Tran et al., 2026), which iteratively samples from (hierarchical) histograms to produce synthetic datasets; and variants of private prediction (Amin et al., 2024; Tang et al., 2024), which synthesize text by privately aggregating next-token predictions from LLMs prompted with sensitive examples. See (Ponomareva et al., 2025) for a comprehensive survey. While some methods leverage public data to reduce privacy cost (Fuentes et al., 2024; Liu et al., 2021a), the underlying DP mechanisms remain largely domain-agnostic (Gaussian, Laplace, or exponential mechanisms). In contrast, we introduce the concept of domain-aware privacy mechanisms.

**DP multi-sampling** provides sample-complexity bounds for drawing independent samples from a given private distribution under DP, covering broader distribution families

(Cheu & Nayak, 2025). We extend the DP multi-sampling setting by incorporating public data into the problem formulation, while restricting to a more structured setting (linear mixing-based DP mechanisms) to make public-data integration tractable.

**Existing linear mixing methods** include variants of private prediction (Flemings et al., 2024; Ginart et al., 2022) that serve as alternatives to existing DP training methods. Instead of privatizing the model itself, these methods enforce privacy at inference time by perturbing the model's output. In LLMs, this typically involves mixing or projecting the private model's next-token distribution toward a public distribution to limit leakage from memorized private data. While both PUBMIX and private prediction involve linear mixing and public distributions, they differ fundamentally in goal and technical design (More details in App. A).

**DP histograms** are a core primitive underlying many synthesis pipelines, with work spanning basic privatization (Dwork & Roth, 2014; Dwork et al., 2006), hierarchical constructions (Qardaji et al., 2013), and adaptive/improved binning (Hay et al., 2010). Recent private evolution methods and their federated learning extensions (Abacha et al., 2025; Hou et al., 2025) further highlight the practical role of histogram sampling, demonstrating strong empirical performance across modalities when combined with foundation models. PUBMIX can be incorporated as a drop-in privacy mechanism within the histogram-based pipelines.

## 2. Problem Formulation

Let $D_{prv} = \{x_1, \ldots, x_n\} \subseteq \mathcal{X}^n$ denote a private dataset where $\mathcal{X}$ is a discrete sample space with $|\mathcal{X}| = d$. Each $x_i \in D_{prv}$ is sampled i.i.d. from an underlying distribution $P_{prv} \in \Delta_d$, where $\Delta_d$ is the probability simplex over $\mathcal{X}$. Let $D_{pub} = \{v_i\}_{i=1}^{\ell} \subseteq \mathcal{X}^{\ell}$ be a public dataset from the same domain as $D_{prv}$, sampled i.i.d from an underlying distribution $P_{pub} \in \Delta_d$. We estimate $P_{prv}$ and $P_{pub}$ using the normalized histograms of $D_{prv}$ and $D_{pub}$ over $\mathcal{X}$, denoted by $\bar{P}_{prv}$ and $\bar{P}_{pub}$, respectively.

Our goal is to generate a synthetic dataset $D_{syn} = \{y_i\}_{i=1}^{m}$ consisting of $m$ samples that: (i) satisfies DP with respect to $D_{prv}$, (ii) is statistically *close* to $D_{prv}$. For this, we design a synthetic data distribution $P_{syn} \in \Delta_d$, from which we draw $m$ samples i.i.d. to obtain the synthetic dataset $D_{syn}$. Specifically, we consider synthetic distributions of the form:

$$P_{syn} = (1 - \beta)\bar{P}_{prv} + \beta P_{nse} \tag{1}$$

where $\beta \in [0, 1]$ is a mixing coefficient and $P_{nse} \in \Delta_d$ is a noise distribution. $\beta$ and $P_{nse}$ are globally known fixed parameters that are independent of the private data $D_{prv}$.

Our goal is to find the optimum parameters $\beta$ and $P_{nse}$ in (1) that achieves the maximum *utility* while satisfying DP.

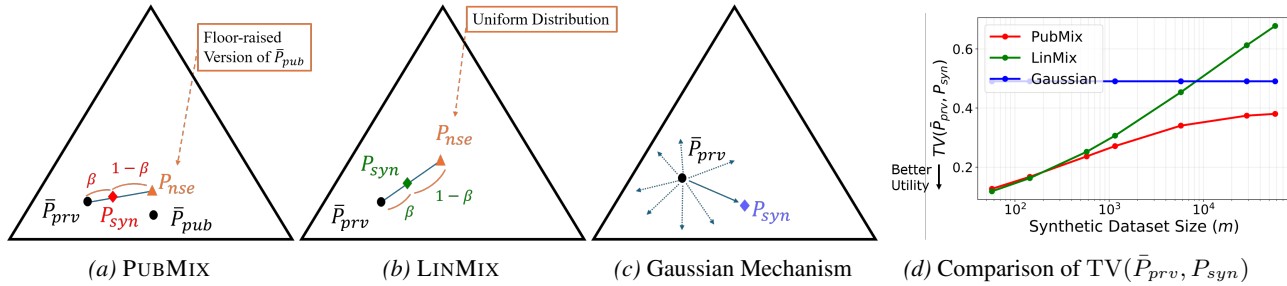

*Figure 1.* Comparison of PUBMIX, LINMIX, and the Gaussian Mechanism for differentially private histograms. (a)–(c) provide a geometric illustration of the perturbations in the probability simplex for (a) PUBMIX, (b) LINMIX, and (c) the Gaussian mechanism. (d) shows the total variation distance between $(\bar{P}_{prv}, P_{syn})$ as a function of the number of synthetic samples $m$, for PUBMIX, LINMIX, and the Gaussian mechanism for $n = 57717, \varepsilon = 5, \delta = 1/n$ based on the 2023 ACS PUMS dataset.

When the mixing parameter $\beta = 0$, we have $P_{syn} = \bar{P}_{prv}$, which is ideal for utility but provides no privacy guarantee. The other extreme, $\beta = 1$, yields $P_{syn} = P_{nse}$, and the generated synthetic data are independent of the private dataset, resulting in a substantial loss of utility while ensuring perfect privacy. Accordingly, for a fixed noise distribution $P_{nse}$ we first identify the smallest $\beta$ that satisfies DP, and then optimize $P_{nse}$ to obtain the best achievable privacy–utility trade-off. The resulting $P_{nse}$ determines the optimal direction along which the private distribution $\bar{P}_{prv}$ should be perturbed. Here, $P_{nse}$ will depend on the public data, as we will see shortly. First, we formally define the privacy constraint and the utility metric.

**Privacy constraint:** We consider $(\varepsilon, \delta)$-DP of the $m$ generated synthetic samples $D_{syn}$. Let $D_{prv}$ and $D'_{prv}$ be any two neighboring datasets that differ in only one data entry. Let $P_{syn}$ and $P'_{syn}$ be the corresponding synthetic distributions from (1) for any fixed $\beta \in [0,1]$ and $P_{nse} \in \Delta_{\mathcal{X}}$. Then, the $(\varepsilon, \delta)$-DP constraint is given by,

$$\mathbb{P}(D_{syn} \in S) \le e^{\varepsilon} \mathbb{P}(D'_{syn} \in S) + \delta,$$
$$\forall S \subseteq \mathcal{X}^m, \ \forall D_{prv}, D'_{prv} \quad (2)$$

where $D_{syn}$ and $D'_{syn}$ correspond to the $m$ i.i.d. samples drawn from $P_{syn}$ and $P'_{syn}$.

**Utility:** The utility of $D_{syn}$ is defined as the total variation (TV) distance between $\bar{P}_{prv}$ and $P_{syn}$, i.e., $\mathrm{TV}(\bar{P}_{prv}, P_{syn})$.

**The admissible set:** To select a noise distribution $P_{nse}$ in (1), we define an *admissible set* of private distributions *close* to the public distribution $\bar{P}_{pub}$. This set is fixed and independent of the given private data $D_{prv}$. We define the admissible set as all distributions within a $\gamma$-TV ball around $\bar{P}_{pub}$:

$$\mathcal{B}_{\gamma, \bar{P}_{pub}} = \{P \in \Delta_d : \mathrm{TV}(P, \bar{P}_{pub}) \le \gamma\}$$

for some proximity parameter $\gamma \in (0, 1]$, independent of

$D_{prv}$. The parameter $\gamma$ is chosen based on public knowledge (see Sec. 5). The key motivation for this definition is that datasets drawn from the same domain induce distributions that are close to each other on the probability simplex with high probability. We therefore seek a $P_{nse}$ that are optimized for distributions in $\mathcal{B}_{\gamma, \bar{P}_{pub}}$. Concretely, as we describe next, we optimize $P_{nse}$ considering the worst-case private distribution within $\mathcal{B}_{\gamma, \bar{P}_{pub}}$.[1]

**An optimization formulation for mechanism design:** Given a target $(\varepsilon, \delta)$, a public distribution $\bar{P}_{pub}$, and a proximity parameter $\gamma$, our goal is to find the optimal noise distribution $P_{nse}$ and mixing coefficient $\beta$ in (1) that satisfies the privacy constraint in (2) while maximizing utility. The optimization problem we solve to obtain these parameters is given by:

$$\min_{P_{nse} \in \Delta_d} \quad \min_{\beta \in [0,1]} \quad \max_{\bar{P}_{prv} \in \mathcal{B}_{\gamma, \bar{P}_{pub}}} \mathrm{TV}\Big(\bar{P}_{prv}, P_{syn}\Big)$$
$$\text{s.t.} \quad \mathbb{P}(D_{syn} \in S) \le e^{\varepsilon}\mathbb{P}(D'_{syn} \in S) + \delta,$$
$$\forall S \subseteq \mathcal{X}^m, \forall D_{prv}, D'_{prv} \quad (3)$$

We denote the parameters that solve (3) as $(\beta^*, P_{nse}^*)$.[2]

**PUBMIX:** Once the parameters $\beta^*$ and $P_{nse}^*$ are obtained, for any given $D_{prv}$, we compute $P_{syn} = (1 - \beta^*)\bar{P}_{prv} + \beta^* P_{nse}^*$ using the corresponding $\bar{P}_{prv}$, and draw $m$ samples from $P_{syn}$ as private synthetic data $D_{syn}$. We call this mechanism PUBMIX, and it is outlined in Alg. 1.

**Why use PUBMIX instead of just adding noise to histograms?** Standard DP histogram mechanisms (Dwork & Roth, 2014) add i.i.d. Gaussian or Laplace noise to

---

[1]If the private distribution of $D_{prv}$ lies outside $\mathcal{B}_{\gamma, \bar{P}_{pub}}$ (which cannot be verified a priori), PUBMIX still satisfies DP. However, a weaker utility guarantee will be achieved compared to (3).

[2]The parameters $\beta^*$ and $P_{nse}^*$ in PUBMIX are not derived from the given private dataset. They are pre-computed parameters based on the worst case private distribution within a $\gamma$-TV ball around $\bar{P}_{pub}$ (see (3)), and are independent of the given private dataset.

each histogram bin to achieve DP. Since the noise is zero-mean and independent across bins, each bin can increase or decrease arbitrarily. When normalized to obtain a probability distribution, the perturbed histogram can move in arbitrary directions on the probability simplex relative to the non-private distribution. This domain-agnostic approach does not consider which perturbation directions optimize the privacy-utility trade-off for the given data domain (Fig.1c). In contrast, PUBMIX leverages public data from the same domain to identify a principled direction in which to perturb $\bar{P}_{prv}$, yielding a synthetic distribution $P_{syn}$ that better preserves utility while still satisfying DP (Fig.1a).

A key feature of PUBMIX is *sample dependency*. Most existing DP synthesis methods incur a high privacy cost by releasing the entire privacy-protected distribution $P_{syn}$ (Xie et al., 2024; Lin et al., 2024). For instance, in existing DP histogram-based synthesis methods that add Gaussian or Laplace noise to each of the $d$ bins independently, $\text{TV}(\bar{P}_{prv}, P_{syn})$ increases linearly with $d$ (Qardaji et al., 2013). However, many applications require only a finite number of synthetic samples, and not the full distribution.

We show that, when the goal is to release only $m$ synthetic samples from the histogram rather than the histogram itself, it is possible to construct a synthetic sampling distribution $P_{syn}$ that is significantly closer to $\bar{P}_{prv}$, resulting in more accurate synthetic data (Fig. 1-d). The key insight is that the normalized histogram of a private dataset $D_{prv}$ differs from that of any neighboring dataset $D'_{prv}$ in at most two coordinates. Specifically, as $n$ denotes the number of samples in $D_{prv}$, replacing a single sample causes one bin to increase by $1/n$ while another bin decreases by $1/n$. When releasing an entire DP histogram, this sparsity of the neighboring difference cannot be leveraged, and noise must be injected across all $d$ coordinates. However, when releasing only samples, and working under $(\varepsilon, \delta)$-DP, this sparse structure can be leveraged, yielding tighter privacy accounting.

## 3. Main Results: Overview and Intuition

Before presenting the full mathematical treatment of how to solve (3) in Section 4, we first provide a high-level intuition for our solution and how PUBMIX leverages public data.

### 3.1. Technical overview

Recall that PUBMIX constructs the synthetic dataset $D_{syn}$ by drawing $m$ samples from a carefully designed distribution $P_{syn} = (1 - \beta^*)\bar{P}_{prv} + \beta^* P^*_{nse}$. Here, $\beta^*$ and $P^*_{nse}$ are the optimal parameters. To solve the optimization in (3), we first fix a noise distribution $P_{nse}$ and find the minimum mixing weight $\beta_{min}$ (for this specific $P_{nse}$) for which the DP constraint in (2) is satisfied.

---

> **Minimum mixing weight (details in Theorem 4.1):**
> Given a noise distribution $P_{nse}$, privacy parameters $\varepsilon, \delta$, number of synthetic samples $m$, and private dataset size $n$, there exists a threshold $\beta_{min}$ such that any $\beta \geq \beta_{min}$ satisfies $(\varepsilon, \delta)$-DP. Moreover,
>
> 1) $\beta_{min}$ is a function of $\varepsilon, \delta, m, n$ and $P_{nse}$.
>
> 2) $\beta_{min}$ is increasing in $m$.
>
> 3) $\beta_{min}$ depends on $P_{nse}$ only through its two smallest probability masses.

**Intuition:** For given parameters $\varepsilon, \delta, m, n, P_{nse}$, the privacy constraint in (2) can be equivalently written as:

$$E_{e^\varepsilon}(P_{syn}^{\otimes m} \| P'^{\otimes m}_{syn}) \leq \delta, \quad \forall D_{prv}, D'_{prv} \qquad (4)$$

where $E_{e^\varepsilon}(P \| Q)$ denotes the hockey-stick divergence between distributions $P$ and $Q$ (or the $E_\gamma$ divergence with $\gamma = e^\varepsilon$), and $P^{\otimes m}$ denotes the $m$-fold product distribution corresponding to $m$ i.i.d. draws of $P$. Solving (4) with equality yields the minimum mixing weight $\beta_{min}$ for the chosen $P_{nse}$. Importantly, since neighboring private datasets differ in exactly one sample, the ratio $\frac{P_{syn}}{P'_{syn}}$ is exactly 1 in all but (at most) two coordinates. Leveraging this 2-point sparse structure tightens the privacy accounting compared to existing DP histogram release methods.

For a given $P_{nse}$, the value of $\beta_{min}$ specifies the minimum separation that $P_{syn}$ must maintain from $\bar{P}_{prv}$ to satisfy DP. Since the information leakage increases with the number of synthetic samples released, $\beta_{min}$ increases with $m$ (see Fig. 2). The next property that $\beta_{min}$ depends on $P_{nse}$ only through its two smallest probability masses is also intuitive: Since the same mixing weight $\beta$ is applied to every coordinate in (1), the coordinates with the smallest noise masses in $P_{nse}$ receive the least added randomness. The worst case therefore occurs when the two coordinates that differ in neighboring private histograms coincide with the indices of the two smallest entries of $P_{nse}$. Consequently, letting $N_{min_1} = \min_i P_{nse}(i)$ and $N_{min_2} = \min_{i \neq i_1} P_{nse}(i)$ where $i_1 = \arg\min_i P_{nse}(i)$, the pair $(N_{min_1}, N_{min_2})$ determines the smallest $\beta$ for which DP is satisfied.

Since $\beta_{min}$ depends on $P_{nse}$ only through its two smallest probability masses $N_{min_1}$ and $N_{min_2}$, we write $\beta_{min}(N_{min_1}, N_{min_2})$ to make this dependence explicit in what follows.

Consider $N_{min_1} = N_{min_2} = N_{min}$. The behavior of $\beta_{min}(N_{min}, N_{min})$ for $0 \leq N_{min} \leq \frac{1}{d}$ is shown in Fig. 2, where $d = |\mathcal{X}|$. $\beta_{min}(N_{min}, N_{min})$ is decreasing in $N_{min}$ as a larger value of $N_{min}$ corresponds to a larger baseline noise being injected across all coordinates of $\bar{P}_{prv}$. Since more noise is already guaranteed at each coordinate, a smaller mixing weight $\beta$ suffices to satisfy DP.

Having characterized the minimum mixing weight for a fixed noise distribution $P_{nse}$, we next optimize over $P_{nse}$.

> **Asymptotically optimum noise distribution (details in Theorem 4.4):** For a given public distribution $\bar{P}_{pub}$ and $\gamma > 0$, the asymptotically optimum noise distribution $P_{nse}^*$ is a *floor-raised* version of $\bar{P}_{pub}$ with an optimal threshold $t$. Floor-raising a distribution $\bar{P}_{pub}$ with threshold $t$ raises all coordinates below $t$ up to $t$, and compensates by reducing the mass from coordinates above $t$ in a way that preserves normalization (see Fig. 3). Moreover, the optimum threshold $t$ is a function of $\gamma, \bar{P}_{pub}, \varepsilon, \delta, m$ and $n$.

**Intuition:** To determine the optimum noise distribution, we first observe that $\mathrm{TV}(\bar{P}_{prv}, P_{syn}) = \beta \mathrm{TV}(\bar{P}_{prv}, P_{nse})$. Then, the optimization in (3) can be upper bounded by:

$$\min_{P_{nse} \in \Delta_d} \beta_{min}(N_{min_1}, N_{min_1}) \left( \max_{\bar{P}_{prv} \in \mathcal{B}_{\gamma, \bar{P}_{pub}}} \mathrm{TV}(\bar{P}_{prv}, P_{nse}) \right)$$
(5)

since the mixing weight $\beta$ is chosen independently of $\bar{P}_{prv}$ and $\beta_{min}(N_{min_1}, N_{min_2})$ is decreasing in $N_{min_2}$. The optimal $P_{nse}^*$ in (5) is an asymptotic optimizer for (3) since (5) converges to (3) as $d \to \infty$. Details in Sec. 4.

Consider the two terms in (5). $\beta_{min}(N_{min_1}, N_{min_1})$ is minimized when $N_{min_1}$ is maximized (see Fig. 2). Thus, the $P_{nse}$ minimizing the first term is *the uniform distribution*, where $N_{min_1} = \frac{1}{d}$. Now consider the second term (inner max). For any given $P_{nse}$, the inner max is achieved by some $\bar{P}_{prv}$ on the boundary of $\mathcal{B}_{\gamma, \bar{P}_{pub}}$. Therefore, the minmax TV distance is achieved by the $P_{nse}$ at the $L_1$-*Chebyshev center*[3] of $\mathcal{B}_{\gamma, \bar{P}_{pub}}$ (close to $\bar{P}_{pub}$). The first term pushes $P_{nse}$ toward the center of the simplex $\Delta_d$ (the uniform distribution), while the second term pushes it toward the Chebyshev center of $\mathcal{B}_{\gamma, \bar{P}_{pub}}$, which lies close to $\bar{P}_{pub}$. The optimal $P_{nse}^*$ balances these two effects: it stays close to $\bar{P}_{pub}$ to minimize the worst-case TV term, while simultaneously increasing $N_{min_1}$ to reduce the $\beta_{min}(N_{min_1}, N_{min_1})$ term. We show that the resulting optimum $P_{nse}^*$ has a simple and interpretable form: $P_{nse}^*$ *is a floor-raised version of* $\bar{P}_{pub}$, *at an optimal threshold* $t$ (see Fig. 3). This floor-raising operation slightly moves $\bar{P}_{pub}$ towards the uniform distribution (details in App. B), thereby increasing $N_{min_1}$ and lowering $\beta_{min}(N_{min_1}, N_{min_1})$. At the same time, it preserves proximity to the minimizer of the inner maximization term.

**Significance of public data:** Consider the asymptotic solution to (3) when no public data is available, i.e., $\mathcal{B}_{\gamma, \bar{P}_{pub}} = \Delta_d$. For this, we again analyze the two terms in (5) sepa-

---

[3] $L_1$-Chebyshev center of a set $Q$: $\arg\min_c \max_{x \in Q} \|x - c\|_1$.

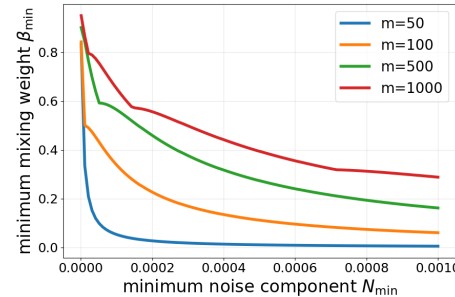

*Figure 2.* Comparison of minimum mixing weight $\beta_{min}$ for different numbers of synthetic samples $m$ and minimum noise components $N_{min}$ with $n = 1200, d = 1000, \varepsilon = 5$ and $\delta = 1/n$.

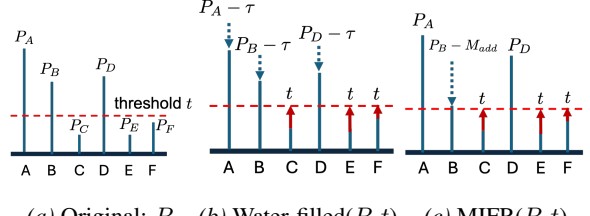

*(a) Original: $P$    (b) Water-filled$(P, t)$    (c) MIFR$(P, t)$*

*Figure 3.* Floor-raising: (a) original PMF, (b) Water-filling: raises smaller masses to $t$ and removes the added mass uniformly from larger masses (Def. 4.2), (c) Maximum Invariant Floor Raising (MIFR): raises smaller masses to $t$ and removes the added mass $M_{add}$ from larger ones to maximize the unchanged probability mass with respect to the original (Def. 4.3).

rately. The inner maximization in (5) is attained by a $\bar{P}_{prv}$ at an extreme point of the simplex (e.g., a corner of the triangle in Fig. 1). Thus, the noise distribution minimizing the inner maximum term in (5), i.e., the minmax optimal distribution, is the uniform distribution, which is at the center of the simplex. Now, consider the $\beta_{min}(N_{min_1}, N_{min_1})$ term in (5). Since $\beta_{min}(N_{min_1}, N_{min_1})$ is decreasing in $N_{min_1}$, it is minimized when $N_{min_1} = \frac{1}{d}$. Both of these facts imply that the objective in (5) is minimized by $P_{nse}^* = \frac{1}{d}\mathbf{1}_d$, i.e., the uniform distribution (see Corollary 4.5). We call this mechanism LINMIX.

However, this solution is driven by the worst-case private distribution, which is a point mass at one of the corners of the simplex. A point mass is a highly atypical and practically irrelevant distribution. Optimizing $P_{nse}^*$ based on a point mass forces $P_{nse}^*$ to be far from most realistic private distributions $\bar{P}_{prv}$, which results in a large TV distance between the private and synthetic distributions, as shown in Fig. 1. To mitigate this, we leverage public data from the same domain and solve (3) with $\mathcal{B}_{\gamma, \bar{P}_{pub}} \subset \Delta_d$. This results in PUBMIX, which achieves significantly improved performance (red curves).

## 3.2. Beyond Histograms

PUBMIX extends naturally to general discrete distribution estimators that satisfy a bounded sensitivity condition,

**Algorithm 1** PUBMIX

**Input:** : Private distribution $\bar{P}_{prv}$, public distribution $\bar{P}_{pub}$, private dataset size $n$, required sample size $m$, proximity parameter $\gamma$, privacy parameters $(\varepsilon, \delta)$

**Output:** : Synthetic dataset $D_{syn}$

1: Compute optimal noise distribution $P^*_{nse}$ and mixing weight $\beta^*$ using $\bar{P}_{pub}, \gamma, \varepsilon, \delta, m, n$ (Thm. 4.4)
2: Form the synthetic distribution:

$$P_{syn} \leftarrow (1 - \beta^*)\,\bar{P}_{prv} + \beta^*\,P^*_{nse}$$

3: Draw $m$ i.i.d. samples from $P_{syn}$ to obtain $D_{\text{syn}}$
4: **return** $D_{syn}$

---

namely, $\max_i |\bar{P}^{[i]}_{prv} - \bar{P}'^{[i]}_{prv}| \leq s$ for some $s > 0$, where $\bar{P}^{[i]}_{prv}$ denotes the $i$-th coordinate of $\bar{P}_{prv}$. In this setting, the domain-aware noise distribution continues to take the form of a floor-raised version of $\bar{P}_{pub}$, while the minimum mixing weight $\beta$ is increased as we lose the two-point sparsity structure of neighboring histograms (see Appendix G). The same principles suggest potential extensions of PUBMIX to parametric distribution families, such as Gaussian mixtures.

## 4. Main Results: Technical Details

In this section, we formalize the arguments made in Sec. 3. We begin by deriving a condition on the mixing weight $\beta$ that guarantees DP for a given noise distribution $P_{nse}$.

**Theorem 4.1** (Feasible mixing weights $\beta$). *Fix $\varepsilon > 0$, $\delta \in (0, 1)$, private dataset size $n$, and the required number of synthetic samples $m \geq 1$. Let $P_{nse} \in \Delta_d$ be any noise distribution and denote the two smallest probability masses of $P_{nse}$ by $N_{min_1}$ and $N_{min_2}$, with $N_{min_1} \leq N_{min_2}$. Then, for every $\beta \geq \beta_{\min}(N_{min_1}, N_{min_2})$, the $m$ synthetic samples drawn from $P_{syn}$ in (1) satisfy the DP constraint in (2), where $\beta_{min}(N_{min_1}, N_{min_2})$ is the solution to $\beta$ in:*

$$\max\{h_1(\beta, \varepsilon, n, m, N_{min_1}, N_{min_2}),$$
$$h_2(\beta, \varepsilon, n, m, N_{min_1}, N_{min_2})\} = \delta \quad (6)$$

*where*

$$h_1(\beta, \varepsilon, n, m, N_{min_1}, N_{min_2})$$
$$= \sum_{\substack{\lambda,\mu \geq 0 \\ \lambda+\mu \leq m}} \frac{m!}{\lambda!\mu!(m-\lambda-\mu)!} \left(1 - \beta(N_{min_1} + N_{min_2}) - \frac{1-\beta}{n}\right)^{m-\lambda-\mu}$$
$$\left[\left(\beta N_{min_1} + \frac{1-\beta}{n}\right)^\lambda (\beta N_{min_2})^\mu - e^\varepsilon \left(\beta N_{min_2} + \frac{1-\beta}{n}\right)^\mu (\beta N_{min_1})^\lambda\right]_+ \quad (7)$$

*with $[x]_+ = \max\{x, 0\}$, and $h_2(\beta, \varepsilon, n, m, N_{min_1}, N_{min_2})$ is the same as $h_1(\cdot)$ with $N_{min_1}$ and $N_{min_2}$ swapped.*

The proof of Theorem 4.1 is given in App. C. Since we need the mixing weight $\beta$ to be as small as possible for optimal utility (see Fig. 1a), for any given $P_{nse}$, the optimum mixing weight is the corresponding $\beta_{min}(N_{min_1}, N_{min_2})$ from Theorem 4.1.

Next, we analyze the optimal noise distribution $P^*_{nse}$ that solves (3). For this, we first define *Water-filling* and *Maximum-invariance floor raising*.

**Definition 4.2.** [Water-Filling (WF)] Let $A$ be a probability mass function (PMF) on $\Omega = \{\omega_1, \ldots, \omega_d\}$ with $A_i := A(\omega_i)$. Assume without loss of generality that the indices are arranged such that $A_1 \leq A_2 \leq \ldots \leq A_d$. Let $t \in [0, 1/d]$ be any constant satisfying $A_1 \leq \ldots \leq A_k \leq t < A_{k+1} \leq \ldots \leq A_d$ for some $k < d$. Then, the $t$-water-filled version of $A$, denoted by $B = \text{WF}(A, t)$ is given by,

$$B_i = \max\{t, A_i - \tau\}, \quad \forall i \in \{1, \ldots, d\} \quad (8)$$

where $\tau = \frac{t(k+\ell) - \sum_{i=1}^{k+\ell} A_i}{d-k-\ell}$ is a unique constant satisfying $\sum_{i=1}^d B_i = 1$ and $A_{k+\ell} - t \leq \tau < A_{k+\ell+1} - t$ for a unique $\ell \in \{0, 1, \ldots, d-k-1\}$.

**Definition 4.3.** [Maximum-Invariance Floor Raising (MIFR)] Let $A$ be a probability mass function on $\Omega = \{\omega_1, \ldots, \omega_d\}$ with $A_i := A(\omega_i)$. Fix $t \in [0, 1/d]$. The *maximum-invariance floor raised* version of $A$ at level $t$ is given by $B = \text{MIFR}(A, t)$ with coordinates

$$B_i = \begin{cases} t, & i \in L, \\ b_i, & i \in U^\star, \\ A_i, & i \notin L \cup U^\star, \end{cases}$$

where $L := \{i : A_i \leq t\}$ and $L^c := \{i : A_i > t\}$. The donor set $U^\star \subseteq L^c$ is defined as

$$U^\star = \arg\min_{U \subseteq L^c} \sum_{i \in U} A_i \quad \text{s.t.} \quad \sum_{i \in U}(A_i - t) \geq \sum_{i \in L}(t - A_i).$$

Finally, the values $\{b_i\}_{i \in U^\star}$ satisfy $b_i \geq t, \forall i \in U^\star$ and

$$\sum_{i \in U^\star}(A_i - b_i) = \sum_{i \in L}(t - A_i),$$

so that the total mass removed from indices in $U^\star$ exactly matches the total mass added on $L$.

Both WF and MIFR (Def. 4.2 and 4.3) raise every probability mass below the threshold $t$ up to $t$. The extra mass added in this step is then removed from entries above $t$, while ensuring no entry is reduced below $t$. In this way, both procedures produce a modified PMF $B$ that satisfies $B_i \geq t$ and attains the minimum possible TV distance to the original PMF $A$. The two methods differ in how they choose which

entries above $t$ to decrease. WF spreads the required mass removal across all entries above $t$ in a uniform "leveling" manner until some reach $t$. In contrast, MIFR concentrates the mass removal on a carefully chosen subset of entries above $t$ so as to keep as many coordinates unchanged as possible, i.e., it maximizes the invariant mass $\sum_{i:B_i=A_i} A_i$. An illustration of Def. 4.2 and Def. 4.3 is given in Fig. 3.

We provide an asymptotically optimal solution to (3) in Theorem 4.4. For a fixed $P_{nse}$, we write the corresponding minimum mixing weight $\beta_{min}$ from Theorem 4.1 as $\beta_{min}(N_{min_1}, N_{min_2})$, highlighting its dependence on $P_{nse}$ through its two minimum masses $N_{min_1}$ and $N_{min_2}$. Equivalently, for any $0 \leq z_1 \leq 1/d$ and $z_1 \leq z_2 \leq \frac{1-z_1}{d-1}$, $\beta_{min}(z_1, z_2)$ denotes the threshold corresponding to any $P_{nse}$ whose two minimum entries equal $z_1$ and $z_2$ (the dependence on $P_{nse}$ is only through these minimum elements).

The original problem in (3) can be equivalently written as,

$$\min_{\substack{P_{nse} \in \Delta_d}} \min_{\substack{\beta \in [0,1] \\ \beta \geq \beta_{min}(N_{min_1}, N_{min_2})}} \max_{\bar{P}_{prv} \in \mathcal{B}_{\gamma, \bar{P}_{pub}}} \mathrm{TV}(\bar{P}_{prv}, P_{syn}) \tag{9}$$

which can be bounded as:

$$\min_{\substack{P_{nse} \in \Delta_d}} \min_{\substack{\beta \in [0,1] \\ \beta \geq \beta_{min}(N_{min_1}, N_{min_2})}} \max_{\bar{P}_{prv} \in \mathcal{B}_{\gamma, \bar{P}_{pub}}} \mathrm{TV}(\bar{P}_{prv}, P_{syn})$$
$$\leq \min_{\substack{P_{nse} \in \Delta_d}} \min_{\substack{\beta \in [0,1] \\ \beta \geq \beta_{min}(N_{min_1}, N_{min_1})}} \max_{\bar{P}_{prv} \in \mathcal{B}_{\gamma, \bar{P}_{pub}}} \mathrm{TV}(\bar{P}_{prv}, P_{syn}) \tag{10}$$

as $N_{min_1} \leq N_{min_2}$ and $\beta(N_{min_1}, N_{min_2})$ is non-increasing in both $N_{min_1}$ and $N_{min_2}$ (see Lemma C.5). In Theorem 4.4, we provide the solution to the upper bound in (10). Notice that $\sum_i P_{nse}^{[i]} = 1$ requires:

$$0 \leq N_{min_1} \leq N_{min_2} \leq \frac{1 - N_{min_1}}{d - 1} \tag{11}$$

which makes $N_{min_2} \to N_{min_1}$ for larger $d$, resulting in $\beta_{min}(N_{min_1}, N_{min_2}) \to \beta_{min}(N_{min_1}, N_{min_1})$. Thus, the solution to the upper bound in (10) is asymptotically optimal for (3) (see App. D.1 for rigorous proof).

**Theorem 4.4** (Asymptotically optimal parameters:). *Let* $\gamma \in (0, 1]$ *be a constant and let* $\bar{P}_{pub} \in \Delta_d$ *be a public distribution with the indices arranged as* $\bar{P}_{pub}^{[1]} \leq \bar{P}_{pub}^{[2]} \leq \ldots \leq \bar{P}_{pub}^{[d]}$. *Fix the privacy parameters* $\varepsilon > 0$, $\delta \in (0, 1)$ *and the required number of synthetic samples* $m \geq 1$. *Then,*

$$\min_{\substack{P_{nse} \in \Delta_d}} \min_{\substack{\beta \in [0,1] \\ \beta \geq \beta_{min}(N_{min_1}, N_{min_1})}} \max_{\bar{P}_{prv} \in \mathcal{B}_{\gamma, \bar{P}_{pub}}} \mathrm{TV}(\bar{P}_{prv}, P_{syn})$$
$$= \min \{\beta_{min}(t_1, t_1) \Phi_1(t_1), \beta_{min}(t_2, t_2) \Phi_2(t_2)\} \tag{12}$$

*and the corresponding optimum* $P_{nse}^*$ *and* $\beta^*$ *are given by,*

$$P_{nse}^* = \begin{cases} \mathrm{MIFR}(\bar{P}_{pub}, t_1), & \frac{\beta_{min}(t_1, t_1)}{\beta_{min}(t_2, t_2)} \leq \frac{\Phi_2(t_2)}{\Phi_1(t_1)} \\ \mathrm{WF}(\bar{P}_{pub}, t_2), & o.w. \end{cases} \tag{13}$$

$$\beta^* = \beta_{min}(N_{min}^*, N_{min}^*) \tag{14}$$

*where* $N_{min}^*$ *is the minimum probability mass of* $P_{nse}^*$, *and* $t_1, t_2, \Phi_1(\cdot), \Phi_2(\cdot)$ *are functions of* $\varepsilon, \delta, m, n, \gamma$ *and* $\bar{P}_{pub}$.

The complete statement of Theorem 4.4 with the characterizations of $t_1, t_2, \Phi_1(\cdot), \Phi_2(\cdot)$ is given in App. D.2. A geometric interpretation of Theorem 4.4 is given in App. D.3, and the proof of Theorem 4.4 is given in App. E.

The **Key insight from Theorem 4.4** is that the optimal noise distribution $P_{nse}^*$ is always a floor-raised version of $\bar{P}_{pub}$, where both $\mathrm{MIFR}(\bar{P}_{pub}, t_1)$ and $\mathrm{WF}(\bar{P}_{pub}, t_2)$ raise small elements of $\bar{P}_{pub}$ to their respective thresholds and redistribute probability mass from larger elements. This design balances two objectives: since $\bar{P}_{prv}$ and $\bar{P}_{pub}$ come from the same domain with typically small $\mathrm{TV}(\bar{P}_{pub}, \bar{P}_{prv})$, choosing $P_{nse}$ close to $\bar{P}_{pub}$ is beneficial. However, many elements in $\bar{P}_{pub}$ can be small (even zero), which would add minimal noise to corresponding $\bar{P}_{prv}^{[i]}$ elements and require a larger mixing weight $\beta$ to maintain privacy, pushing $P_{syn}$ too close to $\bar{P}_{pub}$ (not to $\bar{P}_{prv}$). The optimal $P_{nse}^*$ resolves this tradeoff by remaining close to $\bar{P}_{pub}$ while floor-raising smaller probabilities to guarantee a sufficient $N_{min} = \min_i P_{nse}^{[i]}$, which allows a smaller $\beta$ and yields a $P_{syn}$ that is closer to $\bar{P}_{prv}$.

To see the significance of using domain information via public data, consider the solution to (3) when no public dataset $D_{pub}$ is available, i.e., when $\mathcal{B}_{\gamma, \bar{P}_{pub}} = \Delta_d$.

**Corollary 4.5** (Asymptotic optimality with no $D_{pub}$). *For any* $D_{prv}$, $m \geq 1$, $\varepsilon > 0$ *and* $\delta \in (0, 1)$, *the asymptotic solution to* (3) *with* $\mathcal{B}_{\gamma, \bar{P}_{pub}} = \Delta_d$ *is given by,*

$$\min_{\substack{P_{nse} \in \Delta_d}} \min_{\substack{\beta \in [0,1] \\ \beta \geq \beta_{min}(N_{min_1}, N_{min_1})}} \max_{\bar{P}_{prv} \in \Delta_d} \mathrm{TV}(\bar{P}_{prv}, P_{syn})$$
$$= (1 - 1/d) \beta_{min}(1/d, 1/d) \tag{15}$$

*and the corresponding optimum* $P_{nse}^*$ *and* $\beta^*$ *are given by,*

$$P_{nse}^* = \frac{1}{d}\mathbf{1}_d \quad and \quad \beta^* = \beta_{min}(1/d, 1/d) \tag{16}$$

*where* $\mathbf{1}_d$ *is the all ones vector of size d.*

Corollary 4.5 (proof in App. F) shows that without domain information (i.e., no public data from the same domain as $D_{prv}$), the optimal noise distribution is uniform and $P_{syn}$ is formed by mixing uniform noise with $\bar{P}_{prv}$. We call this mechanism LINMIX. This effectively pulls $P_{syn}$ toward the

uniform distribution and away from $\bar{P}_{prv}$. In contrast, when relevant public data is available, Theorem 4.4 establishes that the optimal noise distribution is close to $\bar{P}_{pub}$. Since $\bar{P}_{pub}$ and $\bar{P}_{prv}$ share similar structural properties, linearly combining them preserves much of this structure, allowing $P_{syn}$ to remain closer to $\bar{P}_{prv}$ while still satisfying the privacy constraint.

# 5. Experiments

In the first part of this section, we illustrate the core ideas of PUBMIX with a simplified setting. In the second part, we integrate PUBMIX into state-of-the-art DP synthetic data generation pipelines (histogram-based) and show improved performance.[4] First, we explain how the proximity parameter $\gamma$ is chosen in both experimental settings. $\gamma$ captures the variation of the domain data: if distributions estimated from different public datasets in the same domain vary substantially, $\gamma$ should be larger. If they are similar, a smaller $\gamma$ is appropriate. In our experiments, we either collect $k$ public datasets from the same domain, or randomly partition the available public dataset into $k$ folds, compute pairwise TV distances between the fold distributions, and use the distribution of these distances to estimate $\gamma$.

## 5.1. Illustrating the Core Concepts

In this section, we use a simplified setting in which the normalized histogram of the private data serves as the distribution estimate, and PUBMIX is used to directly sample synthetic data. To make this setting tractable, we use the 2023 ACS Public Use Microdata Sample (PUMS) dataset (U.S. Census Bureau, 2024) by restricting each record with 7 features: race, sex, class of work, marital status, education level, disability status, and English proficiency. We analyze the quality of the PUBMIX-generated samples in terms of TV distance and downstream-task performance by comparing with the standard Gaussian mechanism and the setting with no public data (LINMIX). We use Texas data as $D_{prv}$ and California data as $D_{pub}$.[5]

Fig. 4 compares the TV distance between private distribution and the empirical distribution of the generated synthetic data. LINMIX performs well for small $m$ but degrades as $m$ increases, while PUBMIX avoids this degradation by leveraging public data.

We assess downstream utility by training a random forest regressor to predict *English proficiency* (values from 1-5) using the remaining features, with the model trained on the generated synthetic samples. Here, we use $80\%$ of Texas

---

[4]Our repository can be found at: https://github.com/sajani-vithana/PUBMIX

[5]We set $\gamma$ to the average pairwise TV distance among state-level distributions from California, Colorado, Florida, and Utah.

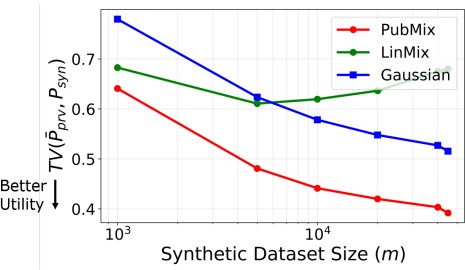

*Figure 4.* Comparison of the total variation (TV) distance between $\bar{P}_{prv}$ and the empirical distribution of the synthetic dataset $D_{syn}$ for $n = 57,717$, $\varepsilon = 5$, and $\delta = 1/n$ on the PUMS dataset.

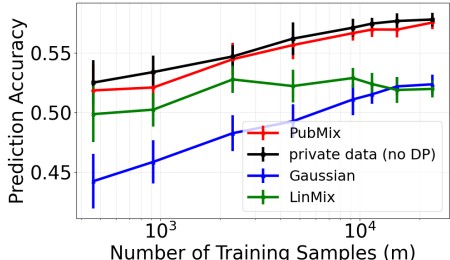

*Figure 5.* Accuracy of a regression model trained on synthetic data by PUBMIX, LINMIX, and the Gaussian mechanism, measured relative to a model trained on real data from the PUMS dataset.

data as $D_{prv}$ to generate the synthetic samples. We then use the remaining $20\%$ of Texas data as the test set to evaluate the performance of the model trained on the synthetic data. Figure 5 compares PUBMIX against a non-private model trained directly on $D_{prv}$, the Gaussian baseline, and LINMIX (no $D_{pub}$).

## 5.2. Evaluation on Existing Pipelines

We next evaluate PUBMIX on two modalities, tabular and text, to demonstrate that it can be used as a *drop-in* replacement for the DP histogram sampling subroutine in existing state-of-the-art synthetic data generation pipelines. Concretely, we adopt Private Evolution (PE) based methods (Tran et al., 2026; Xie et al., 2024), which are recently proposed frameworks generating synthetic data through an iterative process: they repeatedly produce candidate variations of the data and then randomly samples from Gaussian-noised histograms obtained from the variations. In our experiments, we keep the PE pipeline unchanged and replace only the internal DP sampling-from-histograms component with PUBMIX.

### 5.2.1. Tabular: Person Activity Records

**Experimental setup.** We use the Person Activity dataset (Vidulin et al., 2010), which contains sensor-based activity records collected from five users. To mimic a realistic deployment where related public data is available, we construct the private dataset $D_{prv}$ using records from three

*Table 1.* Downstream classification accuracy (%) on Person Activity under varying privacy budgets. We report mean ± std over multiple runs.

| Method | $\varepsilon = 1.0$ | $\varepsilon = 2.0$ | $\varepsilon = 4.0$ |
|---|---|---|---|
| GEM (Liu et al., 2021b) | $28.24\pm0.43$ | $28.51\pm1.84$ | $28.46\pm0.40$ |
| PrivSyn (Zhang et al., 2021) | $27.40\pm2.70$ | $28.41\pm2.03$ | $28.06\pm1.82$ |
| GSD (Liu et al., 2023) | $49.48\pm0.27$ | $50.00\pm0.25$ | $50.46\pm0.54$ |
| AIM (McKenna et al., 2022) | $56.77\pm0.23$ | $58.42\pm0.11$ | $59.12\pm0.30$ |
| JAM-PGM (Fuentes et al., 2024) | $55.96\pm0.24$ | $56.51\pm0.28$ | $57.37\pm0.23$ |
| PMW$_{pub}$ (Liu et al., 2021a) | $49.21\pm0.80$ | $48.96\pm1.05$ | $49.37\pm0.91$ |
| TABPE (Tran et al., 2026) | $60.80\pm0.54$ | $63.22\pm0.48$ | $64.24\pm0.45$ |
| PUBMIX | $\mathbf{66.40}\pm\mathbf{0.49}$ | $\mathbf{66.47}\pm\mathbf{0.42}$ | $\mathbf{66.85}\pm\mathbf{0.08}$ |

*Table 2.* Utility on Yelp Reviews under varying privacy budgets. We report mean ± std over multiple runs. Category classification accuracy (%, higher is better) and Rating RMSE (lower is better). **PE** ($\varepsilon = \infty$) : Category 72.13%, RMSE 0.873.

| Metric | Method | $\varepsilon = 1$ | $\varepsilon = 2$ | $\varepsilon = 4$ |
|---|---|---|---|---|
| **Category** ($\uparrow$) | PE (Xie et al., 2024) | $\mathbf{71.67}\pm\mathbf{0.21}$ | $70.69\pm0.41$ | $69.93\pm0.38$ |
| | PUBMIX | $71.01\pm0.17$ | $\mathbf{71.57}\pm\mathbf{0.34}$ | $\mathbf{72.35}\pm\mathbf{0.20}$ |
| **Rating** ($\downarrow$) | PE (Xie et al., 2024) | $0.90\pm0.025$ | $0.91\pm0.009$ | $0.89\pm0.021$ |
| | PUBMIX | $\mathbf{0.81}\pm\mathbf{0.016}$ | $\mathbf{0.81}\pm\mathbf{0.022}$ | $\mathbf{0.80}\pm\mathbf{0.039}$ |

users, and the public dataset $D_{pub}$ using records from the remaining two users by splitting the original dataset. This split preserves the domain and task semantics while introducing a natural distribution shift across users, matching the intended use of public-data-guided sample generation.

As our primary baseline, we adopt TabPE (Tran et al., 2026), a state-of-the-art DP synthetic data generator for tabular data. Importantly, TABPE relies on a DP histogram-based sampling subroutine as a core primitive in its update procedure. Building on this modular structure, our main comparison is TABPE+PUBMIX, where we *only* replace this DP histogram sampling component in TABPE with PUBMIX while keeping all other components (e.g., model architecture, iterations, and optimization settings) unchanged. This experimental design isolates the benefit of PUBMIX as a drop-in sampling primitive. We also compare PUBMIX against additional tabular DP baselines, including GSD (Liu et al., 2023), AIM (McKenna et al., 2022), GEM (Liu et al., 2021b), PRIVSYN (Zhang et al., 2021), as well as JAM-PGM (Fuentes et al., 2024) and PWM$_{pub}$ (Liu et al., 2021a) that utilize public data. Following prior work (Tran et al., 2026), we evaluate synthetic data utility by training a downstream classifier on synthetic samples and reporting test accuracy of classification performance for human activity.

**Results.** Table 1 reports downstream accuracy across privacy budgets. Across all $\varepsilon \in \{1, 2, 4\}$, TABPE + PUBMIX achieves the best performance, consistently outperforming the PE baseline as well as all other DP tabular methods. Notably, the gains over PE are largest in the stricter privacy regime: PE+PUBMIX improves PE by $+5.80$ points at $\varepsilon = 1$ (66.60 vs. 60.80), and remains advantageous at $\varepsilon = 2$ and $\varepsilon = 4$ ($+2.93$ and $+1.88$ points, respectively). This pattern shows that PUBMIX leverages related public data to improve utility under stronger privacy constraints.

*5.2.2. Text: Yelp Reviews*

**Experimental setup.** We consider a real-world text synthesis setting where the private dataset is Yelp reviews (Yelp Inc., 2023), and the public dataset is Google Local Reviews from 10 states (Li et al., 2022; Yan et al., 2023). This public dataset provides a realistic public signal from a related

but non-identical distribution, reflecting practical scenarios where public web-scale reviews exist but do not exactly match the private domain. Yelp provides two downstream tasks that capture complementary aspects of utility. (i) Category classification: a trained classifier on the synthetic reviews predicts the business category. (ii) Rating prediction: a trained regressor predicts the star rating in $\{1, \ldots, 5\}$. We report classification accuracy and the root mean squared error (RMSE) for the rating, on the test set.

We adopt a PE variant method, tailored for DP text synthesis as our main baseline (Xie et al., 2024). We apply PUBMIX as a plug-in replacement in the DP sampling component of the PE pipeline, while keeping all remaining settings identical. Additional details (length constraints, model configuration, and training schedule) are provided in App. H.

**Results.** Table 2 summarizes utility on Yelp across privacy budgets. Overall, PUBMIX improves the privacy-utility tradeoff over the PE baseline on both tasks. In particular, PUBMIX consistently reduces rating prediction error (RMSE) for all $\varepsilon$, indicating better preservation of the continuous rating signal under DP. These results suggest that public-guided DP sampling can enhance PE-based text synthesis without changing the rest of the pipeline.

## 6. Conclusion

In this paper, we introduced the concept of domain-aware DP mechanisms for synthetic data generation and provided a corresponding theoretical framework. For normalized histograms, we characterized the optimal domain-aware DP mechanism within a class of distribution mixing functions. To this end, we introduced PUBMIX, a public-data-aware privacy mechanism that can be used as a drop-in replacement for Laplace or Gaussian mechanisms in histogram-based data synthesis pipelines. Empirically, we show that incorporating PUBMIX in existing data synthesis pipelines consistently improves the privacy–utility trade-off.

**Limitations and future work:** Our analysis is restricted to linear mixing-based DP mechanisms, requires public data from the same domain, and considers only histogram-based distribution estimators. Future work will extend domain-aware privacy mechanisms to more complex distribution estimators, broadening the scope of PUBMIX's applications.

## Acknowledgements

This work was supported by the National Science Foundation under Grant No: CIF 2231707, CIF 2312667, 2506573, and a grant from the Center for Wireless Intelligence at Georgia Tech. We also acknowledge support from Coefficient Giving and JPMorgan Chase. F.P. Calmon is also affiliated with Google Research as a Visiting Faculty Researcher.

## Impact Statement

This work advances DP synthetic generation by proposing public-data-aware privacy mechanisms. This method incorporates auxiliary public data directly into the privacy-critical sampling step, improving utility while maintaining DP guarantees. By enabling the generation of higher-fidelity synthetic data under a given privacy budget, the proposed approach increases the practicality of privacy-preserving data sharing in domains where data access is limited (e.g., healthcare, finance and user-generated content). At the same time, synthetic data can be misused or misinterpreted. First, since DP protections depend on correct choice and reporting of privacy parameters, loose privacy budgets can undermine protections. Second, synthetic data may still encode and amplify social biases present in the private dataset. Finally, because our method leverages public data, careful governance is needed to ensure the public source is legitimately shareable and properly licensed.

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

# A. Additional Details on Related Work

**On methods using linear mixing:** Certain private prediction variants (Flemings et al., 2024; Ginart et al., 2022) are alternatives to DP training methods such as DP stochastic gradient descent (DP-SGD), where instead of privatizing the model itself, privacy is enforced at inference time by perturbing the model's output. In the LLM setting, this typically involves mixing or projecting the private model's next-token distribution toward a public distribution to limit leakage from memorized private data. While both PubMix and private prediction variants involve linear mixing and public distributions, they differ fundamentally in goal and technical design.

- **Goal:** Private prediction protects the privacy of an LLM's fine-tuning data at inference time, one token prediction at a time. In contrast, PUBMIX introduces domain-aware, sample-aware synthetic data generation from histograms, with the goal of producing m privacy-protected samples whose distribution is as close as possible to the private data distribution.

- **Optimization:** Private prediction methods (Flemings et al., 2024; Ginart et al., 2022) linearly mix private token distributions with a fixed public distribution and optimize only the mixing coefficient to remain as close as possible to the private distribution while satisfying privacy. PUBMIX instead mixes a private normalized histogram with a noise distribution and jointly optimizes both the mixing weight $\beta$ and the noise distribution $P_{nse}$ to minimize the distance to the private distribution. In this sense, PubMix generalizes the optimization step of private prediction by introducing an additional optimization variable, which provides extra flexibility to improve the utility of the mixing step.

- **Role of public data:** In private prediction, the public distribution mainly serves as a source of noise. In PUBMIX, public data is used more structurally: it informs the design of a domain-aware noise distribution. In other words, using the public distribution as the noise distribution is only a special case of PUBMIX. Generally, the optimal noise distribution drifts toward a floor-raised version of the public distribution. This distinction matters especially when the public distribution is sparse (which is common in token distributions), since fixing it directly can force the mixed distribution to be much closer to the public distribution than to the private ones, which degrades utility. PUBMIX avoids this by jointly optimizing the noise distribution and mixing parameter.

While these differences exist between PUBMIX and private prediction methods, the extension of PUBMIX to general discrete distributions beyond histograms (briefly discussed in App. G) can be applied on token distributions. In that sense, PUBMIX can potentially serve as a drop-in privacy module for the linear mixing step in private prediction, provided it is paired with an appropriate privacy accounting across generated tokens.

# B. Proofs of Section 3

**Lemma B.1.** *Let $A = (A_1, \ldots, A_d)$ be a probability mass function (PMF) on a finite set $\Omega = \{\omega_1, \ldots, \omega_d\}$, and let $U = (U_1, \ldots, U_d)$ denote the uniform distribution on $\Omega$, i.e., $U_i = 1/d$ for all $i$. For any threshold $t \in [0, 1/d]$, let $B = \mathrm{WF}(A, t)$ denote the $t$-water-filled version of $A$ as in Definition 4.2. Then, the total variation distance to the uniform distribution does not increase under water-filling, i.e., $\|B - U\|_{\mathrm{TV}} \leq \|A - U\|_{\mathrm{TV}}$.*

*Proof.* Let $A = (A_1, \ldots, A_d)$ be the original PMF, $B = (B_1, \ldots, B_d)$ the water-filled PMF, and $U = (U_1, \ldots, U_d)$ the uniform distribution with $U_i = 1/d$. We also denote by $Q = (Q_1, \ldots, Q_d)$ an arbitrary PMF for intermediate arguments. For any PMF $Q$ on $\Omega$, the total variation distance to the uniform distribution $U$ satisfies

$$\|Q - U\|_{\mathrm{TV}} = \frac{1}{2} \sum_{i=1}^{d} |Q_i - \frac{1}{d}| = \sum_{i: Q_i > \frac{1}{d}} \left( Q_i - \frac{1}{d} \right). \tag{17}$$

Therefore, it suffices to show that

$$\sum_{i: B_i > \frac{1}{d}} \left( B_i - \frac{1}{d} \right) \leq \sum_{i: A_i > \frac{1}{d}} \left( A_i - \frac{1}{d} \right). \tag{18}$$

Recall that $B_i = \max\{t, A_i - \tau\}$, where $\tau$ is chosen such that $\sum_{i=1}^{d} B_i = 1$. We first show that $\tau \geq 0$. Suppose, for the sake of contradiction, that $\tau < 0$. Then $A_i - \tau > A_i$ for all $i$, which implies $B_i \geq A_i - \tau > A_i$ for all $i$. Consequently, $\sum_{i=1}^{d} B_i > \sum_{i=1}^{d} A_i = 1$, which contradicts the normalization condition $\sum_{i=1}^{d} B_i > 1$. Hence, $\tau \geq 0$.

Consider any index $i$ such that $A_i \leq \frac{1}{d}$. Since $t \leq \frac{1}{d}$ and $\tau \geq 0$, we have

$$B_i = \max\{t, A_i - \tau\} \leq \frac{1}{d}. \tag{19}$$

Hence, no coordinate whose original mass is at most $1/d$ can exceed $1/d$ after water-filling.

Consider any index $i$ such that $A_i > \frac{1}{d}$. As $A_i > t$ and $\tau \geq 0$,

$$B_i = A_i - \tau \leq A_i. \tag{20}$$

Hence, any excess mass above $1/d$ is weakly reduced by the water-filling operation.

Combining the above observations, the set $\{i : B_i > \frac{1}{d}\}$ is a subset of $\{i : A_i > \frac{1}{d}\}$, and for each such index, $B_i - \frac{1}{d} \leq A_i - \frac{1}{d}$. Summing over all indices yields

$$\|B - U\|_{\mathrm{TV}} \leq \|A - U\|_{\mathrm{TV}}, \tag{21}$$

which completes the proof. $\qquad\square$

## C. Proof of Theorem 4.1

In this section, we provide the proof of Theorem 4.1. First, we restate Theorem 4.1, and then provide the proof in the following order.

- Rewrite the privacy constraint in (2) in terms of the privacy parameters $\varepsilon, \delta$, required samples $m$, noise distribution $P_{nse}$, mixing coefficient $\beta$, and $a = P'_{syn}(i)$, $b = P'_{syn}(j)$, where $i$ and $j$ are the two distinct indices at which the normalized histograms, $\bar{P}_{prv}$ and $\bar{P}'_{prv}$, of neighboring datasets $D_{prv}$ and $D'_{prv}$ differ. Specifically, we show that the privacy constraint simplifies to:

$$\sup_{a,b} E_{e^\varepsilon}(P^{\otimes m} \| Q^{\otimes m}) \leq \delta \tag{22}$$

  where $P$ and $Q$ are distributions given by $P = \left[a + \frac{1-\beta}{n}, \, b - \frac{1-\beta}{n}, \, 1 - a - b\right]$ and $Q = [a, \, b, \, 1 - a - b]$, and $E_{e^\varepsilon}(\cdot\|\cdot)$ is the $E_\gamma$ divergence.

- We show that $\sup_{a,b} E_{e^\varepsilon}(P^{\otimes m} \| Q^{\otimes m})$ is non-increasing in $a$ and $b$ for any given $\varepsilon, \delta, m, n, \beta, P_{nse}$, and characterize the exact $\sup_{a,b} E_{e^\varepsilon}(P^{\otimes m} \| Q^{\otimes m})$, which leads to Theorem 4.1.

**Theorem 4.1 restated:**[Feasible mixing weights $\beta$] Fix $\varepsilon > 0$, $\delta \in (0, 1)$, private dataset size $n$, and the required number of synthetic samples $m \geq 1$. Let $P_{nse} \in \Delta_d$ be any noise distribution and define $N_{min_1}$ and $N_{min_2}$ to be the two smallest elements of $P_{nse}$ with $N_{min_1} \leq N_{min_2}$. Then, for every $\beta \geq \beta_{\min}(N_{min_1}, N_{min_2})$, the $m$ synthetic samples drawn from $P_{syn}$ in (1) satisfy the DP constraint in (2), where $\beta_{min}(N_{min_1}, N_{min_2})$ is given by the solution to $\beta$ in:

$$\max\{h_1(\beta, \varepsilon, n, m, N_{min_1}, N_{min_2}), h_2(\beta, \varepsilon, n, m, N_{min_1}, N_{min_2})\} = \delta \tag{23}$$

where

$$h_1(\beta, \varepsilon, n, m, N_{min_1}, N_{min_2}) = \sum_{\substack{\lambda,\mu \geq 0 \\ \lambda+\mu \leq m}} \frac{m!}{\lambda!\mu!(m-\lambda-\mu)!} \left(1 - \beta(N_{min_1} + N_{min_2}) - \frac{1-\beta}{n}\right)^{m-\lambda-\mu}$$

$$\left[\left(\beta N_{min_1} + \frac{1-\beta}{n}\right)^\lambda (\beta N_{min_2})^\mu - e^\varepsilon \left(\beta N_{min_2} + \frac{1-\beta}{n}\right)^\mu (\beta N_{min_1})^\lambda\right]_+ \tag{24}$$

where $[x]_+ = \max\{x, 0\}$, and $h_2(\beta, \varepsilon, n, m, N_{min_1}, N_{min_2})$ is the same as $h_1(\beta, \varepsilon, n, m, N_{min_1}, N_{min_2})$ with $N_{min_1}$ and $N_{min_2}$ swapped. Moreover, $\beta_{min}(N_{min_1}, N_{min_2})$ is decreasing in $N_{min_1}$ and $N_{min_2}$.

**Lemma C.1.** *For any fixed privacy parameters $\varepsilon > 0$ and $\delta \in (0, 1)$, required number of synthetic samples $m$, and any fixed noise distribution $P_{nse} \in \Delta_d$, the privacy constraint in (2) is simplified to:*

$$\sup_{(a,b)\in\mathcal{F}(\beta,P_{nse})} E_{e^\varepsilon}(P^{\otimes m}\|Q^{\otimes m}) = \sup_{(a,b)\in\mathcal{F}(\beta,P_{nse})} f(a,b,\varepsilon,\beta) \leq \delta \tag{25}$$

*where $E_{e^\varepsilon}(P^{\otimes m}\|Q^{\otimes m})$ denotes the hockey-stick divergence ($E_\gamma$ divergence) between the product distributions $P^{\otimes m}$ and $Q^{\otimes m}$ with $P$ and $Q$ denoting the three-point distributions $P = \left[a + \frac{1-\beta}{n}, \ b - \frac{1-\beta}{n}, \ 1 - a - b\right]$ and $Q = [a, \ b, \ 1 - a - b]$. $\mathcal{F}(\beta, P_{nse})$ is the feasible set of $(a, b)$, and,*

$$f(a,b,\varepsilon,\beta) = \sum_{\substack{\lambda,\mu\geq 0 \\ \lambda+\mu\leq m}} \frac{m!}{\lambda!\mu!(m-\lambda-\mu)!}(1-a-b)^{m-\lambda-\mu}\left[\left(a+\frac{1-\beta}{n}\right)^\lambda\left(b-\frac{1-\beta}{n}\right)^\mu - e^\varepsilon a^\lambda b^\mu\right]_+ \leq \delta \tag{26}$$

**Proof:** Consider the privacy constraint in (2): $\mathbb{P}(D_{syn} \in S) \leq e^\varepsilon\mathbb{P}(D'_{syn} \in S) + \delta, \forall S \subseteq \mathcal{X}^m, \forall D_{prv}, D'_{prv}$, which is equivalent to:

$$\max_{S\subseteq\mathcal{X}^m} \mathbb{P}(D_{syn} \in S) - e^\varepsilon\mathbb{P}(D'_{syn} \in S) \leq \delta, \quad \forall D_{prv}, D'_{prv} \tag{27}$$

Let $\mathbf{Y} = \{y_1, \ldots, y_m\} \in \mathcal{X}^m$ denote any generic candidate set of synthetic samples. Then, we can write the privacy constraint in (27) as:

$$\sum_{\mathbf{Y}\in\mathcal{X}^m} \left[\prod_{k=1}^m P_{syn}(y_k) - e^\varepsilon \prod_{k=1}^m P'_{syn}(y_k)\right]_+ \leq \delta, \quad \forall D_{prv}, D'_{prv} \tag{28}$$

where $[x]_+ = \max\{x, 0\}$. From the construction of $P_{syn}$ in (1), we have,

$$P_{syn}(y) = (1 - \beta)\bar{P}_{prv}(y) + \beta P_{nse}(y) \tag{29}$$
$$P'_{syn}(y) = (1 - \beta)\bar{P}'_{prv}(y) + \beta P_{nse}(y) \tag{30}$$

for any $y \in \mathcal{X}$ where,

$$\bar{P}_{prv}(y) = \begin{cases} \bar{P}'_{prv}(y) + \frac{1}{n}, & y = i \\ \bar{P}'_{prv}(y) - \frac{1}{n}, & y = j \\ \bar{P}'_{prv}(y), & y \neq i, j \end{cases} \tag{31}$$

for some $i, j \in \mathcal{X}$. Since each $y_k$ is sampled i.i.d. from $P_{syn}$ (or $P'_{syn}$), (28) can be written in terms of the $E_\gamma$ divergence between the product distributions $P_{syn}^{\otimes m}$ and $P'^{\otimes m}_{syn}$.

$$E_{e^\varepsilon}(P_{syn}^{\otimes m}\|P'^{\otimes m}_{syn}) \leq \delta, \quad \forall D_{prv}, D'_{prv} \tag{32}$$

For product distributions, the likelihood ratio of any realization $\mathbf{Y} = (y_1, \ldots, y_m)$ is $\prod_{k=1}^m \frac{P_{syn}(y_k)}{P'_{syn}(y_k)}$, which depends only on the number of times $i$ and $j$ appear among the $m$ samples, since $\frac{P_{syn}(y)}{P'_{syn}(y)} = 1$ for all $y \notin \{i, j\}$. Therefore, $E_{e^\varepsilon}(P_{syn}^{\otimes m}\|P'^{\otimes m}_{syn})$ depends on $P_{syn}$ and $P'_{syn}$ only through the probabilities assigned to $i, j$, and the total remaining mass. With the notation $P'_{syn}(i) = a$, $P'_{syn}(j) = b$, $P_{syn}(i) = a + \frac{1-\beta}{n}$ and $P_{syn}(j) = b - \frac{1-\beta}{n}$, we can equivalently write (32) as:

$$E_{e^\varepsilon}(P^{\otimes m}\|Q^{\otimes m}) \leq \delta, \quad \forall D_{prv}, D'_{prv} \tag{33}$$

where $P$ and $Q$ are three-point distributions given by $P = \left[a + \frac{1-\beta}{n}, \ b - \frac{1-\beta}{n}, \ 1 - a - b\right]$ and $Q = [a, \ b, \ 1 - a - b]$, corresponding to the probabilities of outcomes $i, j$, and the aggregate remainder in $P_{syn}$ and $P'_{syn}$, respectively. Moreover, we can replace the $\forall D_{prv}, D'_{prv}$ term in (33) by the supremum over $(a, b)$ for any given $\beta$ and $P_{nse}$. This gives the following equivalent form of the privacy constraint in (33):

$$\sup_{(a,b)\in\mathcal{F}(\beta,P_{nse})} E_{e^\varepsilon}(P^{\otimes m}\|Q^{\otimes m}) \leq \delta \tag{34}$$

The numbers of samples "$i$", "$j$", or "anything other than $i, j$" (sampled i.i.d.) follow a multinomial distribution with probabilities $(a + \frac{1-\beta}{n}, b - \frac{1-\beta}{n}, 1 - a - b)$ and $(a, b, 1 - a - b)$, respectively, for $P^{\otimes m}$ and $Q^{\otimes m}$. This simplifies the privacy constraint in (34) to:

$$\sup_{(a,b)\in\mathcal{F}(\beta,P_{nse})} \sum_{\substack{\lambda,\mu\geq 0 \\ \lambda+\mu\leq m}} \frac{m!}{\lambda!\mu!(m-\lambda-\mu)!}(1-a-b)^{m-\lambda-\mu}\left[\left(a+\frac{1-\beta}{n}\right)^{\lambda}\left(b-\frac{1-\beta}{n}\right)^{\mu} - e^{\varepsilon}a^{\lambda}b^{\mu}\right]_{+} \leq \delta \quad (35)$$

This completes the proof of Lemma C.1. ∎

**Lemma C.2.** *For any fixed $P_{nse} \in \Delta_d$ and $\beta \in [0,1]$, $E_{e^{\varepsilon}}(P^{\otimes m}\|Q^{\otimes m})$ is non-increasing in $a$ and $b$.*

**Proof:** First, we show that $E_{e^{\varepsilon}}(P^{\otimes m}\|Q^{\otimes m})$ is non-increasing in $a$ for any fixed $b$. For this, define $P_a = \left[a+\frac{1-\beta}{n}, b-\frac{1-\beta}{n}, 1-a-b\right]$ and $Q_a = [a, b, 1-a-b]$, and similarly $P_{a+\Delta} = \left[a+\Delta+\frac{1-\beta}{n}, b-\frac{1-\beta}{n}, 1-a-\Delta-b\right]$ and $Q_{a+\Delta} = [a+\Delta, b, 1-a-\Delta-b]$ for some small $\Delta > 0$. Consider the following stochastic channel $T$, applied independently to each sample $y$, sampled from $P_a$ and $Q_a$:

$$T(y' \mid y) = \begin{cases} 1, & y' = y, \quad y \in \{i,j\}, \\ \dfrac{\Delta}{1-a-b}, & y' = i, \quad y = \text{"other"}, \\ 1 - \dfrac{\Delta}{1-a-b}, & y' = \text{"other"}, \quad y = \text{"other"}. \end{cases} \quad (36)$$

where "other" denotes the combined event of all $y \neq i, j$. The output distributions $TP_a$ and $TQ_a$ are given by:

$$(TP_a)(i) = a + \frac{1-\beta}{n} + \Delta, \quad (37)$$

$$(TP_a)(j) = b - \frac{1-\beta}{n}, \quad (38)$$

$$(TP_a)(\text{"other"}) = 1 - a - b - \Delta, \quad (39)$$

and

$$(TQ_a)(i) = a + \Delta, \quad (40)$$

$$(TQ_a)(j) = b, \quad (41)$$

$$(TQ_a)(\text{"other"}) = 1 - a - b - \Delta. \quad (42)$$

These match $P_{a+\Delta}$ and $Q_{a+\Delta}$, respectively:

$$TP_a = P_{a+\Delta} \quad \text{and} \quad TQ_a = Q_{a+\Delta}. \quad (43)$$

Since the channel $T$ is applied to all $m$ samples independently and identically, we have

$$T^{\otimes m} : P_a^{\otimes m} \mapsto P_{a+\Delta}^{\otimes m} \quad \text{and} \quad T^{\otimes m} : Q_a^{\otimes m} \mapsto Q_{a+\Delta}^{\otimes m}. \quad (44)$$

By the data processing inequality for the hockey-stick divergence (Sason & Verdú, 2016), which states that $E_\gamma(TP\|TQ) \leq E_\gamma(P\|Q)$ for any stochastic channel $T$, we conclude that

$$E_{e^{\varepsilon}}(P_{a+\Delta}^{\otimes m}\|Q_{a+\Delta}^{\otimes m}) \leq E_{e^{\varepsilon}}(P_a^{\otimes m}\|Q_a^{\otimes m}). \quad (45)$$

This establishes that $E_{e^{\varepsilon}}(P^{\otimes m}\|Q^{\otimes m})$ is non-increasing in $a$. An analogous argument, with the channel $T$ modified to move mass from outcome "other" to outcome $j$ instead of outcome $i$, shows that $E_{e^{\varepsilon}}(P^{\otimes m}\|Q^{\otimes m})$ is also non-increasing in $b$.

∎

From Lemma C.2, for fixed $\varepsilon$, $P_{nse}$, and $\beta$, the maximum $E_{e^\varepsilon}(P^{\otimes m}\|Q^{\otimes m})$ is achieved at the smallest values of $a$ and $b$, denoted by $a_{min}$ and $b_{min}$. Recall that $a = P'_{syn}(i)$ and $b = P'_{syn}(j)$ where,

$$P'_{syn}(y) = (1 - \beta)\bar{P}'_{prv}(y) + \beta P_{nse}(y), \quad \forall y \in \mathcal{X} \tag{46}$$

Thus, the smallest $a$ and $b$ for fixed $\beta$ and $P_{nse}$ are given by,

$$a_{min} = \beta N_{min_1} \tag{47}$$

$$b_{min} = \frac{1 - \beta}{n} + \beta N_{min_2} \tag{48}$$

or in the reverse order:

$$a'_{min} = \beta N_{min_2} \tag{49}$$

$$b'_{min} = \frac{1 - \beta}{n} + \beta N_{min_1} \tag{50}$$

where $N_{min_1}$ and $N_{min_2}$ are the two smallest elements of $P_{nse}$. This simplifies the privacy constraint in Lemma C.1 to:

$$\max\{f(a_{min}, b_{min}, \varepsilon, \beta), f(a'_{min}, b'_{min}, \varepsilon, \beta)\} \le \delta \tag{51}$$

Thus, for given $\varepsilon, \delta, m$ and $P_{nse}$, the optimum mixing parameter is given by,

$$\beta_{min}(P_{nse}) = \inf\left\{\beta \in [0, 1] : \max\{f(a_{min}, b_{min}, \varepsilon, \beta), f(a'_{min}, b'_{min}, \varepsilon, \beta)\} \le \delta\right\} \tag{52}$$

where we have ignored the dependency of $\beta$ on $\varepsilon, \delta$ and $m$ since they are fixed for a given setting, and $P_{nse}$ is the only variable that is yet to be optimized. Note that $\beta_{min}(P_{nse})$ depends on $P_{nse}$ only though its two smallest elements. Thus, for a given $P_{nse}$, the minimum mixing coefficient is given by:

$$\beta_{min}(N_{min_1}, N_{min_2}) = \inf\left\{\beta \in [0, 1] : \max\{f(a_{min}, b_{min}, \varepsilon, \beta), f(a'_{min}, b'_{min}, \varepsilon, \beta)\} \le \delta\right\} \tag{53}$$

where

$$f(a_{min}, b_{min}, \varepsilon, \beta) = \sum_{\substack{\lambda,\mu \ge 0 \\ \lambda+\mu \le m}} \frac{m!}{\lambda!\mu!(m - \lambda - \mu)!} \left(1 - \beta(N_{min_1} + N_{min_2}) - \frac{1 - \beta}{n}\right)^{m-\lambda-\mu}$$
$$\left[\left(\beta N_{min_1} + \frac{1 - \beta}{n}\right)^\lambda (\beta N_{min_2})^\mu - e^\varepsilon \left(\beta N_{min_2} + \frac{1 - \beta}{n}\right)^\mu (\beta N_{min_1})^\lambda\right]_+ \tag{54}$$

and

$$f(a'_{min}, b'_{min}, \varepsilon, \beta) = \sum_{\substack{\lambda,\mu \ge 0 \\ \lambda+\mu \le m}} \frac{m!}{\lambda!\mu!(m - \lambda - \mu)!} \left(1 - \beta(N_{min_1} + N_{min_2}) - \frac{1 - \beta}{n}\right)^{m-\lambda-\mu}$$
$$\left[\left(\beta N_{min_2} + \frac{1 - \beta}{n}\right)^\lambda (\beta N_{min_1})^\mu - e^\varepsilon \left(\beta N_{min_1} + \frac{1 - \beta}{n}\right)^\mu (\beta N_{min_2})^\lambda\right]_+ \tag{55}$$

Next, we show that $f(a_{min}, b_{min}, \varepsilon, \beta)$ and $f(a'_{min}, b'_{min}, \varepsilon, \beta)$ are decreasing in $\beta$. Thus, for any $P_{nse}$ with a fixed $N_{min_1}$ and $N_{min_2}$, any $\beta \ge \beta(N_{min_1}, N_{min_2})$ satisfies the privacy constraint in (2).

**Lemma C.3.** *For any fixed $N_{min_1}, N_{min_2}$ and $\varepsilon > 0$, $\max\{f(a_{min}, b_{min}, \varepsilon, \beta), f(a'_{min}, b'_{min}, \varepsilon, \beta)\}$ is decreasing in $\beta$.*

**Proof:** From the monotonicity of $E_{e^\varepsilon}(P^{\otimes m}\|Q^{\otimes m})$ in $a$ and $b$ established in the proof of Lemma C.2, for any fixed $\beta, \varepsilon$, $N_{min_1}$ and $N_{min_2}$, we have the privacy constraint in (2) simplified to:

$$\sup_{D_{prv}, D'_{prv}} E_{e^\varepsilon}(P^{\otimes m}_{syn}\|P'^{\otimes m}_{syn}) = \sup_{(a,b) \in \mathcal{F}(\beta, N_{min_1}, N_{min_2})} E_{e^\varepsilon}(P^{\otimes m}\|Q^{\otimes m}) = \max\{E_{e^\varepsilon}(P^{\otimes m}_*\|Q^{\otimes m}_*), E_{e^\varepsilon}(P^{\otimes m}_\#\|Q^{\otimes m}_\#)\} \le \delta$$
$$\tag{56}$$

where

$$P_* = \left[a_{min} + \frac{1-\beta}{n}, \ b_{min} - \frac{1-\beta}{n}, \ 1 - a_{min} - b_{min}\right] \tag{57}$$

$$Q_* = [a_{min}, \ b_{min}, \ 1 - a_{min} - b_{min}] \tag{58}$$

and

$$P_\# = \left[a'_{min} + \frac{1-\beta}{n}, \ b'_{min} - \frac{1-\beta}{n}, \ 1 - a'_{min} - b'_{min}\right] \tag{59}$$

$$Q_\# = [a'_{min}, \ b'_{min}, \ 1 - a'_{min} - b'_{min}] \tag{60}$$

with $a_{min} = \beta N_{min_1}$, $b_{min} = \frac{1-\beta}{n} + \beta N_{min_2}$ and $a'_{min} = \beta N_{min_2}$ and $b'_{min} = \frac{1-\beta}{n} + \beta N_{min_1}$. Note that $f(a_{min}, b_{min}, \varepsilon, \beta) = E_{e^\varepsilon}(P_*^{\otimes m} \| Q_*^{\otimes m})$ and $f(a'_{min}, b'_{min}, \varepsilon, \beta) = E_{e^\varepsilon}(P_\#^{\otimes m} \| Q_\#^{\otimes m})$. Moreover, $a_{min}, a'_{min}, b_{min}, b'_{min}$ are functions of $\beta$. Suppose for a given $N_{min_1}, N_{min_2}, \beta_0$ satisfies:[6]

$$\max\{f(a_{min}(\beta_0), b_{min}(\beta_0), \varepsilon, \beta_0), f(a'_{min}(\beta_0), b'_{min}(\beta_0), \varepsilon, \beta_0)\} \le \delta \tag{61}$$

In other words, the privacy constraint holds for all neighboring datasets at $\beta = \beta_0$. For any $\beta > \beta_0$, we can write:

$$
\begin{aligned}
P_{syn,\beta} &= (1-\beta)\bar{P}_{prv} + \beta P_{nse} \\
&= \frac{1-\beta}{1-\beta_0}\left[(1-\beta_0)\bar{P}_{prv} + \beta_0 P_{nse}\right] + \left(1 - \frac{1-\beta}{1-\beta_0}\right)P_{nse} \\
&= \alpha\, P_{syn,\beta_0} + (1-\alpha)\, P_{nse}
\end{aligned}
\tag{62}
$$

where $\alpha = \frac{1-\beta}{1-\beta_0} \in [0, 1]$ and $P_{syn,\beta_0} = (1-\beta_0)\bar{P}_{prv} + \beta_0 P_{nse}$. Similarly, for any neighboring dataset:

$$P'_{syn,\beta} = \alpha\, P'_{syn,\beta_0} + (1-\alpha)\, P_{nse}. \tag{63}$$

That is, both $P_{syn,\beta}$ and $P'_{syn,\beta}$ are obtained by applying the same stochastic channel $T$ to $P_{syn,\beta_0}$ and $P'_{syn,\beta_0}$, respectively, where $T$ takes the synthetic distribution as input and outputs a sample from the synthetic distribution with mixing parameter $\beta_0$ with probability $\alpha$ and a sample from $P_{nse}$ with probability $1-\alpha$. Since $T$ is applied independently to each of the $m$ samples, by the data processing inequality for the hockey-stick divergence:

$$E_{e^\varepsilon}(P_{syn,\beta}^{\otimes m} \| P'^{\otimes m}_{syn,\beta}) \le E_{e^\varepsilon}(P_{syn,\beta_0}^{\otimes m} \| P'^{\otimes m}_{syn,\beta_0}) \tag{64}$$

Since (64) holds for every neighboring pair $(D_{prv}, D'_{prv})$, and the channel $T$ does not depend on the choice of neighboring pair, we can take the supremum over all neighboring pairs on both sides:

$$\sup_{D_{prv}, D'_{prv}} E_{e^\varepsilon}(P_{syn,\beta}^{\otimes m} \| P'^{\otimes m}_{syn,\beta}) \le \sup_{D_{prv}, D'_{prv}} E_{e^\varepsilon}(P_{syn,\beta_0}^{\otimes m} \| P'^{\otimes m}_{syn,\beta_0}) \le \delta. \tag{65}$$

where the last inequality follows from the fact that privacy is guaranteed at $\beta = \beta_0$. From the monotonicity of $E_{e^\varepsilon}(P^{\otimes m} \| Q^{\otimes m})$ in $a$ and $b$ established in (45), the supremum over all neighboring pairs at any fixed $\beta$ is achieved at $(a_{min}, b_{min})$ or $(a'_{min}, b'_{min})$ for a given $P_{nse}$. Therefore,

$$
\begin{aligned}
\max\{f(a_{min}(\beta), b_{min}(\beta), \varepsilon, \beta), f(a'_{min}(\beta), b'_{min}(\beta), \varepsilon, \beta)\} &\le \max\{f(a_{min}(\beta_0), b_{min}(\beta_0), \varepsilon, \beta_0), f(a'_{min}(\beta_0), b'_{min}(\beta_0), \varepsilon, \beta_0)\} \\
&\le \delta,
\end{aligned}
\tag{66}
$$

establishing that $\max\{f(a_{min}, b_{min}, \varepsilon, \beta), f(a'_{min}, b'_{min}, \varepsilon, \beta)\}$ is decreasing in $\beta$. ∎

With Lemma C.3 and the characterization of $\beta_{min}(N_{min_1}, N_{min_2})$ in (53), we claim that for any fixed $N_{min_1}, N_{min_2}$ and $\varepsilon$, any $\beta \ge \beta(N_{min_1}, N_{min_2})$ satisfies the privacy constraint in (2). To simplify the process of computing $\beta_{min}(N_{min_1}, N_{min_2})$ for given $N_{min_1}, N_{min_2}$ from (53), we next show that $\max\{f(a_{min}(\beta), b_{min}(\beta), \varepsilon, \beta), f(a'_{min}(\beta), b'_{min}(\beta), \varepsilon, \beta)\}$ is continuous in $\beta$ (see Lemma C.4). Then, together with the fact that $\max\{f(a_{min}(\beta), b_{min}(\beta), \varepsilon, \beta), f(a'_{min}(\beta), b'_{min}(\beta), \varepsilon, \beta)\}$ is decreasing in $\beta$ (Lemma C.3), we find $\beta_{min}(N_{min_1}, N_{min_2})$ for any given $N_{min_1}, N_{min_2}$ by solving:

$$\max\{f(a_{min}(\beta), b_{min}(\beta), \varepsilon, \beta), f(a'_{min}(\beta), b'_{min}(\beta), \varepsilon, \beta)\} = \delta \tag{67}$$

---

[6] Since $a_{min}, a'_{min}, b_{min}, b'_{min}$ are functions of $\beta$, we make it explicit by writing $f(a_{min}, b_{min}, \varepsilon, \beta) = f(a_{min}(\beta), b_{min}(\beta), \varepsilon, \beta)$ and $f(a'_{min}, b'_{min}, \varepsilon, \beta) = f(a'_{min}(\beta), b'_{min}(\beta), \varepsilon, \beta)$.

**Lemma C.4.** *For any fixed $N_{min_1}, N_{min_2}$ and $\varepsilon$, $\max\{f(a_{min}(\beta), b_{min}(\beta), \varepsilon, \beta), f(a'_{min}(\beta), b'_{min}(\beta), \varepsilon, \beta)\}$ is continuous in $\beta$.*

**Proof:** Consider:

$$f(a_{min}(\beta), b_{min}(\beta), \varepsilon, \beta) = \sum_{\substack{\lambda, \mu \geq 0 \\ \lambda + \mu \leq m}} \frac{m!}{\lambda! \mu! (m - \lambda - \mu)!} \left(1 - \beta(N_{min_1} + N_{min_2}) - \frac{1 - \beta}{n}\right)^{m - \lambda - \mu}$$
$$\left[\left(\beta N_{min_1} + \frac{1 - \beta}{n}\right)^{\lambda} (\beta N_{min_2})^{\mu} - e^{\varepsilon}\left(\beta N_{min_2} + \frac{1 - \beta}{n}\right)^{\mu} (\beta N_{min_1})^{\lambda}\right]_{+} \quad (68)$$

Each term in the summation is indexed by a fixed $(\lambda, \mu)$ with $\lambda, \mu \geq 0$ and $\lambda + \mu \leq m$, and takes the form

$$g_{\lambda, \mu}(\beta) = \frac{m!}{\lambda! \mu! (m - \lambda - \mu)!} c(\beta)^{m - \lambda - \mu} \left[r_1(\beta)^{\lambda} q_2(\beta)^{\mu} - e^{\varepsilon} r_2(\beta)^{\mu} q_1(\beta)^{\lambda}\right]_{+} \quad (69)$$

where $c(\beta) = 1 - \beta(N_{min_1} + N_{min_2}) - \frac{1-\beta}{n}$, $q_i(\beta) = \beta N_{min_i}$, and $r_i(\beta) = \beta N_{min_i} + \frac{1-\beta}{n}$. Each of $c(\beta)$, $q_i(\beta)$, and $r_i(\beta)$ is continuous in $\beta$ on $[0, 1]$, and therefore $h(\beta) \triangleq r_1(\beta)^{\lambda} q_2(\beta)^{\mu} - e^{\varepsilon} r_2(\beta)^{\mu} q_1(\beta)^{\lambda}$ is also continuous in $\beta$. Since the function $x \mapsto [x]_{+} = \max\{0, x\}$ is continuous, the composition $[h(\beta)]_{+}$ is continuous. It follows that each $g_{\lambda, \mu}(\beta)$ is a product of continuous functions and hence continuous. Since the summation is over a finite set of $(\lambda, \mu)$ pairs that does not depend on $\beta$, the function

$$f(a_{min}(\beta), b_{min}(\beta), \varepsilon, \beta) = \sum_{\substack{\lambda, \mu \geq 0 \\ \lambda + \mu \leq m}} g_{\lambda, \mu}(\beta) \quad (70)$$

is a finite sum of continuous functions, and is therefore continuous in $\beta$. Similarly, $f(a'_{min}(\beta), b'_{min}(\beta), \varepsilon, \beta)$ is also continuous in $\beta$. Thus, the maximum of the two functions is also continuous in $\beta$. ∎

**Lemma C.5.** $\beta_{min}(N_{min_1}, N_{min_2})$ *as defined in* (53) *is non-increasing in both $N_{min_1}$ and $N_{min_2}$.*

**Proof:** Recall the two worst-case pairs:

$$(a_{min}, b_{min}) = \left(\beta N_{min_1}, \frac{1-\beta}{n} + \beta N_{min_2}\right), \quad (71)$$

$$(a'_{min}, b'_{min}) = \left(\beta N_{min_2}, \frac{1-\beta}{n} + \beta N_{min_1}\right). \quad (72)$$

All $a_{min}, b_{min}, a'_{min}, b'_{min}$ are linear and non-decreasing in $N_{min_1}$ and $N_{min_2}$ as $\beta \in [0, 1]$. Since $f(a, b, \varepsilon, \beta)$ is non-increasing in $a$ and $b$ (Lemma C.2), it follows that for any fixed $\beta$, $f(a_{min}, b_{min}, \varepsilon, \beta)$ and $f(a'_{min}, b'_{min}, \varepsilon, \beta)$ are each non-increasing in $N_{min_1}$ and $N_{min_2}$. Therefore their maximum,

$$g(\beta, N_{min_1}, N_{min_2}) := \max\{f(a_{min}, b_{min}, \varepsilon, \beta), f(a'_{min}, b'_{min}, \varepsilon, \beta)\}, \quad (73)$$

is also non-increasing in $N_{min_1}$ and $N_{min_2}$ for any fixed $\beta$.

Now fix any $N_{min_1} \leq N'_{min_1}$ (the argument for $N_{min_2}$ is identical). Let $\beta^* = \beta_{min}(N_{min_1}, N_{min_2})$ in (53), so that by definition:

$$g(\beta^*, N_{min_1}, N_{min_2}) \leq \delta. \quad (74)$$

Since $g$ is non-increasing in $N_{min_1}$:

$$g(\beta^*, N'_{min_1}, N_{min_2}) \leq g(\beta^*, N_{min_1}, N_{min_2}) \leq \delta, \quad (75)$$

Therefore, $\beta^*$ is in the feasible range in (53):

$$\beta^* \in \{\beta \in [0, 1] : g(\beta, N'_{min_1}, N_{min_2}) \leq \delta\} \quad (76)$$

Since $\beta_{min}(N'_{min_1}, N_{min_2})$ is the infimum over all feasible $\beta$:

$$\beta_{min}(N'_{min_1}, N_{min_2}) \leq \beta^* = \beta_{min}(N_{min_1}, N_{min_2}). \quad (77)$$

Hence $\beta_{min}(N_{min_1}, N_{min_2})$ is non-increasing in $N_{min_1}$, and by the same argument, in $N_{min_2}$. ∎

This completes the proof of Theorem 4.1.

# D. Details from Section 4

## D.1. Proof of Asymptotic Optimality

The optimization in (10) replaces $\beta_{\min}(N_{\min_1}, N_{\min_2})$ by the more conservative quantity $\beta_{\min}(N_{\min_1}, N_{\min_1})$. Since $N_{\min_2} \geq N_{\min_1}$ and $\beta_{\min}$ is non-increasing in its arguments (Lemma C.5), this yields an upper bound on the original objective. Moreover, the simplex constraint $N_{min_1} + (d-1)N_{min_2} \leq 1$ implies:

$$0 \leq N_{\min_2} - N_{\min_1} \leq \frac{1}{d-1}. \tag{78}$$

**Lemma D.1.** *For any fixed $N_{min_1} \in [0, 1/d]$, $\beta_{\min}(N_{min_1}, N_{min_2})$ is continuous in $N_{min_2}$. Consequently,*

$$\sup_{P_{\mathrm{nse}} \in \Delta_d} |\beta_{\min}(N_{\min_1}, N_{\min_2}) - \beta_{\min}(N_{\min_1}, N_{\min_1})| \to 0 \qquad as\ d \to \infty. \tag{79}$$

*Moreover, the optimal value of the upper-bound problem in (10) converges to the optimal value of the original problem. Any optimizer of the upper-bound problem is therefore asymptotically optimal for the original problem.*

*Proof.* Assume that $\beta_{\min}(N_{min_1}, N_{min_2})$ is continuous in $N_{min_2}$. The proof is provided at the end.

Let $N_1(P_{\mathrm{nse}}) := N_{\min_1}$ and $N_2(P_{\mathrm{nse}}) := N_{\min_2}$, where $N_1(P_{\mathrm{nse}}) \leq N_2(P_{\mathrm{nse}})$ are the two smallest coordinates of $P_{\mathrm{nse}}$. Define

$$\Psi(P_{\mathrm{nse}}) := \max_{\bar{P}_{\mathrm{prv}} \in B_{\gamma, \bar{P}_{\mathrm{pub}}}} TV(\bar{P}_{\mathrm{prv}}, P_{\mathrm{nse}}). \tag{80}$$

After optimizing over $\beta$, the objective in the left-hand side of (10) can be written as

$$F_d(P_{\mathrm{nse}}) = \beta_{\min}(N_1(P_{\mathrm{nse}}), N_2(P_{\mathrm{nse}}))\Psi(P_{\mathrm{nse}}), \tag{81}$$

while the upper-bound objective in the right-hand side of (10) is

$$\widetilde{F}_d(P_{\mathrm{nse}}) = \beta_{\min}(N_1(P_{\mathrm{nse}}), N_1(P_{\mathrm{nse}}))\Psi(P_{\mathrm{nse}}). \tag{82}$$

Since $N_2(P_{\mathrm{nse}}) \geq N_1(P_{\mathrm{nse}})$, and since $\beta_{\min}$ is non-increasing in each of its arguments, we have

$$\beta_{\min}(N_1(P_{\mathrm{nse}}), N_2(P_{\mathrm{nse}})) \leq \beta_{\min}(N_1(P_{\mathrm{nse}}), N_1(P_{\mathrm{nse}})). \tag{83}$$

Therefore,

$$F_d(P_{\mathrm{nse}}) \leq \widetilde{F}_d(P_{\mathrm{nse}}) \qquad \forall P_{\mathrm{nse}} \in \Delta_d. \tag{84}$$

Thus,

$$V_d := \inf_{P_{\mathrm{nse}} \in \Delta_d} F_d(P_{\mathrm{nse}}) \leq \inf_{P_{\mathrm{nse}} \in \Delta_d} \widetilde{F}_d(P_{\mathrm{nse}}) =: \widetilde{V}_d, \tag{85}$$

so the right-hand side of (10) is indeed an upper bound on the original objective.

Next, we bound the gap between the two objectives. For any $P_{\mathrm{nse}} \in \Delta_d$,

$$0 \leq \widetilde{F}_d(P_{\mathrm{nse}}) - F_d(P_{\mathrm{nse}}) = [\beta_{\min}(N_1(P_{\mathrm{nse}}), N_1(P_{\mathrm{nse}})) - \beta_{\min}(N_1(P_{\mathrm{nse}}), N_2(P_{\mathrm{nse}}))]\,\Psi(P_{\mathrm{nse}}). \tag{86}$$

Since total variation distance is at most 1, we have

$$0 \leq \Psi(P_{\mathrm{nse}}) \leq 1. \tag{87}$$

Hence,

$$0 \leq \widetilde{F}_d(P_{\mathrm{nse}}) - F_d(P_{\mathrm{nse}}) \leq \beta_{\min}(N_1(P_{\mathrm{nse}}), N_1(P_{\mathrm{nse}})) - \beta_{\min}(N_1(P_{\mathrm{nse}}), N_2(P_{\mathrm{nse}})). \tag{88}$$

We now use the simplex constraint. Since $P_{\text{nse}} \in \Delta_d$, its coordinates sum to one. Because $N_1$ and $N_2$ are the smallest and second-smallest coordinates, all coordinates except the smallest one are at least $N_2$. Therefore, for any $P_{nse} \in \Delta_d$,

$$1 = \sum_{i=1}^{d} P_{\text{nse}}(i) \geq N_1 + (d-1)N_2. \tag{89}$$

Thus,

$$N_2 \leq \frac{1 - N_1}{d - 1}. \tag{90}$$

Since $N_2 \geq N_1$, we obtain

$$0 \leq N_2 - N_1 \leq \frac{1 - N_1}{d - 1} - N_1 = \frac{1 - dN_1}{d - 1} \leq \frac{1}{d - 1}. \tag{91}$$

Therefore,

$$0 \leq N_2 - N_1 \leq \frac{1}{d - 1} \tag{92}$$

Assume that $\beta_{\min}(z_1, z_2)$ is uniformly continuous in its second argument over the feasible region

$$\mathcal{A}_d = \{(z_1, z_2) : 0 \leq z_1 \leq z_2, \ z_1 + (d-1)z_2 \leq 1\}. \tag{93}$$

Then, since

$$|N_2(P_{\text{nse}}) - N_1(P_{\text{nse}})| \leq \frac{1}{d - 1}, \qquad \forall P_{nse} \in \Delta_d \tag{94}$$

we have

$$\sup_{P_{\text{nse}} \in \Delta_d} |\beta_{\min}(N_1(P_{\text{nse}}), N_2(P_{\text{nse}})) - \beta_{\min}(N_1(P_{\text{nse}}), N_1(P_{\text{nse}}))| \to 0 \tag{95}$$

as $d \to \infty$. Consequently,

$$\sup_{P_{\text{nse}} \in \Delta_d} \left| \widetilde{F}_d(P_{\text{nse}}) - F_d(P_{\text{nse}}) \right| \to 0. \tag{96}$$

It remains to show convergence of the optimal values. Since $F_d(P_{\text{nse}}) \leq \widetilde{F}_d(P_{\text{nse}})$ pointwise, we have

$$V_d \leq \widetilde{V}_d. \tag{97}$$

Let $P_d^\star \in \arg\min_{P_{\text{nse}} \in \Delta_d} F_d(P_{\text{nse}})$. Then

$$\widetilde{V}_d = \inf_{P_{\text{nse}} \in \Delta_d} \widetilde{F}_d(P_{\text{nse}}) \leq \widetilde{F}_d(P_d^\star). \tag{98}$$

Therefore,

$$\widetilde{V}_d \leq F_d(P_d^\star) + \sup_{P_{\text{nse}} \in \Delta_d} \left| \widetilde{F}_d(P_{\text{nse}}) - F_d(P_{\text{nse}}) \right|. \tag{99}$$

Since $F_d(P_d^\star) = V_d$, we get

$$0 \leq \widetilde{V}_d - V_d \leq \sup_{P_{\text{nse}} \in \Delta_d} \left| \widetilde{F}_d(P_{\text{nse}}) - F_d(P_{\text{nse}}) \right|. \tag{100}$$

Taking $d \to \infty$, the right-hand side converges to zero. Hence,

$$\widetilde{V}_d - V_d \to 0. \tag{101}$$

Thus, the upper-bound problem in (10) has the same asymptotic optimal value as the original problem. Moreover, if

$$\widetilde{P}_d^\star \in \arg \min_{P_{\text{nse}} \in \Delta_d} \widetilde{F}_d(P_{\text{nse}}), \tag{102}$$

then

$$F_d(\widetilde{P}_d^\star) \le \widetilde{F}_d(\widetilde{P}_d^\star) = \widetilde{V}_d. \tag{103}$$

Therefore,

$$0 \le F_d(\widetilde{P}_d^\star) - V_d \le \widetilde{V}_d - V_d \to 0. \tag{104}$$

Hence, any optimizer of the upper-bound problem is asymptotically optimal for the original problem.

Now, the only remaining component of the proof is the continuity of $\beta_{min}(N_{min_1}, N_{min_2})$ in $N_{min_2}$.

**Continuity of $\beta_{\min}(N_{\min_1}, N_{\min_2})$ in $N_{\min_2}$.**

Fix $\varepsilon > 0$, $\delta \in (0, 1)$, the private dataset size $n$, the number of released samples $m$, and $N_{\min_1}$. For notational simplicity, write

$$z_1 := N_{\min_1}, \qquad z_2 := N_{\min_2}. \tag{105}$$

Recall from Theorem 4.1 that $\beta_{\min}(z_1, z_2)$ is defined as

$$\beta_{\min}(z_1, z_2) = \inf \left\{ \beta \in [0, 1] : G(\beta, z_1, z_2) = \delta \right\}, \tag{106}$$

where

$$G(\beta, z_1, z_2) := \max \left\{ h_1(\beta, \varepsilon, n, m, z_1, z_2), h_2(\beta, \varepsilon, n, m, z_1, z_2) \right\}. \tag{107}$$

Here $h_1$ and $h_2$ are the two worst-case privacy expressions from Theorem 4.1.

We first observe that $G$ is continuous in $(\beta, z_1, z_2)$. Indeed, $h_1$ is a finite sum of terms of the form

$$\frac{m!}{\lambda! \mu! (m - \lambda - \mu)!} \left( 1 - \beta(z_1 + z_2) - \frac{1 - \beta}{n} \right)^{m - \lambda - \mu} \left[ \left( \beta z_1 + \frac{1 - \beta}{n} \right)^\lambda (\beta z_2)^\mu - e^\varepsilon \left( \beta z_2 + \frac{1 - \beta}{n} \right)^\mu (\beta z_1)^\lambda \right]_+ \tag{108}$$

Each factor inside the summation is continuous in $(\beta, z_1, z_2)$, and the map $x \mapsto [x]_+ = \max\{x, 0\}$ is continuous. Therefore $h_1$ is continuous. The same argument applies to $h_2$, and since the maximum of two continuous functions is continuous, $G$ is continuous.

Now fix $z_1$, and consider $z_2 \mapsto \beta_{\min}(z_1, z_2)$. Let

$$z_2^{(k)} \to z_2. \tag{109}$$

We show that

$$\beta_{\min}(z_1, z_2^{(k)}) \to \beta_{\min}(z_1, z_2). \tag{110}$$

Let

$$\beta^\star := \beta_{\min}(z_1, z_2). \tag{111}$$

Then, we have,

$$G(\beta^\star, z_1, z_2) = \delta, \tag{112}$$

***Claim* D.2.** For every $\eta > 0$,

$$G(\beta^\star - \eta, z_1, z_2) > \delta, \qquad G(\beta^\star + \eta, z_1, z_2) < \delta. \tag{113}$$

Equivalently, the privacy curve crosses the level $\delta$ at $\beta^\star$, rather than remaining flat at $\delta$.

The proof of Claim D.2 is given at the end. Using Claim D.2, we prove upper and lower semicontinuity.

First, fix any $\eta > 0$. By Claim D.2,

$$G(\beta^\star + \eta, z_1, z_2) < \delta. \tag{114}$$

Since $G$ is continuous in $z_2$, for all sufficiently large $k$,

$$G(\beta^\star + \eta, z_1, z_2^{(k)}) < \delta. \tag{115}$$

Therefore,

$$\beta_{\min}(z_1, z_2^{(k)}) \leq \beta^\star + \eta \tag{116}$$

for all sufficiently large $k$. Taking the limsup gives

$$\limsup_{k \to \infty} \beta_{\min}(z_1, z_2^{(k)}) \leq \beta^\star + \eta. \tag{117}$$

Since $\eta > 0$ was arbitrary,

$$\limsup_{k \to \infty} \beta_{\min}(z_1, z_2^{(k)}) \leq \beta^\star. \tag{118}$$

Next, again fix any $\eta > 0$. By Claim D.2,

$$G(\beta^\star - \eta, z_1, z_2) > \delta. \tag{119}$$

By continuity of $G$ in $z_2$, for all sufficiently large $k$,

$$G(\beta^\star - \eta, z_1, z_2^{(k)}) > \delta. \tag{120}$$

Thus $\beta^\star - \eta$ is not feasible for the problem defining $\beta_{\min}(z_1, z_2^{(k)})$. Since $G$ is non-increasing in $\beta$, no value

$$\beta \leq \beta^\star - \eta$$

is feasible either. Therefore,

$$\beta_{\min}(z_1, z_2^{(k)}) \geq \beta^\star - \eta$$

for all sufficiently large $k$. Taking the liminf gives

$$\liminf_{k \to \infty} \beta_{\min}(z_1, z_2^{(k)}) \geq \beta^\star - \eta.$$

Since $\eta > 0$ was arbitrary,

$$\liminf_{k \to \infty} \beta_{\min}(z_1, z_2^{(k)}) \geq \beta^\star.$$

Combining the two inequalities,

$$\limsup_{k \to \infty} \beta_{\min}(z_1, z_2^{(k)}) \leq \beta^\star \leq \liminf_{k \to \infty} \beta_{\min}(z_1, z_2^{(k)}).$$

Hence,

$$\beta_{\min}(z_1, z_2^{(k)}) \to \beta^\star = \beta_{\min}(z_1, z_2).$$

Therefore, $\beta_{\min}(N_{\min_1}, N_{\min_2})$ is continuous in $N_{\min_2}$. $\qquad\square$

**Proof of Claim D.2**

*Proof.* The first inequality in (113) is straightforward from the definition of $\beta^*$. For the second inequality, we show that for every $\eta > 0$ such that $\beta^* + \eta \leq 1$,

$$G(\beta^* + \eta, z_1, z_2) < \delta.$$

Let $\beta_0 := \beta^*$ and $\beta_1 := \beta^* + \eta$. Since $\eta > 0$, we have $\beta_1 > \beta_0$. Define $\alpha := \frac{1-\beta_1}{1-\beta_0}$. Then $0 \leq \alpha < 1$. Now consider any neighboring pair of private histograms $\bar{P}_{\text{prv}}$ and $\bar{P}'_{\text{prv}}$. For a fixed noise distribution $P_{\text{nse}}$, define

$$P_{\text{syn},\beta} = (1 - \beta)\bar{P}_{\text{prv}} + \beta P_{\text{nse}},$$

and

$$P'_{\text{syn},\beta} = (1 - \beta)\bar{P}'_{\text{prv}} + \beta P_{\text{nse}}.$$

Then

$$P_{\text{syn},\beta_1} = \alpha P_{\text{syn},\beta_0} + (1 - \alpha)P_{\text{nse}},$$

and similarly

$$P'_{\text{syn},\beta_1} = \alpha P'_{\text{syn},\beta_0} + (1 - \alpha)P_{\text{nse}}.$$

For compactness, write

$$P_0 := P_{\text{syn},\beta_0}, \qquad Q_0 := P'_{\text{syn},\beta_0}, \qquad R := P_{\text{nse}}.$$

Then

$$P_1 := P_{\text{syn},\beta_1} = \alpha P_0 + (1 - \alpha)R,$$

and

$$Q_1 := P'_{\text{syn},\beta_1} = \alpha Q_0 + (1 - \alpha)R.$$

We now compare the $m$-fold product distributions. Since

$$P_1^{\otimes m} = (\alpha P_0 + (1 - \alpha)R)^{\otimes m},$$

we can expand[7]

$$P_1^{\otimes m} = \sum_{S \subseteq [m]} \alpha^{|S|}(1 - \alpha)^{m-|S|} P_0^{\otimes S} \otimes R^{\otimes S^c}.$$

Similarly,

$$Q_1^{\otimes m} = \sum_{S \subseteq [m]} \alpha^{|S|}(1 - \alpha)^{m-|S|} Q_0^{\otimes S} \otimes R^{\otimes S^c}.$$

Using convexity of the hockey-stick divergence in the pair of distributions, we obtain

$$E_\gamma(P_1^{\otimes m} \| Q_1^{\otimes m}) \leq \sum_{S \subseteq [m]} \alpha^{|S|}(1 - \alpha)^{m-|S|} E_\gamma \left( P_0^{\otimes S} \otimes R^{\otimes S^c} \middle\| Q_0^{\otimes S} \otimes R^{\otimes S^c} \right).$$

Since the same distribution $R^{\otimes S^c}$ appears under both hypotheses on the coordinates in $S^c$, those coordinates contain no information for distinguishing the two distributions. By data processing, or equivalently by marginalizing out the coordinates in $S^c$,

$$E_\gamma \left( P_0^{\otimes S} \otimes R^{\otimes S^c} \middle\| Q_0^{\otimes S} \otimes R^{\otimes S^c} \right) = E_\gamma \left( P_0^{\otimes |S|} \middle\| Q_0^{\otimes |S|} \right).$$

---

[7]Sampling one coordinate from $P_1$ is equivalent to first drawing a Bernoulli variable with success probability $\alpha$. With probability $\alpha$, the coordinate is sampled from $P_0$, and with probability $1 - \alpha$, it is sampled from $R$. For $m$ independent samples, let $S \subseteq [m]$ denote the set of coordinates sampled from $P_0$. Then the coordinates in $S^c$ are sampled from $R$. The probability of this choice of $S$ is $\alpha^{|S|}(1 - \alpha)^{m-|S|}$, and the corresponding product distribution is $P_0^{\otimes S} \otimes R^{\otimes S^c}$. Summing over all subsets $S \subseteq [m]$ gives the stated expansion.

Therefore,

$$E_\gamma(P_1^{\otimes m} \| Q_1^{\otimes m}) \leq \sum_{S \subseteq [m]} \alpha^{|S|}(1-\alpha)^{m-|S|} E_\gamma\left(P_0^{\otimes |S|} \middle\| Q_0^{\otimes |S|}\right).$$

Grouping subsets by cardinality $s = |S|$, this becomes

$$E_\gamma(P_1^{\otimes m} \| Q_1^{\otimes m}) \leq \sum_{s=0}^{m} \binom{m}{s} \alpha^s (1-\alpha)^{m-s} E_\gamma\left(P_0^{\otimes s} \| Q_0^{\otimes s}\right).$$

For $s = 0$, the two distributions are identical, so

$$E_\gamma(P_0^{\otimes 0} \| Q_0^{\otimes 0}) = 0.$$

For every $1 \leq s \leq m$, marginalizing an $m$-sample observation down to its first $s$ coordinates is a stochastic channel. Hence, by data processing,

$$E_\gamma(P_0^{\otimes s} \| Q_0^{\otimes s}) \leq E_\gamma(P_0^{\otimes m} \| Q_0^{\otimes m}).$$

Thus,

$$E_\gamma(P_1^{\otimes m} \| Q_1^{\otimes m}) \leq \sum_{s=1}^{m} \binom{m}{s} \alpha^s (1-\alpha)^{m-s} E_\gamma(P_0^{\otimes m} \| Q_0^{\otimes m}).$$

Since

$$\sum_{s=1}^{m} \binom{m}{s} \alpha^s (1-\alpha)^{m-s} = 1 - (1-\alpha)^m,$$

we get

$$E_\gamma(P_1^{\otimes m} \| Q_1^{\otimes m}) \leq (1 - (1-\alpha)^m) \, E_\gamma(P_0^{\otimes m} \| Q_0^{\otimes m}).$$

Now $\gamma = e^\varepsilon$. Therefore,

$$E_{e^\varepsilon}\left(P_{\text{syn},\beta_1}^{\otimes m} \middle\| P_{\text{syn},\beta_1}'^{\otimes m}\right) \leq (1 - (1-\alpha)^m) \, E_{e^\varepsilon}\left(P_{\text{syn},\beta_0}^{\otimes m} \middle\| P_{\text{syn},\beta_0}'^{\otimes m}\right).$$

This inequality holds for every neighboring pair $(D_{\text{prv}}, D_{\text{prv}}')$. Taking the supremum over neighboring pairs gives

$$G(\beta_1, z_1, z_2) \leq (1 - (1-\alpha)^m) \, G(\beta_0, z_1, z_2).$$

Using $\beta_0 = \beta^\star$ and $G(\beta^\star, z_1, z_2) = \delta$ we obtain,

$$G(\beta^\star + \eta, z_1, z_2) \leq (1 - (1-\alpha)^m) \, \delta.$$

Since $0 \leq \alpha < 1$ we have $0 < 1 - (1-\alpha)^m < 1$ for $m \geq 1$ and $\alpha > 0$. Hence,

$$G(\beta^\star + \eta, z_1, z_2) < \delta.$$

If $\alpha = 0$, then $\beta_1 = 1$, and

$$P_{\text{syn},\beta_1} = P_{\text{syn},\beta_1}' = P_{\text{nse}}.$$

Therefore,

$$G(1, z_1, z_2) = 0 < \delta.$$

Thus, in all cases,

$$G(\beta^\star + \eta, z_1, z_2) < \delta$$

$\square$

## D.2. Complete Statement of Theorem 4.4

**Theorem 4.4 restated** Let $\gamma \in (0, 1]$ be a constant and let $\bar{P}_{pub} \in \Delta_d$ be a public distribution with the indices arranged as $\bar{P}_{pub}^{[1]} \le \bar{P}_{pub}^{[2]} \le \dots \le \bar{P}_{pub}^{[d]}$. Fix the privacy parameters $\varepsilon > 0$, $\delta \in (0, 1)$ and the required number of synthetic samples $m \ge 1$. Then,

$$\min_{\substack{P_{nse}\in\Delta_d}} \min_{\substack{\beta\in[0,1]\\ \beta\ge\beta_{min}(N_{min_1},N_{min_1})}} \max_{\bar{P}_{prv}\in\mathcal{B}_{\gamma,\bar{P}_{pub}}} \mathrm{TV}(\bar{P}_{prv}, P_{syn}) = \min\{\beta_{min}(t_1,t_1)\Phi_1(t_1), \beta_{min}(t_2,t_2)\Phi_2(t_2)\} \quad (121)$$

and the corresponding optimum $P_{nse}^*$ and $\beta^*$ are given by,

$$P_{nse}^* = \begin{cases} \mathrm{MIFR}(\bar{P}_{pub}, t_1), & \text{if } \beta_{min}(t_1,t_1)\Phi_1(t_1) \le \beta_{min}(t_2,t_2)\Phi_2(t_2) \\ \mathrm{WF}(\bar{P}_{pub}, t_2), & \text{if } \beta_{min}(t_1,t_1)\Phi_1(t_1) > \beta_{min}(t_2,t_2)\Phi_2(t_2) \end{cases} \quad (122)$$

$$\beta^* = \beta_{min}(N_{min}^*, N_{min}^*) \quad (123)$$

where $N_{min}^*$ is the minimum probability mass of $P_{nse}^*$. The parameters $t_1$ and $t_2$ are given by,

$$t_1 = \arg\min_{t\in T_1} \beta_{min}(t, t)\Phi_1(t) \quad (124)$$

$$t_2 = \arg\min_{t\in T_2} \beta_{min}(t, t)\Phi_2(t) \quad (125)$$

$$T_1 = \{t \in [0, 1/d] : \sum_{i:\bar{P}_{pub}^{[i]}\le P_{\mathrm{MIFR}(\bar{P}_{pub},t)}^{[i]}} \bar{P}_{pub}^{[i]} \ge \gamma\} \quad (126)$$

$$T_2 = \{t \in [0, 1/d] : \sum_{i:\bar{P}_{pub}^{[i]}\le P_{\mathrm{WF}(\bar{P}_{pub},t)}^{[i]}} \bar{P}_{pub}^{[i]} \le \gamma\} \quad (127)$$

with $P_{\mathrm{MIFR}(\bar{P}_{pub},t)}^{[i]}$, $P_{\mathrm{WF}(\bar{P}_{pub},t)}^{[i]}$, and $\bar{P}_{pub}^{[i]}$ denoting the $i$th element of $\mathrm{MIFR}(\bar{P}_{pub}, t)$, $\mathrm{WF}(\bar{P}_{pub}, t)$, and $\bar{P}_{pub}$, respectively.

$$\Phi_1(t) = \left(\gamma + \sum_{i:\bar{P}_{pub}^{[i]}\le t} (t - \bar{P}_{pub}^{[i]})\right) \quad (128)$$

$$\Phi_2(t) = tk_t + \max_{1\le u\le d-1-k_t} \min\left\{\left[\gamma - \sum_{i=1}^{k_t+u} \bar{P}_{pub}^{[i]} + \sum_{i=k_t+1}^{k_t+u} P_{\mathrm{WF}(\bar{P}_{pub},t)}^{[i]}\right]_+, \sum_{i=k_t+1}^{k_t+u} P_{\mathrm{WF}(\bar{P}_{pub},t)}^{[i]}\right\} \quad (129)$$

where $k_t = \sum_{i=1}^d \mathbf{1}_{\{\bar{P}_{pub}^{[i]}\le t\}}$ with $\mathbf{1}_{\{\cdot\}}$ denoting the identity function.

## D.3. Interpretation of Theorem 4.4:

To analyze the minmax TV distance in (3), we split it into two cases.

$$\underbrace{\min_{P_{nse}\in\Delta_d} \min_{\substack{\beta\in[0,1]\\ \beta\ge\beta_{min}(N_{min_1},N_{min_1})}} \max_{\bar{P}_{prv}\in\mathcal{B}_{\gamma,\bar{P}_{pub}}} \mathrm{TV}(\bar{P}_{prv}, P_{syn})}_{f(P_{nse})} = \min\left\{\min_{P_{nse}\in\mathcal{S}} f(P_{nse}), \min_{P_{nse}\in\Delta_d\setminus\mathcal{S}} f(P_{nse})\right\} \quad (130)$$

where $\mathcal{S} = \{P \in \Delta_d : \sum_{i:\bar{P}_{pub}^{[i]}\le P^{[i]}} \bar{P}_{pub}^{[i]} \ge \gamma\}$, with $P^{[i]}$ denoting the $i$th element of $P$ in the same order as $\bar{P}_{pub}$. To interpret the two regimes for $P_{nse}$ in (130), consider the simplified version of the LHS of (130) in (5). For a fixed $P_{nse}$, the inner max chooses the worst-case $\bar{P}_{prv} \in \mathcal{B}_{\gamma,\bar{P}_{pub}}$ (i.e., farthest from $P_{nse}$). If $P_{nse} \in \mathcal{S}$, this worst-case $\bar{P}_{prv}$ lies on the same line through $\bar{P}_{pub}$ and $P_{nse}$, giving $\mathrm{TV}(P_{nse}, \bar{P}_{pub}) + \gamma$ for the inner max (Fig. 6: blue). If $P_{nse} \in \Delta_d \setminus \mathcal{S}$, the worst-case $\bar{P}_{prv}$ is off that line, so the inner maximum is strictly smaller than $\mathrm{TV}(P_{nse}, \bar{P}_{pub}) + \gamma$ (Fig. 6: red). These two cases require different analyses, so we treat them separately and obtain the overall minimizer $P_{nse}^*$. This results in the two-case structure in (12)–(13) and the corresponding pairs of variables in Theorem 4.4.

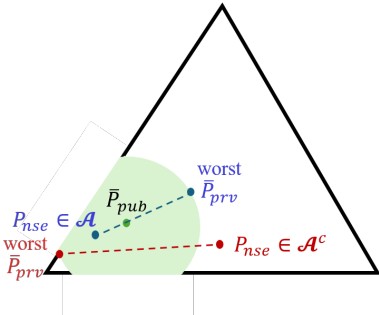

*Figure 6.* The two cases in Theorem 4.4 corresponds to two sets of potential noise distributions: 1) gives a worst-case $\bar{P}_{prv}$ in (3) that is on the same line as $P_{nse}$ and $\bar{P}_{pub}$, 2) Not on same line.

The **Key insight from Theorem 4.4** is that the optimal noise distribution $P^*_{nse}$ is always a floor-raised version of $\bar{P}_{pub}$, where both $\text{MIFR}(\bar{P}_{pub}, t_1)$ and $\text{WF}(\bar{P}_{pub}, t_2)$ raise small elements to their respective thresholds and redistribute probability mass from larger elements. This design balances two objectives: since $\bar{P}_{prv}$ and $\bar{P}_{pub}$ come from the same domain with typically small $\text{TV}(\bar{P}_{pub}, \bar{P}_{prv})$, choosing $P_{nse}$ close to $\bar{P}_{pub}$ is beneficial. However, many elements in $\bar{P}_{pub}$ can be small (even zero), which would add minimal noise to corresponding $\bar{P}^{[i]}_{prv}$ elements and require a larger mixing weight $\beta$ to maintain privacy, pushing $P_{syn}$ too close to $\bar{P}_{pub}$ (not to $\bar{P}_{prv}$). The optimal $P^*_{nse}$ resolves this tradeoff by remaining close to $\bar{P}_{pub}$ while floor-raising smaller probabilities to guarantee a sufficient $p = \min_i P^{[i]}_{nse}$, which allows a smaller mixing weight $\beta$ and yields a $P_{syn}$ that is closer to $\bar{P}_{prv}$.

# E. Proof of Theorem 4.4

**Theorem 4.4 restated:** Let $\gamma \in (0, 1]$ be a constant and let $\bar{P}_{pub} \in \Delta_d$ be a public distribution with the indices arranged as $\bar{P}^{[1]}_{pub} \leq \bar{P}^{[2]}_{pub} \leq \ldots \leq \bar{P}^{[d]}_{pub}$. Fix the privacy parameters $\varepsilon > 0, \delta \in (0, 1)$ and the required number of synthetic samples $m \geq 1$. Then,

$$\min_{\substack{P_{nse} \in \Delta_d}} \min_{\substack{\beta \in [0,1] \\ \beta \geq \beta_{min}(N_{min_1}, N_{min_1})}} \max_{\bar{P}_{prv} \in \mathcal{B}_{\gamma, \bar{P}_{pub}}} \text{TV}(\bar{P}_{prv}, P_{syn}) = \min\left\{\beta_{min}(t_1, t_1)\Phi_1(t_1), \beta_{min}(t_2, t_2)\Phi_2(t_2)\right\} \quad (131)$$

and the corresponding optimum $P^*_{nse}$ and $\beta^*$ are given by,

$$P^*_{nse} = \begin{cases} \text{MIFR}(\bar{P}_{pub}, t_1), & \text{if } \beta_{min}(t_1, t_1)\Phi_1(t_1) \leq \beta_{min}(t_2, t_2)\Phi_2(t_2) \\ \text{WF}(\bar{P}_{pub}, t_2), & \text{if } \beta_{min}(t_1, t_1)\Phi_1(t_1) > \beta_{min}(t_2, t_2)\Phi_2(t_2) \end{cases} \quad (132)$$

$$\beta^* = \beta_{min}(t^*, t^*) \quad (133)$$

where $t^* = \min_i P^{*[i]}_{nse}$ where $P^{*[i]}_{nse}$ denotes the $i$th element of $P^*_{nse}$. The parameters $t_1$ and $t_2$ are given by,

$$t_1 = \arg\min_{t \in T_1} \beta_{min}(t, t)\Phi_1(t) \quad (134)$$

$$t_2 = \arg\min_{t \in T_2} \beta_{min}(t, t)\Phi_2(t) \quad (135)$$

$$T_1 = \left\{t \in [0, 1/d] : \sum_{i:\bar{P}^{[i]}_{pub} \leq P^{[i]}_{\text{MIFR}(\bar{P}_{pub}, t)}} \bar{P}^{[i]}_{pub} \geq \gamma\right\} \quad (136)$$

$$T_2 = \left\{t \in [0, 1/d] : \sum_{i:\bar{P}^{[i]}_{pub} \leq P^{[i]}_{\text{WF}(\bar{P}_{pub}, t)}} \bar{P}^{[i]}_{pub} \leq \gamma\right\} \quad (137)$$

with $P^{[i]}_{\mathrm{MIFR}(\bar{P}_{pub},t)}$, $P^{[i]}_{\mathrm{WF}(\bar{P}_{pub},t)}$, and $\bar{P}^{[i]}_{pub}$ denoting the $i$th element of $\mathrm{MIFR}(\bar{P}_{pub},t)$, $\mathrm{WF}(\bar{P}_{pub},t)$, and $\bar{P}_{pub}$, respectively.

$$\Phi_1(t) = \left( \gamma + \sum_{i:\bar{P}^{[i]}_{pub} \leq t} (t - \bar{P}^{[i]}_{pub}) \right) \tag{138}$$

$$\Phi_2(t) = tk(t) + \max_{1 \leq u \leq d-1-k(t)} \min \left\{ \left[ \gamma - \sum_{i=1}^{k(t)+u} \bar{P}^{[i]}_{pub} + \sum_{i=k(t)+1}^{k(t)+u} P^{[i]}_{\mathrm{WF}(\bar{P}_{pub},t)} \right]_+ , \sum_{i=k(t)+1}^{k(t)+u} P^{[i]}_{\mathrm{WF}(\bar{P}_{pub},t)} \right\} \tag{139}$$

where $k(t) = \sum_{i=1}^{d} \mathbf{1}_{\{\bar{P}^{[i]}_{pub} \leq t\}}$ with $\mathbf{1}_{\{\cdot\}}$ denoting the identity function.

**Proof:** The proof of Theorem 4.4 is structured as follows. The optimization problem we solve is:

$$\min_{\substack{P_{nse} \in \Delta_d \\ \beta \geq \beta_{min}(N_{min_1}, N_{min_1})}} \min_{\beta \in [0,1]} \max_{\bar{P}_{prv} \in \mathcal{B}_{\gamma,\bar{P}_{pub}}} \mathrm{TV}(\bar{P}_{prv}, (1-\beta)\bar{P}_{prv} + \beta P_{nse})$$

$$= \min_{\substack{P_{nse} \in \Delta_d \\ \beta \geq \beta_{min}(N_{min_1}, N_{min_1})}} \min_{\beta \in [0,1]} \max_{\bar{P}_{prv} \in \mathcal{B}_{\gamma,\bar{P}_{pub}}} \beta\mathrm{TV}(\bar{P}_{prv}, P_{nse}) \tag{140}$$

- First, we fix $P_{nse}$ and $\beta$, and consider the inner-most optimization by characterizing the $\bar{P}^*_{prv}$ that maximizes the objective. In this stage, we identify two sets of $P_{nse}$. 1) $P_{nse} \in \mathcal{S}$ that results in $\bar{P}^*_{prv}$ that lie on the same straight line connecting $\bar{P}_{pub}$ and $P_{nse}$, with exactly $\gamma$-TV distance from $\bar{P}_{pub}$, i.e., $\mathrm{TV}(\bar{P}^*_{prv}, P_{nse}) = \gamma + \mathrm{TV}(P_{nse}, \bar{P}_{pub})$, and 2) $P_{nse} \notin \mathcal{S}$ that results in $\bar{P}^*_{prv}$ that does not lie on the same straight line connecting $\bar{P}_{pub}$ and $P_{nse}$. These two cases are analyzed separately.

- Case 1, where the optimization problem is solved considering only $P_{nse} \in \mathcal{S}$ results in $P^*_{nse} = \mathrm{MIFR}(\bar{P}_{pub}, t)$, where the optimal threshold $t$ is obtained by a 1D search at the end.

- In case 2, the maximizing $\bar{P}^*_{prv}$ lies at a TV distance strictly smaller than $\gamma + \mathrm{TV}(P_{nse}, \bar{P}_{pub})$ from $P_{nse}$. In this case, the main step is to characterize the exact $TV(\bar{P}^*_{prv}, P_{nse})$ (solution to inner-most optimization for given $P_{nse}$ and $\beta$). Then, we show that $P^*_{nse} = \mathrm{WF}(\bar{P}_{pub}, t)$ minimizes the objective (Lemma E.2).

- In both cases, the optimal $\beta$ is obtained by Theorem 4.1.

- Finally, for a given $\bar{P}_{pub}$ and $\gamma$, we compute the optimal parameters of case 1 and case 2, and select the $(P^*_{nse}, \beta^*)$ pair corresponding to the case that results in the overall minimum objective value.

First, we consider the inner most optimization considering any fixed (feasible) $P_{nse}$ and $\beta$. Any $\bar{P}_{prv} \in \mathcal{B}_{\gamma,\bar{P}_{pub}}$ can be written as $\bar{P}_{prv} = \bar{P}_{pub} + U$ where $U \in \mathcal{A}$ with:

$$\mathcal{A} = \{U : \|U\|_1 \leq 2\gamma, \quad \sum_{i=1}^{d} U_i = 0, \quad -\bar{P}^{[i]}_{pub} \leq U_i \leq 1 - \bar{P}^{[i]}_{pub}, \forall i\} \tag{141}$$

as $\mathrm{TV}(\bar{P}_{pub}, \bar{P}_{prv}) = \frac{1}{2}\sum_{i=1}^{d}|U_i| \leq \gamma$. Without loss of generality, for given distributions $\bar{P}_{pub}$ and $P_{nse}$, assume that $\bar{P}^{[1]}_{pub} - P^{[1]}_{nse} \leq \bar{P}^{[2]}_{pub} - P^{[2]}_{nse} \leq \ldots \bar{P}^{[k]}_{pub} - P^{[k]}_{nse} \leq 0 < \bar{P}^{[k+1]}_{pub} - P^{[k+1]}_{nse} \leq \ldots \leq \bar{P}^{[d]}_{pub} - P^{[d]}_{nse}$ for some $k < d$. Then,[8]

---

[8] $k = d$ is the special case where $P_{nse} = \bar{P}_{pub}$.

$$\max_{\bar{P}_{prv} \in \mathcal{B}_{\gamma, \bar{P}_{pub}}} \mathrm{TV}(\bar{P}_{prv}, (1-\beta)\bar{P}_{prv} + \beta P_{nse}) = \max_{\bar{P}_{prv} \in \mathcal{B}_{\gamma, \bar{P}_{pub}}} \frac{\beta}{2} \sum_{i=1}^{d} |\bar{P}_{prv}^{[i]} - P_{nse}^{[i]}| \tag{142}$$

$$= \max_{U \in \mathcal{A}} \quad \frac{\beta}{2} \sum_{i=1}^{d} |U_i + \bar{P}_{pub}^{[i]} - P_{nse}^{[i]}| \tag{143}$$

$$= \max_{U \in \mathcal{A}} \quad \frac{\beta}{2} \sum_{i=1}^{k} |U_i + \bar{P}_{pub}^{[i]} - P_{nse}^{[i]}| + \frac{\beta}{2} \sum_{i=k+1}^{d} |U_i + \bar{P}_{pub}^{[i]} - P_{nse}^{[i]}| \tag{144}$$

To maximize (144), we seek $U \in \mathcal{A}$ that pushes each term $|U_i + \bar{P}_{pub}^{[i]} - P_{nse}^{[i]}|$ as large as possible. Since $\bar{P}_{pub}^{[i]} - P_{nse}^{[i]} \leq 0$ for $i \in \{1, \ldots, k\}$, choosing $U_i \leq 0$ increases $|U_i + \bar{P}_{pub}^{[i]} - P_{nse}^{[i]}|$; similarly, since $\bar{P}_{pub}^{[i]} - P_{nse}^{[i]} > 0$ for $i \in \{k+1, \ldots, d\}$, choosing $U_i > 0$ increases $|U_i + \bar{P}_{pub}^{[i]} - P_{nse}^{[i]}|$. Subject to $\sum_{i=1}^{d} U_i = 0$, the maximum is achieved by saturating the $\|U\|_1 \leq 2\gamma$ constraint with equality. However, the per-coordinate constraints $-\bar{P}_{pub}^{[i]} \leq U_i \leq 1 - \bar{P}_{pub}^{[i]}$ may prevent this saturation from being simultaneously achievable with the sign conditions $U_i \leq 0$ for $i \in \{1, \ldots, k\}$ and $U_i > 0$ for $i \in \{k+1, \ldots, d\}$. We analyze these two cases:

1. $\gamma \leq \sum_{i=1}^{k} \bar{P}_{pub}^{[i]}$: This is the case where $\|U\|_1 \leq 2\gamma$ is achieved with equality while satisfying the sign conditions. This case ultimately leads to $P_{nse} \in \mathcal{S}$ in App. D.3, where $P_{nse}, \bar{P}_{pub}$ and worst-case $\bar{P}_{prv}$ are on the same straight line on the simplex.[9]

2. $\gamma > \sum_{i=1}^{k} \bar{P}_{pub}^{[i]}$: This is the case where it is impossible to achieve equality in $\|U\|_1 \leq 2\gamma$ while also satisfying the sign conditions. This leads to the $P_{nse} \in \Delta_d \setminus \mathcal{S}$ case in App. D.3.

All $P_{nse} \in \Delta_d$ can be divided into two categories: case 1 above consists of $P_{nse}$ distributions that satisfy:

$$\sum_{i: \bar{P}_{pub}^{[i]} \leq P_{nse}^{[i]}} \bar{P}_{pub}^{[i]} \geq \gamma \tag{145}$$

for the given $\bar{P}_{pub}$. We call this set of $P_{nse}$ distributions $\mathcal{S}$. Similarly, case 2 consists of $P_{nse}$ distributions that satisfy:

$$\sum_{i: \bar{P}_{pub}^{[i]} \leq P_{nse}^{[i]}} \bar{P}_{pub}^{[i]} < \gamma \tag{146}$$

We consider case 1 and case 2 separately, obtain the optimum $P_{nse}$ within sets $\mathcal{S}$ and $\Delta_d \setminus \mathcal{S}$ separately, and then select the best out of the two, to get the optimum $P_{nse}^*$. This is explained in App. D.3. Next, we begin the analysis of case 1.

**Case 1:** First, we establish the solution to (144) for $P_{nse} \in \mathcal{S}$.

**Lemma E.1.** *For any fixed $\bar{P}_{pub} \in \Delta_d$, $\gamma > 0$, and $P_{nse} \in \mathcal{S}$, the optimal solution to (144) is given by any $U$ satisfying the following properties:*

1. *$U_i \leq 0$ if $i \leq k$ and $U_i \geq 0$ if $i > k$*

2. *$\sum_{i=1}^{k} U_i = -\gamma$ and $\sum_{i=k+1}^{d} U_i = \gamma \implies \|U\|_1 = 2\gamma$*

*and the resulting maximum TV distance is given by,*

$$\max_{\bar{P}_{prv} \in \mathcal{B}_{\gamma, \bar{P}_{pub}}} \mathrm{TV}(\bar{P}_{prv}, (1-\beta)\bar{P}_{prv} + \beta P_{nse}) = \beta \left(\gamma + \mathrm{TV}(\bar{P}_{pub}, P_{nse})\right) \tag{147}$$

---

[9]To ensure that there exists a $U$ simultaneously satisfying $\sum_{i=1}^{d} U_i = 0$, $\|U\|_1 = 2\gamma$ and the sign conditions at the same time, we need both $\gamma \leq \sum_{i=1}^{k} \bar{P}_{pub}^{[i]}$ and $\gamma \leq \sum_{i=k+1}^{d} (1 - \bar{P}_{pub}^{[i]})$. Note that $\gamma \leq \sum_{i=1}^{k} \bar{P}_{pub}^{[i]}$ automatically implies and $\gamma \leq \sum_{i=k+1}^{d} (1 - \bar{P}_{pub}^{[i]})$ for any $k < d$.

**Proof:** Consider the optimization in (144):

$$\max_{\bar{P}_{prv} \in \mathcal{B}_{\gamma, \bar{P}_{pub}}} \text{TV}(\bar{P}_{prv}, (1-\beta)\bar{P}_{prv} + \beta P_{nse}) = \max_{U \in \mathcal{A}} \frac{\beta}{2} \sum_{i=1}^{d} |U_i + \bar{P}_{pub}^{[i]} - P_{nse}^{[i]}| \tag{148}$$

$$\leq \max_{U \in \mathcal{A}} \frac{\beta}{2} \left( \sum_{i=1}^{d} |U_i| + \sum_{i=1}^{d} |\bar{P}_{pub}^{[i]} - P_{nse}^{[i]}| \right) \tag{149}$$

$$\leq \beta \left( \gamma + \text{TV}(\bar{P}_{pub}, P_{nse}) \right) \tag{150}$$

Now consider any $U \in \mathcal{A}$ satisfying properties 1 and 2 above. Then, we have,

$$\text{TV}(\bar{P}_{prv}, (1-\beta)\bar{P}_{prv} + \beta P_{nse}) = \frac{\beta}{2} \sum_{i=1}^{d} |U_i + \bar{P}_{pub}^{[i]} - P_{nse}^{[i]}| \tag{151}$$

$$= \frac{\beta}{2} \sum_{i=1}^{k} |U_i + \bar{P}_{pub}^{[i]} - P_{nse}^{[i]}| + \frac{\beta}{2} \sum_{i=k+1}^{d} |U_i + \bar{P}_{pub}^{[i]} - P_{nse}^{[i]}| \tag{152}$$

$$= \frac{\beta}{2} \left( \gamma + \sum_{i=1}^{k} |\bar{P}_{pub}^{[i]} - P_{nse}^{[i]}| \right) + \frac{\beta}{2} \left( \gamma + \sum_{i=k+1}^{d} |\bar{P}_{pub}^{[i]} - P_{nse}^{[i]}| \right) \tag{153}$$

$$= \beta \left( \gamma + \text{TV}(\bar{P}_{pub}, P_{nse}) \right) \tag{154}$$

Note that the conditions in 1 and 2 are feasible (satisfies $-\bar{P}_{pub}^{[i]} \leq U_i \leq 1 - \bar{P}_{pub}^{[i]}, \forall i$) since $\gamma \leq \sum_{i=1}^{k} \bar{P}_{pub}^{[i]} \leq \sum_{k+1}^{d}(1 - \bar{P}_{pub}^{[i]})$ in case 1 for any $k < d$. It can be directly verified that when $k = d$, which only occurs when $P_{nse} = \bar{P}_{pub}$, also achieves (150). This completes the proof of Lemma E.1, since the upper bound in (150) is achieved with equality by the $U$ vectors satisfying the two properties stated in Lemma E.1. ∎

By applying Lemma E.1 to the complete optimization in the RHS of (140) (for case 1), we get,

$$\min_{\substack{P_{nse} \in \Delta_d \\ \sum_{i: \bar{P}_{pub}^{[i]} \leq P_{nse}^{[i]}} \bar{P}_{pub}^{[i]} \geq \gamma}} \min_{\substack{\beta \in [0,1] \\ \beta \geq \beta_{min}(N_{min_1}, N_{min_1})}} \max_{\bar{P}_{prv} \in \mathcal{B}_{\gamma, \bar{P}_{pub}}} \text{TV}(\bar{P}_{prv}, (1-\beta)\bar{P}_{prv} + \beta P_{nse})$$

$$= \min_{\substack{P_{nse} \in \Delta_d \\ \sum_{i: \bar{P}_{pub}^{[i]} \leq P_{nse}^{[i]}} \bar{P}_{pub}^{[i]} \geq \gamma}} \min_{\substack{\beta \in [0,1] \\ \beta \geq \beta_{min}(N_{min_1}, N_{min_1})}} \beta \left( \gamma + \text{TV}(\bar{P}_{pub}, P_{nse}) \right) \tag{155}$$

For any given $P_{nse}$, $\text{TV}(\bar{P}_{pub}, P_{nse})$ is fixed and (155) is increasing in $\beta$. Thus we select $\beta_{min}(N_{min_1}, N_{min_1})$ from Theorem 4.1 as the optimum $\beta$, which results in:

$$\min_{\substack{P_{nse} \in \Delta_d \\ \sum_{i: \bar{P}_{pub}^{[i]} \leq P_{nse}^{[i]}} \bar{P}_{pub}^{[i]} \geq \gamma}} \min_{\substack{\beta \in [0,1] \\ \beta \geq \beta_{min}(N_{min_1}, N_{min_1})}} \max_{\bar{P}_{prv} \in \mathcal{B}_{\gamma, \bar{P}_{pub}}} \text{TV}(\bar{P}_{prv}, (1-\beta)\bar{P}_{prv} + \beta P_{nse})$$

$$= \min_{\substack{P_{nse} \in \Delta_d \\ \sum_{i: \bar{P}_{pub}^{[i]} \leq P_{nse}^{[i]}} \bar{P}_{pub}^{[i]} \geq \gamma}} \beta_{min}(N_{min_1}, N_{min_1}) \left( \gamma + \text{TV}(\bar{P}_{pub}, P_{nse}) \right) \tag{156}$$

$$= \min_{0 \leq t \leq \frac{1}{d}} \min_{\substack{P_{nse} \in \Delta_d \\ N_{min_1} = t \\ \sum_{i: \bar{P}_{pub}^{[i]} \leq P_{nse}^{[i]}} \bar{P}_{pub}^{[i]} \geq \gamma}} \beta_{min}(t, t) \left( \gamma + \text{TV}(\bar{P}_{pub}, P_{nse}) \right) \tag{157}$$

Since $\beta_{min}(t, t)$ is fixed for a given $t$, (157) simplifies to:

$$\min_{\substack{P_{nse}\in\Delta_d \\ \sum_{i:\bar{P}_{pub}^{[i]}\leq P_{nse}^{[i]}}\bar{P}_{pub}^{[i]}\geq\gamma}} \quad \min_{\substack{\beta\in[0,1] \\ \beta\geq\beta_{min}(N_{min_1},N_{min_2})}} \quad \max_{\bar{P}_{prv}\in\mathcal{B}_{\gamma,\bar{P}_{pub}}} \quad \text{TV}(\bar{P}_{prv},(1-\beta)\bar{P}_{prv}+\beta P_{nse})$$

$$= \min_{0\leq t\leq\frac{1}{d}} \quad \beta_{min}(t,t) \quad \min_{\substack{P_{nse}\in\Delta_d \\ N_{min_1}=t \\ \sum_{i:\bar{P}_{pub}^{[i]}\leq P_{nse}^{[i]}}\bar{P}_{pub}^{[i]}\geq\gamma}} \quad \left(\gamma+\text{TV}(\bar{P}_{pub},P_{nse})\right) \tag{158}$$

$$= \min_{t\in T_1} \quad \beta_{min}(t,t)\left(\gamma+\sum_{i:\bar{P}_{pub}^{[i]}\leq t}(t-\bar{P}_{pub}^{[i]})\right) \tag{159}$$

The last equality is explained as follows. Since $N_{min_1}=t$ we need all the elements of $P_{nse}$ to be larger than or equal to $t$. This means:

$$\text{TV}(\bar{P}_{pub},P_{nse}) \geq \sum_{i:\bar{P}_{pub}^{[i]}\leq t}(t-\bar{P}_{pub}^{[i]}) \tag{160}$$

Note that one optimum $P_{nse}$ that achieves this lower bound on $\text{TV}(\bar{P}_{pub},P_{nse})$ with equality while satisfying $\sum_{i:\bar{P}_{pub}^{[i]}\leq P_{nse}^{[i]}}\bar{P}_{pub}^{[i]}\geq\gamma$ is $P_{nse}^*=\text{MIFR}(\bar{P}_{pub},t)$, as it maximizes $\sum_{i:\bar{P}_{pub}^{[i]}=P_{nse}^{[i]}}\bar{P}_{pub}^{[i]}$ (consequently, MIFR maximizes the LHS of $\sum_{i:\bar{P}_{pub}^{[i]}\leq P_{nse}^{[i]}}\bar{P}_{pub}^{[i]}\geq\gamma$) for any fixed $t$, with no increase in $\text{TV}(\bar{P}_{pub},P_{nse})$.[10] However, note that $\sum_{i:\bar{P}_{pub}^{[i]}\leq P_{nse}^{[i]}}\bar{P}_{pub}^{[i]}\geq\gamma$ could be infeasible for certain $(\bar{P}_{pub},\gamma,t)$ tuples for any choice of $P_{nse}$. For a given $\bar{P}_{pub}$ and $\gamma$, $T_1$ denotes the set of feasible values for $t$, i.e.,

$$T_1 = \left\{t\in\left[0,\frac{1}{d}\right]:\sum_{i:\bar{P}_{pub}^{[i]}\leq\text{MIFR}^{[i]}(\bar{P}_{pub},t)}\bar{P}_{pub}^{[i]}\geq\gamma\right\} \tag{161}$$

where $\text{MIFR}^{[i]}(\bar{P}_{pub},t)$ indicates the $i$th element of $\text{MIFR}(\bar{P}_{pub},t)$. This concludes the first case.

**Case 2:** Now, consider case 2 where $\gamma\geq\sum_{i=1}^k\bar{P}_{pub}^{[i]}$. Consider the inner optimization in (144) for any fixed (feasible) $P_{nse}$ and $\beta$.

$$\max_{\bar{P}_{prv}\in\mathcal{B}_{\gamma,\bar{P}_{pub}}}\text{TV}(\bar{P}_{prv},(1-\beta)\bar{P}_{prv}+\beta P_{nse}) = \max_{\bar{P}_{prv}\in\mathcal{B}_{\gamma,\bar{P}_{pub}}}\frac{\beta}{2}\sum_{i=1}^d|\bar{P}_{prv}^{[i]}-P_{nse}^{[i]}| \tag{162}$$

$$= \max_{U\in\mathcal{A}}\frac{\beta}{2}\sum_{i=1}^d|U_i+\bar{P}_{pub}^{[i]}-P_{nse}^{[i]}| \tag{163}$$

$$= \max_{U\in\mathcal{A}}\frac{\beta}{2}\sum_{i=1}^k|U_i+\bar{P}_{pub}^{[i]}-P_{nse}^{[i]}|+\frac{\beta}{2}\sum_{i=k+1}^d|U_i+\bar{P}_{pub}^{[i]}-P_{nse}^{[i]}| \tag{164}$$

This analysis is done in two steps:

1. The maximum gain from the first term is achieved when $U_i=-\bar{P}_{pub}^{[i]}$ for $i=1,\ldots,k$. Since $\sum_{i=1}^d U_i=0$, the mass $-\sum_{i=1}^k\bar{P}_{pub}^{[i]}$ must be included in $\{U_{k+1},\ldots,U_d\}$, which contributes to the gain from the second term. Note that

---

[10]Note that $\text{MIFR}(\bar{P}_{pub},t)$ obtains $P_{nse}$ by raising the minimum amount of probability mass from $\bar{P}_{pub}$ to satisfy $N_{min_1}=t$. Thus, the probability mass reduced from the larger $\bar{P}_{pub}^{[i]}$ probabilities is also minimal. This is why $\text{MIFR}(\bar{P}_{pub},t)$ maximizes $\sum_{i:\bar{P}_{pub}^{[i]}\leq P_{nse}^{[i]}}\bar{P}_{pub}^{[i]}$ among all possible $P_{nse}$ that satisfy $N_{min_1}=t$.

this is feasible since $\sum_{i=1}^{k} \bar{P}_{pub}^{[i]} \leq \sum_{i=k+1}^{d} (1 - \bar{P}_{pub}^{[i]})$ for any $k < d$. The total gain to $\mathrm{TV}(\bar{P}_{prv}, P_{nse})$ from this selection of $U_i$'s (in addition to $\mathrm{TV}(\bar{P}_{pub}, P_{nse})$) is $\sum_{i=1}^{k} \bar{P}_{pub}^{[i]}$.

2. Next, we need to find optimum $\{U_{k+1}, \ldots, U_d\}$, which already includes a mass of $\sum_{i=1}^{k} \bar{P}_{pub}^{[i]}$. Since $U_i$, $i \leq k$ are already saturated and $\sum_{i=1}^{d} U_i = 0$, any +mass (except for the $\sum_{i=1}^{k} \bar{P}_{pub}^{[i]}$ from step 1) for $U_i$ in $i = k+1, \ldots, d$ must be matched with the corresponding -mass among $U_i$ in $i = k+1, \ldots, d$. While $-(\bar{P}_{pub}^{[i]} - P_{nse}^{[i]}) < U_i < 0$ hurts the second term in (164), $U_i \leq -(\bar{P}_{pub}^{[i]} - P_{nse}^{[i]})$ starts gaining back. But whenever these *cross-overs* occur, the net gain to the TV expression (in addition to $\mathrm{TV}(\bar{P}_{pub}, P_{nse})$) is only $|U_i| - (\bar{P}_{pub}^{[i]} - P_{nse}^{[i]})$ (recall this was entirely $|U_i|$ in step 1). Thus, we need to make $U_i < 0$ for $i$ with smaller $\bar{P}_{pub}^{[i]} - P_{nse}^{[i]}$ in $i \in \{k+1, \ldots, d\}$. If we do the *crossover* by selecting $U_i \leq -(\bar{P}_{pub}^{[i]} - P_{nse}^{[i]})$ for $i = k+1, \ldots, k+u$, where $u < d - k$, the gain for the TV expression (in addition to $\mathrm{TV}(\bar{P}_{pub}, P_{nse})$ and $\sum_{i=1}^{k} \bar{P}_{pub}^{[i]}$) is given by,[11]

$$
\phi(u, P_{nse}) = \min \left\{ \left[ \gamma - \sum_{i=1}^{k} \bar{P}_{pub}^{[i]} - \sum_{i=k+1}^{k+u} (\bar{P}_{pub}^{[i]} - P_{nse}^{[i]}) \right]_{+}, \sum_{i=k+1}^{k+u} P_{nse}^{[i]}, \right.
$$
$$
\left. \left[ \sum_{i=k+u+1}^{d} (1 - \bar{P}_{pub}^{[i]}) - \sum_{i=1}^{k} \bar{P}_{pub}^{[i]} - \sum_{i=k+1}^{k+u} (\bar{P}_{pub}^{[i]} - P_{nse}^{[i]}) \right]_{+} \right\} \tag{165}
$$

$$
= \min \left\{ \left[ \gamma - \sum_{i=1}^{k} \bar{P}_{pub}^{[i]} - \sum_{i=k+1}^{k+u} (\bar{P}_{pub}^{[i]} - P_{nse}^{[i]}) \right]_{+}, \sum_{i=k+1}^{k+u} P_{nse}^{[i]} \right\} \tag{166}
$$

$$
= \min \left\{ \left[ \gamma - \sum_{i=1}^{k+u} \bar{P}_{pub}^{[i]} + \sum_{i=k+1}^{k+u} P_{nse}^{[i]} \right]_{+}, \sum_{i=k+1}^{k+u} P_{nse}^{[i]} \right\} \tag{167}
$$

where the first, second, and third terms inside the $\min$ of (165) come from the $\|U\|_1 \leq 2\gamma$, $U_i \geq -\bar{P}_{pub}^{[i]}$, and $U_i \leq 1 - \bar{P}_{pub}^{[i]}$ constraints, and (166) is a result of $\sum_{i=k+u+1}^{d} (1 - \bar{P}_{pub}^{[i]}) - \sum_{i=1}^{k} \bar{P}_{pub}^{[i]} - \sum_{i=k+1}^{k+u} (\bar{P}_{pub}^{[i]} - P_{nse}^{[i]}) \geq \sum_{i=k+1}^{k+u} P_{nse}^{[i]} \geq 0$. Therefore, the maximum gain is achieved when:

$$
\max_{1 \leq u \leq d-1-k} \phi(u, P_{nse}) = \max_{1 \leq u \leq d-1-k} \min \left\{ \left[ \gamma - \sum_{i=1}^{k+u} \bar{P}_{pub}^{[i]} + \sum_{i=k+1}^{k+u} P_{nse}^{[i]} \right]_{+}, \sum_{i=k+1}^{k+u} P_{nse}^{[i]} \right\} \tag{168}
$$

Note that if $k = d - 1$, *crossover* is not possible as $U$ is fixed at $U_i = -\bar{P}_{pub}^{[i]}$ for $i < d$ and $U_d = \sum_{i=1}^{d-1} \bar{P}_{pub}^{[i]}$. Also, $k = d$ is the special case where $P_{nse} = \bar{P}_{pub}$, where the parameters to be optimized ($\beta^*$ and $P_{nse}^*$) are already determined.

---

[11]Note that $u = d - k$ is infeasible for crossovers as there has to be at least one $U_i$ element that is positive to compensate for all the negative mass and satisfy $\sum_{i=1}^{d} U_i = 0$.

With the above two sources of gains, for any $k < d - 1$, we have,

$$\min_{\substack{P_{nse} \in \Delta_d \\ \sum_{i:\bar{P}_{pub}^{[i]} \leq P_{nse}^{[i]}} \bar{P}_{pub}^{[i]} \leq \gamma}} \min_{\substack{\beta \in [0,1] \\ \beta \geq \beta_{min}(N_{min_1}, N_{min_1})}} \max_{\bar{P}_{prv} \in \mathcal{B}_{\gamma, \bar{P}_{pub}}} \mathrm{TV}(\bar{P}_{prv}, (1-\beta)\bar{P}_{prv} + \beta P_{nse})$$

$$= \min_{\substack{P_{nse} \in \Delta_d \\ \sum_{i:\bar{P}_{pub}^{[i]} \leq P_{nse}^{[i]}} \bar{P}_{pub}^{[i]} \leq \gamma}} \min_{\substack{\beta \in [0,1] \\ \beta \geq \beta_{min}(N_{min_1}, N_{min_1})}} \beta \left( \mathrm{TV}(\bar{P}_{pub}, P_{nse}) + \sum_{i=1}^{k} \bar{P}_{pub}^{[i]} + \max_{1 \leq u \leq d-1-k} \phi(u, P_{nse}) \right) \quad (169)$$

$$= \min_{0 \leq t \leq \frac{1}{d}} \min_{\substack{P_{nse} \in \Delta_d \\ N_{min_1} = t \\ \sum_{i:\bar{P}_{pub}^{[i]} \leq P_{nse}^{[i]}} \bar{P}_{pub}^{[i]} \leq \gamma}} \beta_{min}(t, t) \left( \mathrm{TV}(\bar{P}_{pub}, P_{nse}) + \sum_{i=1}^{k} \bar{P}_{pub}^{[i]} + \max_{1 \leq u \leq d-1-k} \phi(u, P_{nse}) \right) \quad (170)$$

$$= \min_{0 \leq t \leq \frac{1}{d}} \beta_{min}(t, t) \min_{\substack{P_{nse} \in \Delta_d \\ N_{min_1} = t \\ \sum_{i:\bar{P}_{pub}^{[i]} \leq P_{nse}^{[i]}} \bar{P}_{pub}^{[i]} \leq \gamma}} \left( \mathrm{TV}(\bar{P}_{pub}, P_{nse}) + \sum_{i=1}^{k} \bar{P}_{pub}^{[i]} + \max_{1 \leq u \leq d-1-k} \phi(u, P_{nse}) \right) \quad (171)$$

**Lemma E.2.** *For any given $\bar{P}_{pub} \in \Delta_d$, $\gamma \in (0, 1]$, and $t \geq 0$,*

$$\arg \min_{\substack{P_{nse} \in \Delta_d \\ N_{min_1} = t \\ \sum_{i:\bar{P}_{pub}^{[i]} \leq P_{nse}^{[i]}} \bar{P}_{pub}^{[i]} \leq \gamma}} \left( \mathrm{TV}(\bar{P}_{pub}, P_{nse}) + \sum_{i=1}^{k} \bar{P}_{pub}^{[i]} + \max_{1 \leq u \leq d-1-k} \phi(u, P_{nse}) \right) = \begin{cases} \mathrm{WF}(\bar{P}_{pub}, t), & 0 \leq t \leq t_{max} \\ \text{does not exist}, & t > t_{max} \end{cases}$$

$$(172)$$

*where $t_{max} = \arg \max\{t \in \left[0, \frac{1}{d}\right] : \sum_{i:\bar{P}_{pub}^{[i]} \leq t} \bar{P}_{pub}^{[i]} \leq \gamma\}$ and $\mathrm{WF}(\bar{P}_{pub}, t)$ is the water-filled version of $\bar{P}_{pub}$ with threshold $t$ (see Def. 4.2).*

**Proof:** [Proof of Lemma E.2] We first prove that, for any $t \in [0, t_{max}]$:

$$\mathrm{WF}(\bar{P}_{pub}, t) = \arg \min_{\substack{P_{nse} \in \Delta_d \\ N_{min_1} = t}} \left( \mathrm{TV}(\bar{P}_{pub}, P_{nse}) + \sum_{i=1}^{k} \bar{P}_{pub}^{[i]} + \max_{1 \leq u \leq d-1-k} \phi(u, t, P_{nse}) \right) \quad (173)$$

and then show that $\mathrm{WF}(\bar{P}_{pub}, t)$ satisfies $\sum_{i:\bar{P}_{pub}^{[i]} \leq \tilde{P}_{nse}^{[i]}(t)} \bar{P}_{pub}^{[i]} \leq \gamma$ whenever $0 \leq t \leq t_{max}$. Since

$$\min_{\substack{P_{nse} \in \Delta_d \\ N_{min_1} = t}} \left( \mathrm{TV}(\bar{P}_{pub}, P_{nse}) + \sum_{i=1}^{k} \bar{P}_{pub}^{[i]} + \max_{1 \leq u \leq d-1-k} \phi(u, t, P_{nse}) \right)$$

$$\leq \min_{\substack{P_{nse} \in \Delta_d \\ N_{min_1} = t \\ \sum_{i:\bar{P}_{pub}^{[i]} \leq \tilde{P}_{nse}^{[i]}(t)} \bar{P}_{pub}^{[i]} \leq \gamma}} \left( \mathrm{TV}(\bar{P}_{pub}, P_{nse}) + \sum_{i=1}^{k} \bar{P}_{pub}^{[i]} + \max_{1 \leq u \leq d-1-k} \phi(u, t, P_{nse}) \right) \quad (174)$$

This shows that $\mathrm{WF}(\bar{P}_{pub}, t)$ is optimal in (172) for any $t \in [0, t_{max}]$. Furthermore, we prove that there exists no $P_{nse}$ with $N_{min_1} = t > t_{max}$ satisfying $\sum_{i:\bar{P}_{pub}^{[i]} \leq \tilde{P}_{nse}^{[i]}(t)} \bar{P}_{pub}^{[i]} \leq \gamma$, to complete the proof of Lemma E.2.

First consider the proof of (173). Recall that $k$ is a function of $P_{nse}$ for any fixed $\bar{P}_{pub}$. From here onwards, we make this

dependency explicit by writing $k$ as $k(P_{nse})$, to avoid any confusion.

$$\min_{\substack{P_{nse}\in\Delta_d \\ N_{min_1}=t}} \left( \text{TV}(\bar{P}_{pub}, P_{nse}) + \sum_{i=1}^{k(P_{nse})} \bar{P}_{pub}^{[i]} + \max_{1\le u\le d-1-k(P_{nse})} \phi(u, P_{nse}) \right) \tag{175}$$

$$= \min_{\substack{P_{nse}\in\Delta_d \\ N_{min_1}=t}} \max_{1\le u\le d-1-k(P_{nse})} \left( \text{TV}(\bar{P}_{pub}, P_{nse}) + \sum_{i=1}^{k(P_{nse})} \bar{P}_{pub}^{[i]} + \phi(u, P_{nse}) \right) \tag{176}$$

$$= \min_{\substack{P_{nse}\in\Delta_d \\ N_{min_1}=t}} \max_{1\le u\le d-1-k(P_{nse})} \left( \text{TV}(\bar{P}_{pub}, P_{nse}) + \sum_{i=1}^{k(P_{nse})} \bar{P}_{pub}^{[i]} \right.$$

$$\left. + \min\left\{ \left[ \gamma - \sum_{i=1}^{k(P_{nse})+u} \bar{P}_{pub}^{[i]} + \sum_{i=k(P_{nse})+1}^{k(P_{nse})+u} P_{nse}^{[i]} \right]_+ , \sum_{i=k(P_{nse})+1}^{k(P_{nse})+u} P_{nse}^{[i]} \right\} \right) \tag{177}$$

$$= \min_{\substack{P_{nse}\in\Delta_d \\ N_{min_1}=t}} \max_{1\le u\le d-1-k(P_{nse})} \min\left\{ \left[ \gamma - \sum_{i=1}^{k(P_{nse})+u} \bar{P}_{pub}^{[i]} + \sum_{i=k(P_{nse})+1}^{k(P_{nse})+u} P_{nse}^{[i]} \right]_+ + \text{TV}(\bar{P}_{pub}, P_{nse}) + \sum_{i=1}^{k(P_{nse})} \bar{P}_{pub}^{[i]}, \right.$$

$$\left. \sum_{i=k(P_{nse})+1}^{k(P_{nse})+u} P_{nse}^{[i]} + \text{TV}(\bar{P}_{pub}, P_{nse}) + \sum_{i=1}^{k(P_{nse})} \bar{P}_{pub}^{[i]} \right\} \tag{178}$$

$$= \min_{\substack{P_{nse}\in\Delta_d \\ N_{min_1}=t}} \max_{1\le u\le d-1-k(P_{nse})} \min\left\{ \left[ \gamma - \sum_{i=1}^{k(P_{nse})+u} \bar{P}_{pub}^{[i]} + \sum_{i=k(P_{nse})+1}^{k(P_{nse})+u} P_{nse}^{[i]} \right]_+ + \sum_{i=1}^{k(P_{nse})} P_{nse}^{[i]}, \sum_{i=1}^{k(P_{nse})+u} P_{nse}^{[i]} \right\} \tag{179}$$

$$= \min_{\substack{P_{nse}\in\Delta_d \\ N_{min_1}=t}} \max_{1\le u\le d-1-k(P_{nse})} \min\{f(P_{nse}, u), g(P_{nse}, u)\} \tag{180}$$

where

$$f(P_{nse}, u) = \left[ \gamma - \sum_{i=1}^{k(P_{nse})+u} \bar{P}_{pub}^{[i]} + \sum_{i=k(P_{nse})+1}^{k(P_{nse})+u} P_{nse}^{[i]} \right]_+ + \sum_{i=1}^{k(P_{nse})} P_{nse}^{[i]} \tag{181}$$

$$= \max\left\{ \gamma - \sum_{i=1}^{k(P_{nse})+u} (\bar{P}_{pub}^{[i]} - P_{nse}^{[i]}), \sum_{i=1}^{k(P_{nse})} P_{nse}^{[i]} \right\} \tag{182}$$

$$= \max\left\{ \gamma + \sum_{i=1}^{k(P_{nse})} (P_{nse}^{[i]} - \bar{P}_{pub}^{[i]}) - \sum_{i=k(P_{nse})+1}^{k(P_{nse})+u} (\bar{P}_{pub}^{[i]} - P_{nse}^{[i]}), \sum_{i:\bar{P}_{pub}^{[i]}\le P_{nse}^{[i]}} P_{nse}^{[i]} \right\} \tag{183}$$

and

$$g(P_{nse}, u) = \sum_{i=1}^{k(P_{nse})+u} P_{nse}^{[i]} \tag{184}$$

Next, we characterize the maximizing $u$ in (180). Note that for a given $P_{nse}$, $f(P_{nse}, u)$ is non-increasing in $u$ and $g(P_{nse}, u)$ is non-decreasing in $u$. Thus, for any given $P_{nse}$, the inner max in (180) occurs at either:

$$\tilde{u}_1(P_{nse}) = \max\{u : f(P_{nse}, u) > g(P_{nse}, u)\} \tag{185}$$

or

$$\tilde{u}_2(P_{nse}) = \min\{u : f(P_{nse}, u) \le g(P_{nse}, u)\} \tag{186}$$

satisfying $\tilde{u}_1(P_{nse}) = \tilde{u}_2(P_{nse}) - 1$ or $\tilde{u}_1(P_{nse}) = \tilde{u}_2(P_{nse})$, which simplifies (180) to:

$$\min_{\substack{P_{nse} \in \Delta_d \\ N_{min_1} = t}} \left( \mathrm{TV}(\bar{P}_{pub}, P_{nse}) + \sum_{i=1}^{k(P_{nse})} \bar{P}_{pub}^{[i]} + \max_{1 \leq u \leq d-1-k(P_{nse})} \phi(u, P_{nse}) \right) \tag{187}$$

$$= \min_{\substack{P_{nse} \in \Delta_d \\ N_{min_1} = t}} \max_{1 \leq u \leq d-1-k(P_{nse})} \min\{f(P_{nse}, u), g(P_{nse}, u)\} \tag{188}$$

$$= \min_{\substack{P_{nse} \in \Delta_d \\ N_{min_1} = t}} \max\{g(P_{nse}, \tilde{u}_1(P_{nse}), f(P_{nse}, \tilde{u}_2(P_{nse})))\} \tag{189}$$

**Claim** E.3. Denote $P_{nse}^* = \mathrm{WF}(\bar{P}_{pub}, t)$. Then,

$$\max_{1 \leq u \leq d-1-k(P_{nse}^*)} \min\{f(P_{nse}^*, u), g(P_{nse}^*, u)\} \leq \max_{1 \leq u \leq d-1-k(P_{nse})} \min\{f(P_{nse}, u), g(P_{nse}, u)\} \tag{190}$$

$\forall P_{nse} \in \Delta_d$ with $N_{min_1} = t$, or equivalently,

$$\max\{g(P_{nse}^*, \tilde{u}_1(P_{nse}^*), f(P_{nse}^*, \tilde{u}_2(P_{nse}^*)))\} \leq \max\{g(P_{nse}, \tilde{u}_1(P_{nse}), f(P_{nse}, \tilde{u}_2(P_{nse})))\} \tag{191}$$

$\forall P_{nse} \in \Delta_d$ with $N_{min_1} = t$.

The proof of Claim E.3 is given in sec. E.1. Continuing from (189), together with Claim E.3, we have,

$$\min_{\substack{P_{nse} \in \Delta_d \\ N_{min_1} = t}} \left( \mathrm{TV}(\bar{P}_{pub}, P_{nse}) + \sum_{i=1}^{k(P_{nse})} \bar{P}_{pub}^{[i]} + \max_{1 \leq u \leq d-1-k(P_{nse})} \phi(u, t, P_{nse}) \right) \tag{192}$$

$$= \min_{\substack{P_{nse} \in \Delta_d \\ N_{min_1} = t}} \max\{g(P_{nse}, \tilde{u}_1(P_{nse}), f(P_{nse}, \tilde{u}_2(P_{nse})))\} \tag{193}$$

$$= \max\{g(P_{nse}^*, \tilde{u}_1(P_{nse}^*), f(P_{nse}^*, \tilde{u}_2(P_{nse}^*)))\} \tag{194}$$

In other words,

$$\mathrm{WF}(\bar{P}_{pub}, t) = \arg\min_{\substack{P_{nse} \in \Delta_d \\ N_{min_1} = t}} \left( \mathrm{TV}(\bar{P}_{pub}, P_{nse}) + \sum_{i=1}^{k(P_{nse})} \bar{P}_{pub}^{[i]} + \max_{1 \leq u \leq d-1-k(P_{nse})} \phi(u, t, P_{nse}) \right) \tag{195}$$

This completes the proof of (173), and the next step is to show that $\mathrm{WF}(\bar{P}_{pub}, t)$ satisfies $\sum_{i:\bar{P}_{pub}^{[i]} \leq \mathrm{WF}^{[i]}(\bar{P}_{pub}, t)} \bar{P}_{pub}^{[i]} \leq \gamma$ whenever $0 \leq t \leq t_{max}$.

For any fixed $0 < t \leq \frac{1}{d}$ with $N_{min_1} = t$, we have the bound:

$$\sum_{i:\bar{P}_{pub}^{[i]} \leq t} \bar{P}_{pub}^{[i]} \leq \sum_{i:\bar{P}_{pub}^{[i]} \leq P_{nse}^{[i]}} \bar{P}_{pub}^{[i]} \tag{196}$$

as $P_{nse}^{[i]} \geq t$, $\forall i$. When $P_{nse} = \mathrm{WF}(\bar{P}_{pub}, t)$, we have, $\sum_{i:\bar{P}_{pub}^{[i]} \leq P_{nse}^{[i]}} \bar{P}_{pub}^{[i]} = \sum_{i:\bar{P}_{pub}^{[i]} \leq t} \bar{P}_{pub}^{[i]}$. Thus, if the constraint $\sum_{i:\bar{P}_{pub}^{[i]} \leq P_{nse}^{[i]}} \bar{P}_{pub}^{[i]} \leq \gamma$ is not satisfied by $\mathrm{WF}(\bar{P}_{pub}, t)$ for a given $t$, there exists no other $P_{nse}$ that satisfies it. The maximum value of $t$ for which $\mathrm{WF}(\bar{P}_{pub}, t)$ satisfies $\sum_{i:\bar{P}_{pub}^{[i]} \leq \mathrm{WF}^{[i]}(\bar{P}_{pub}, t)} \bar{P}_{pub}^{[i]} = \sum_{i:\bar{P}_{pub}^{[i]} \leq t} \bar{P}_{pub}^{[i]} \leq \gamma$ is given by $t_{max} = \arg\max\{t \in [0, 1/d] : \sum_{i:\bar{P}_{pub}^{[i]} \leq t} \bar{P}_{pub}^{[i]} \leq \gamma\}$. Therefore, $\mathrm{WF}(\bar{P}_{pub}, t)$ with its minimum element at $t > t_{max}$ does not satisfy $\sum_{i:\bar{P}_{pub}^{[i]} \leq \bar{P}_{nse}^{[i]}(t)} \bar{P}_{pub}^{[i]} \leq \gamma$, and neither does any other $P_{nse}$. This completes the proof of Lemma E.2. ∎

Now, based on $\mathrm{WF}(\bar{P}_{pub}, t)$, we know that the ordering of $\bar{P}_{pub}^{[1]} - \mathrm{WF}^{[1]}(\bar{P}_{pub}, t) \leq \ldots \leq \bar{P}_{pub}^{[d]} - \mathrm{WF}^{[d]}(\bar{P}_{pub}, t)$ is the

same as $\bar{P}_{pub}^{[1]} \leq \ldots \leq \bar{P}_{pub}^{[d]}$. With this, we can simplify (171) as:

$$
\min_{\substack{P_{nse} \in \Delta_d \\ \sum_{i:\bar{P}_{pub}^{[i]} \leq P_{nse}^{[i]}} \bar{P}_{pub}^{[i]} \leq \gamma}} \quad \min_{\substack{\beta \in [0,1] \\ \beta \geq \beta_{min}(N_{min_1}, N_{min_1})}} \quad \max_{\bar{P}_{prv} \in \mathcal{B}_{\gamma, \bar{P}_{pub}}} \quad \mathrm{TV}(\bar{P}_{prv}, (1-\beta)\bar{P}_{prv} + \beta P_{nse})
$$

$$
= \min_{0 \leq t \leq t_{max}} \quad \beta_{min}(t, t) \left( \mathrm{TV}(\bar{P}_{pub}, \mathrm{WF}(\bar{P}_{pub}, t)) + \sum_{i:\bar{P}_{pub}^{[i]} \leq t} \bar{P}_{pub}^{[i]} + \max_{1 \leq u \leq d-1-k_t} \phi(u, \mathrm{WF}(\bar{P}_{pub}, t)) \right) \tag{197}
$$

$$
= \min_{0 \leq t \leq t_{max}} \beta_{min}(t, t) \Phi_2(t) \tag{198}
$$

where

$$
\Phi_2(t) = \mathrm{TV}(\bar{P}_{pub}, \mathrm{WF}(\bar{P}_{pub}, t)) + \sum_{i:\bar{P}_{pub}^{[i]} \leq t} \bar{P}_{pub}^{[i]} + \max_{1 \leq u \leq d-1-k_t} \phi(u, \mathrm{WF}(\bar{P}_{pub}, t)) \tag{199}
$$

$$
= k_t t + \max_{1 \leq u \leq d-1-k_t} \phi(u, \mathrm{WF}(\bar{P}_{pub}, t)) \tag{200}
$$

In the theorem statement, the set $T_2$ in the theorem statement is equivalent to $T_2 = \{t : 0 \leq t \leq t_{max}\}$ considered in (198).

Since cases 1 and 2 consider disjoint sets of $P_{nse}$ we need to choose the minimum of the two case-wise minima. This completes the proof of Theorem 4.4.

∎

The only remaining component of the proof of Theorem 4.4 is the proof of Claim E.3. For this, we need the following arguments.

For any given $P_{nse}$, $k(P_{nse})$ is the cardinality of the set $\{i : \bar{P}_{pub}^{[i]} \leq P_{nse}^{[i]}\}$. Because of the $N_{min_1} = t$ constraint on $P_{nse}$, we have,

$$
k(P_{nse}) \geq k_t = \sum_{i=1}^{d} \mathbf{1}_{\{\bar{P}_{pub}^{[i]} \leq t\}} \tag{201}
$$

For any fixed $u$ we consider the $P_{nse}$ that minimizes $f(P_{nse}, u)$ and $g(P_{nse}, u)$ separately. Note that for any given $P_{nse}$, $u \in \{1, \ldots, d-1-k(P_{nse})\}$ where $k(P_{nse}) = \sum_{i=1}^{d} \mathbf{1}_{\{\bar{P}_{pub}^{[i]} \leq P_{nse}^{[i]}\}}$ where $\mathbf{1}_{\{\cdot\}}$ is the indicator function. Since we have the $N_{min_1} = t$ constraint on all $P_{nse}$, $d - 1 - k(P_{nse}) \leq d - 1 - k_t$ where $k_t = \sum_{i=1}^{d} \mathbf{1}_{\{\bar{P}_{pub}^{[i]} \leq t\}}$. Thus, for any given $P_{nse}$ with $N_{min} = t$, we have $u \in \{1, \ldots, d-1-k(P_{nse})\} \subseteq \{1, \ldots, d-1-k_t\}$. Note that $k_t$ is independent of any specific $P_{nse}$ within the set of $P_{nse}$ under consideration.[12]

***Claim*** E.4. Denote $P_{nse}^* = \mathrm{WF}(\bar{P}_{pub}, t)$. For any fixed $u \in \{1, \ldots, d-1-k_t\}$,

$$
f(P_{nse}^*, u) \leq f(P_{nse}, u), \quad g(P_{nse}^*, u) \leq g(P_{nse}, u), \quad \forall P_{nse} \text{ s.t. } \min_i P_{nse}^{[i]} = t \tag{202}
$$

**Proof:** [Proof of Claim E.4] We first show that for $P_{nse} = \mathrm{WF}(\bar{P}_{pub}, t)$ the range of $u$ is given by $\{1, \ldots, d-1-k_t\}$. Then, we show that for any given $u \in \{1, \ldots, d-1-k_t\}$, (202) holds.

To see why the range of $u$ is $\{1, \ldots, d-1-k_t\}$ for $P_{nse} = \mathrm{WF}(\bar{P}_{pub}, t)$, consider the definition of water-filling in Def. 4.2. In Def. 4.2, all $\bar{P}_{pub}^{[i]} \leq t$ are raised to $t$ to obtain the corresponding $P_{nse}^{[i]}$, and the total raised probability mass is reduced from all $\bar{P}_{pub}^{[i]} > t$ collectively to obtain the rest of the $P_{nse}^{[i]}$, i.e., $\bar{P}_{pub}^{[i]} > P_{nse}^{[i]}$ is maintained for all indices $\{i : \bar{P}_{pub}^{[i]} > t\}$. The only indices $i$ where $\bar{P}_{pub}^{[i]} \leq P_{nse}^{[i]}$ are the ones in $\{i : \bar{P}_{pub}^{[i]} \leq t\}$. Thus,

$$
k(\mathrm{WF}(\bar{P}_{pub}, t)) = \sum_{i=1}^{d} \mathbf{1}_{\{\bar{P}_{pub}^{[i]} \leq P_{nse}^{[i]}\}} = \sum_{i=1}^{d} \mathbf{1}_{\{i:\bar{P}_{pub}^{[i]} \leq t\}} = k_t. \tag{203}
$$

---

[12]All $P_{nse}$ that we consider at this stage satisfy $N_{min_1} = t$.

This shows that for $P_{nse} = \text{WF}(\bar{P}_{pub}, t)$, $u \in \{1, \ldots, d-1-k(\text{WF}(\bar{P}_{pub}, t))\} = \{1, \ldots, d-1-k_t\}$.

Next, we prove that for any $u \in \{1, \ldots, d-1-k_t\}$, (202) holds. First consider the proof of $\arg\min_{P_{nse}:\min_i P_{nse}^{[i]}=t} f(P_{nse}, u) = P_{nse}^* = \text{WF}(\bar{P}_{pub}, t)$. For any fixed $u \in \{1, \ldots, d-1-k_t\}$, define:

$$f_1(P_{nse}, u) = \gamma + \sum_{i=k(P_{nse})+u+1}^{d} (\bar{P}_{pub}^{[i]} - P_{nse}^{[i]}) \tag{204}$$

$$f_2(P_{nse}, u) = \sum_{i:\bar{P}_{pub}^{[i]} \leq P_{nse}^{[i]}} P_{nse}^{[i]} \tag{205}$$

Note that $f_1(P_{nse}, u)$ contains the sum of the largest $d - k(P_{nse}) - u$ of the $\bar{P}_{pub}^{[i]} - P_{nse}^{[i]}$ terms (which are all positive). Since the order $\bar{P}_{pub}^{[i]} - P_{nse}^{[i]} \leq \bar{P}_{pub}^{[i+1]} - P_{nse}^{[i+1]}$ must hold for all $i$, it is clear that:

$$P_{nse}^{[i]} = \max\left\{t, \bar{P}_{pub}^{[i]} - \tau\right\} \tag{206}$$

with $\tau$ chosen to satisfy $\sum_{i=1}^{d} P_{nse}^{[i]} = 1$, minimizes $f_1(P_{nse}, u)$. Moreover, $f_2(P_{nse}, u)$ is minimized when $\{i : \bar{P}_{pub}^{[i]} \leq P_{nse}^{[i]}\} = \{i : \bar{P}_{pub}^{[i]} \leq t\}$ and $P_{nse}^{[i]} = t$ for all $\{i : \bar{P}_{pub}^{[i]} \leq t\}$, since $P_{nse}^{[i]} \geq t$ must be satisfied. Both of these properties are satisfied by the $P_{nse}$ described in (206). Thus, for any fixed $u$, both $f_1(P_{nse}, u)$ and $f_2(P_{nse}, u)$ are minimized by the same $P_{nse}$ in (206), which is exactly equal to $\text{WF}(\bar{P}_{pub}, t)$ (see Def. 4.2). Consequently, the minimizer of $f(P_{nse}, u) = \max\{f_1(P_{nse}, u), f_2(P_{nse}, u)\}$ is also $\text{WF}(\bar{P}_{pub}, t)$.

Now consider the analysis of $g(P_{nse}, u)$. For any fixed $u \in \{1, \ldots, d-1-k_t\}$,

$$g(P_{nse}, u) = \sum_{i=1}^{k(P_{nse})+u} P_{nse}^{[i]} \tag{207}$$

$$= \sum_{i:\bar{P}_{pub}^{[i]} \leq P_{nse}^{[i]}} P_{nse}^{[i]} + \sum_{i=k(P_{nse})+1}^{k(P_{nse})+u} P_{nse}^{[i]} \tag{208}$$

The first term in (208) is minimized by (206) following the same argument as $f_2(P_{nse}, u)$. The second term in (208) is the sum of the $P_{nse}^{[i]}$ terms corresponding to the indices $i$ with the smallest $u$ positive $\bar{P}_{pub}^{[i]} - P_{nse}^{[i]}$ differences. The $P_{nse}$ in (206) ensures that $P_{nse}^{[i]} = t$ (minimum possible) is satisfied for the indices $i$ corresponding to the smallest few positive $\bar{P}_{pub}^{[i]} - P_{nse}^{[i]}$ differences. For the rest of the terms among the $u$ smallest terms, (206) chooses $P_{nse}^{[i]} = \bar{P}_{pub}^{[i]} - \tau$, satisfying the order constraint $\bar{P}_{pub}^{[i]} - P_{nse}^{[i]} \leq \bar{P}_{pub}^{[i+1]} - P_{nse}^{[i+1]}$ while minimizing the sum of the $P_{nse}^{[i]}$ terms corresponding to the smallest $u$ positive $\bar{P}_{pub}^{[i]} - P_{nse}^{[i]}$ differences.[13] This shows that for any given $u \in \{1, \ldots, d-1-k_t\}$, the term $g(P_{nse}, u)$ is also minimized by $\text{WF}(\bar{P}_{pub}, t)$. This completes the proof of Claim E.4.

∎

### E.1. Proof of Claim E.3

**Proof:** [Proof of Claim E.3] Consider any $P_{nse}$ with $N_{min_1} = t$. Let the range of $u$ be given by $\mathcal{R}(P_{nse})$, i.e., $u \in \{1, \ldots, d-1-k(P_{nse})\} = \mathcal{R}(P_{nse})$. We consider several cases to prove Claim E.3

**Case A:** $\tilde{u}_1(P_{nse}^*), \tilde{u}_2(P_{nse}^*) \in \mathcal{R}(P_{nse})$

For any $P_{nse} \in \Delta_d$ satisfying $N_{min_1} = t$, consider any $u \in \mathcal{R}(P_{nse})$, define:

$$h(P_{nse}, u) = \min\{f(P_{nse}, u), g(P_{nse}, u)\} \tag{209}$$

---

[13]One important point here is that since we have $\bar{P}_{pub}^{[i]} - P_{nse}^{[i]} = \bar{P}_{pub}^{[i+1]} - P_{nse}^{[i+1]} = \tau$ for larger $i$ terms, we make sure that the indices are arranged such that $\bar{P}_{pub}^{[i]} \leq \bar{P}_{pub}^{[i+1]}$ so that the differences $\bar{P}_{pub}^{[i]} - P_{nse}^{[i]}$ still satisfy the order constraint and we get $P_{nse}^{[i]} \leq P_{nse}^{[i+1]}$.

we know that:

$$h(P_{nse}^*, u) = \min\{f(P_{nse}^*, u), g(P_{nse}^*, u)\} \leq \min\{f(P_{nse}, u), g(P_{nse}, u)\} = h(P_{nse}, u), \quad \forall u \in \mathcal{R}(P_{nse}) \tag{210}$$

from Claim E.4. Since (210) holds for all $u \in \mathcal{R}(P_{nse})$, we have,

$$\max_{u \in \mathcal{R}(P_{nse}^*)} h(P_{nse}^*, u) = \max_{u \in \mathcal{R}(P_{nse})} h(P_{nse}^*, u) \leq \max_{u \in \mathcal{R}(P_{nse})} h(P_{nse}, u) \tag{211}$$

where the first equality comes from the fact that $\tilde{u}_1(P_{nse}^*), \tilde{u}_2(P_{nse}^*) \in \mathcal{R}(P_{nse})$ by assumption. This completes the proof of Claim E.3 for case A.

Notice that $\mathcal{R}(P_{nse}^*) = \{1, \ldots, d-1-k_t\}$ and $\mathcal{R}(P_{nse}) \subseteq \mathcal{R}(P_{nse}^*)$ for any $P_{nse} \in \Delta_d$ satisfying $N_{min_1} = t$. Moreover, $\mathcal{R}(P_{nse}^*) \setminus \mathcal{R}(P_{nse}) = \{d-k(P_{nse}), \ldots, d-1-k_t\}$. In other words, the additional options for $u$ with $P_{nse}^*$ come only at the end (larger values for $u$).

**Case B:** $\tilde{u}_1(P_{nse}), \tilde{u}_2(P_{nse}) \in \mathcal{R}(P_{nse})$ and either 1) $\tilde{u}_1(P_{nse}^*) \in \mathcal{R}(P_{nse})$ and $\tilde{u}_2(P_{nse}^*) \notin \mathcal{R}(P_{nse})$ or 2) $\tilde{u}_1(P_{nse}^*), \tilde{u}_2(P_{nse}^*) \notin \mathcal{R}(P_{nse})$.

By definition,

$$f(P_{nse}, \tilde{u}_2(P_{nse})) \leq \max\{g(P_{nse}, \tilde{u}_1(P_{nse}), f(P_{nse}, \tilde{u}_2(P_{nse})))\} \leq f(P_{nse}, \tilde{u}_1(P_{nse})) \tag{212}$$

By the assumption in case B,[14]

$$\tilde{u}_1(P_{nse}) \leq \tilde{u}_2(P_{nse}) \leq d-1-k(P_{nse}) \leq \tilde{u}_1(P_{nse}^*) \leq \tilde{u}_2(P_{nse}^*) \tag{213}$$

as $\tilde{u}_1(P_{nse}^*) = \tilde{u}_2(P_{nse}^*) - 1$ or $\tilde{u}_1(P_{nse}^*) = \tilde{u}_2(P_{nse}^*)$ must be satisfied. Since $f(P_{nse}, u)$ is non-increasing in $u$,

$$f(P_{nse}, d-1-k(P_{nse})) \leq f(P_{nse}, \tilde{u}_2(P_{nse})) \tag{214}$$

From Claim E.4,

$$f(P_{nse}^*, d-1-k(P_{nse})) \leq f(P_{nse}, d-1-k(P_{nse})) \tag{215}$$

Now, applying the non-increasing property of $f(P_{nse}^*, u)$ in $u$ again,

$$f(P_{nse}^*, \tilde{u}_1(P_{nse}^*)) \leq f(P_{nse}^*, d-1-k(P_{nse})) \tag{216}$$

Applying (212) to $P_{nse}^*$, we get,

$$f(P_{nse}^*, \tilde{u}_2(P_{nse}^*)) \leq \max\{g(P_{nse}^*, \tilde{u}_1(P_{nse}^*), f(P_{nse}^*, \tilde{u}_2(P_{nse}^*)))\} \leq f(P_{nse}^*, \tilde{u}_1(P_{nse}^*)) \tag{217}$$

Combining (212)-(217) proves Claim E.3 for case B.

**Case C:** $f(P_{nse}, u)$ and $g(P_{nse}, u)$ never cross, i.e., $\tilde{u}_1(P_{nse}) = d-1-k(P_{nse})$ with no corresponding $\tilde{u}_2(P_{nse})$ or $\tilde{u}_2(P_{nse}) = 1$ with no corresponding $\tilde{u}_1(P_{nse})$, and either 1) $\tilde{u}_1(P_{nse}^*) \in \mathcal{R}(P_{nse})$ and $\tilde{u}_2(P_{nse}^*) \notin \mathcal{R}(P_{nse})$ or 2) $\tilde{u}_1(P_{nse}^*), \tilde{u}_2(P_{nse}^*) \notin \mathcal{R}(P_{nse})$.

For any given $P_{nse} \in \Delta_d$ satisfying $N_{min_1} = t$, when $\tilde{u}_1(P_{nse}) = d-1-k(P_{nse})$, we have:

$$\max_{1 \leq u \leq d-1-k(P_{nse})} \min\{f(P_{nse}, u), g(P_{nse}, u)\} = g(P_{nse}, d-1-k(P_{nse})) = \sum_{i=1}^{d-1} P_{nse}^{[i]} = 1 - P_{nse}^{[d]} \tag{218}$$

---

[14]If $\tilde{u}_1(P_{nse}^*), \tilde{u}_2(P_{nse}^*) \notin \mathcal{R}(P_{nse})$, (213) directly holds. If $\tilde{u}_1(P_{nse}^*) \in \mathcal{R}(P_{nse})$ and $\tilde{u}_2(P_{nse}^*) \notin \mathcal{R}(P_{nse})$, note that $\tilde{u}_1(P_{nse}^*) = d-1-k(P_{nse})$ must be satisfied since $\tilde{u}_1(P_{nse}^*) = \tilde{u}_2(P_{nse}^*) - 1$ or $\tilde{u}_1(P_{nse}^*) = \tilde{u}_2(P_{nse}^*)$. This means (213) always holds.

At the same time, since $g(P_{nse}^*, u)$ is non-decreasing in $u$ and $f(P_{nse}^*, u)$ is non-increasing in $u$, we have:

$$\max_{1 \leq u \leq d-1-k(P_{nse}^*)} \min\{f(P_{nse}^*, u), g(P_{nse}^*, u)\} \leq g(P_{nse}^*, d-1-k(P_{nse}^*)) = \sum_{i=1}^{d-1} P_{nse}^{*[i]} = 1 - P_{nse}^{*[d]} \tag{219}$$

Note that we have,

$$1 - P_{nse}^{*[d]} \leq 1 - P_{nse}^{[d]}, \quad \forall P_{nse} \in \Delta_d \text{ with } N_{min_1} = t \tag{220}$$

since in $P_{nse}^* = \mathrm{WF}(\bar{P}_{pub}, t)$, $P_{nse}^{*[d]} = (\max_i \bar{P}_{pub}^{[i]}) - \tau \geq P_{nse}^{[d]}$, $\forall P_{nse} \in \Delta_d$ with $N_{min_1} = t$, as $\tau$ is the smallest $\bar{P}_{pub}^{[i]} - P_{nse}^{[i]}$ difference one could achieve while preserving the order $\bar{P}_{pub}^{[i]} - P_{nse}^{[i]} \leq \bar{P}_{pub}^{[i+1]} - P_{nse}^{[i+1]}$, $\forall i$ (see Def. 4.2).

Then, we have,

$$\max_{1 \leq u \leq d-1-k_t} \min\{f(P_{nse}^*, u), g(P_{nse}^*, u)\} \leq 1 - P_{nse}^{*[d]} \tag{221}$$

$$\leq 1 - P_{nse}^{[d]} \tag{222}$$

$$= \max_{1 \leq u \leq d-1-k(P_{nse})} \min\{f(P_{nse}, u), g(P_{nse}, u)\} \tag{223}$$

from (218) and (219).

On the other extreme, if $f(P_{nse}, u)$ and $g(P_{nse}, u)$ never cross and $\tilde{u}_2(P_{nse}) = 1$,

$$\max_{1 \leq u \leq d-1-k(P_{nse})} \min\{f(P_{nse}, u), g(P_{nse}, u)\} = f(P_{nse}, \tilde{u}_2(P_{nse})) = f(P_{nse}, 1) \tag{224}$$

At the same time, since $g(P_{nse}^*, u)$ is non-decreasing in $u$ and $f(P_{nse}^*, u)$ is non-increasing in $u$, we have:

$$\max_{1 \leq u \leq d-1-k(P_{nse}^*)} \min\{f(P_{nse}^*, u), g(P_{nse}^*, u)\} \leq f(P_{nse}^*, 1) \tag{225}$$

Since $f(P_{nse}^*, 1) \leq f(P_{nse}, 1)$, $\forall P_{nse} \in \Delta_d$ s.t. $N_{min} = t$ from Claim E.4, we have,

$$\max_{1 \leq u \leq d-1-k(P_{nse}^*)} \min\{f(P_{nse}^*, u), g(P_{nse}^*, u)\} \leq f(P_{nse}^*, 1) \tag{226}$$

$$\leq f(P_{nse}, 1) \tag{227}$$

$$= \max_{1 \leq u \leq d-1-k(P_{nse})} \min\{f(P_{nse}, u), g(P_{nse}, u)\} \tag{228}$$

and Claim E.3 is proved. ∎

## F. Proof of Corollary 4.5

**Corollary 4.5 restated:** For any $D_{prv}$, $m \geq 1$, $\varepsilon > 0$ and $\delta \in (0, 1)$, the asymptotic solution to (3) with $B_{\gamma, S} = \Delta_d$ is given by,

$$\min_{\substack{P_{nse} \in \Delta_d}} \min_{\substack{\beta \in [0,1] \\ \beta \geq \beta_{min}(N_{min_1}, N_{min_1})}} \max_{\bar{P}_{prv} \in \Delta_d} \mathrm{TV}(\bar{P}_{prv}, P_{syn}) = \left(1 - \frac{1}{d}\right) \beta_{min}(1/d, 1/d) \tag{229}$$

and the corresponding optimum $\beta^*$ and $P_{nse}^*$ are given by,

$$P_{nse}^* = \frac{1}{d} \mathbf{1}_d \tag{230}$$

$$\beta^* = \beta_{min}(1/d, 1/d) \tag{231}$$

*Proof.* Let $\bar{P}_{prv}^{[i]}$ and $P_{nse}^{[i]}$ denote the $i$th elements of $\bar{P}_{prv}$ and $P_{nse}$, respectively. Consider,

$$\min_{\substack{P_{nse}\in\Delta_d}} \min_{\substack{\beta\in[0,1] \\ \beta\geq\beta_{min}(N_{min_1},N_{min_1})}} \max_{\bar{P}_{prv}\in\Delta_d} \mathrm{TV}(\bar{P}_{prv},(1-\beta)\bar{P}_{prv}+\beta P_{nse})$$

$$= \min_{\substack{P_{nse}\in\Delta_d}} \min_{\substack{\beta\in[0,1] \\ \beta\geq\beta_{min}(N_{min_1},N_{min_1})}} \max_{\bar{P}_{prv}\in\Delta_d} \frac{\beta}{2}\sum_{i=1}^{d}|\bar{P}_{prv}^{[i]} - P_{nse}^{[i]}| \tag{232}$$

$$= \min_{\substack{P_{nse}\in\Delta_d}} \min_{\substack{\beta\in[0,1] \\ \beta\geq\beta_{min}(N_{min_1},N_{min_1})}} \frac{\beta}{2}\sum_{\substack{i=1,i\neq\arg\min_j P_{nse}^{[j]}}}^{d} P_{nse}^{[i]} + \frac{\beta}{2}(1 - N_{min_1}), \quad (\text{since } \min_j P_{nse}^{[j]} = N_{min_1}) \tag{233}$$

$$= \min_{\substack{P_{nse}\in\Delta_d}} \min_{\substack{\beta\in[0,1] \\ \beta\geq\beta_{min}(N_{min_1},N_{min_2})}} \beta(1 - N_{min_1}) \tag{234}$$

$$= \min_{\substack{P_{nse}\in\Delta_d}} \beta_{min}(N_{min_1},N_{min_1})(1 - N_{min_1}) \tag{235}$$

From Lemma C.5, we know that $\beta_{min}(N_{min_1},N_{min_1})$ is decreasing in $N_{min_1}$. Then, as $\beta_{min}(N_{min_1},N_{min_1})$ and $1 - N_{min}$ are both decreasing in $N_{min_1}$, we have the optimal $N_{min_1} = \frac{1}{d}$, and the corresponding optimal $P_{nse}^* = \frac{1}{d}\mathbf{1}_d$. The resulting minimum TV is given in (229).

$\square$

# G. Extension to General Discrete Distributions

**Lemma G.1.** *Fix $\varepsilon > 0$, $\delta \in (0,1)$, and the required number of synthetic samples $m \geq 1$. Let $P_{\mathrm{nse}} \in \Delta_d$ be any noise distribution and define $p := \min_{y\in\{1,\ldots,d\}} P_{\mathrm{nse}}(y) > 0$. Define the sensitivity as $s = \max_i |\bar{P}_{prv}^{[i]} - \bar{P}'^{[i]}_{prv}|$, considering all neighboring private distributions $\bar{P}_{prv}, \bar{P}'_{prv}$ corresponding to neighboring datasets $D_{prv}, D'_{prv}$. Then, for every $\beta \geq \beta_{\min}$, the $m$ synthetic samples drawn from $P_{syn}$ in (1) satisfy the DP constraint in (2). $\beta_{min}$ is a function of $\varepsilon, \delta, m, n, P_{nse}$. Moreover, $\beta_{\min}$ depends on $P_{\mathrm{nse}}$ only through $p$.*

*Proof.* Consider the privacy loss random variable (PLRV) in (2):

$$\log\frac{\mathbb{P}(Y_{P_{syn}} = \{y_1,\ldots,y_m\})}{\mathbb{P}(Y_{P'_{syn}} = \{y_1,\ldots,y_m\})} = \sum_{i=1}^{m}\log\frac{P_{syn}(y_i)}{P'_{syn}(y_i)} \tag{236}$$

Define $L(y_i) = \log\frac{P_{syn}(y_i)}{P'_{syn}(y_i)}$, which are i.i.d. Note that,

$$P_{syn}(y) = (1-\beta)\bar{P}_{prv}(y) + \beta P_{nse}(y) \tag{237}$$
$$P'_{syn}(y) = (1-\beta)\bar{P}'_{prv}(y) + \beta P_{nse}(y) \tag{238}$$

where,

$$\bar{P}'_{prv}(y) - \min\{s,\bar{P}'_{prv}(y)\} \leq \bar{P}_{prv}(y) \leq \bar{P}'_{prv}(y) + \min\{s,1-\bar{P}'_{prv}(y)\}, \quad \forall y \in \{1,\ldots,d\} \tag{239}$$

such that $\sum_{y=1}^{d}\bar{P}_{prv}(y) = 1$. Then,

$$1 - \frac{(1-\beta)\min\{s,\bar{P}'_{prv}(y)\}}{P'_{syn}(y)} \leq \frac{P_{syn}(y)}{P'_{syn}(y)} \leq 1 + \frac{(1-\beta)\min\{s,1-\bar{P}'_{prv}(y)\}}{P'_{syn}(y)} \tag{240}$$

Note that if $s \leq \bar{P}'_{prv}(y)$ the lower bound simplifies to:

$$1 - \frac{(1-\beta)s}{P'_{syn}(y)} \geq 1 - \frac{(1-\beta)s}{(1-\beta)s + \beta p} = 1 - \frac{1}{1 + \frac{\beta p}{(1-\beta)s}} \tag{241}$$

If $s > \bar{P}'_{prv}(y)$ the lower bound simplifies to:

$$1 - \frac{(1-\beta)\bar{P}'_{prv}(y)}{(1-\beta)\bar{P}'_{prv}(y) + \beta p} \geq 1 - \frac{1}{1 + \frac{\beta p}{(1-\beta)s}} \tag{242}$$

Similarly, the upper bound in (240) is simplified to $1 + \frac{(1-\beta)s}{\beta p}$. Thus, we have,

$$L = 1 - \frac{1}{1 + \frac{\beta p}{(1-\beta)s}} \leq \frac{P_{syn}(y)}{P'_{syn}(y)} \leq 1 + \frac{(1-\beta)s}{\beta p} = U \tag{243}$$

Therefore, any $t = \frac{P_{syn}(y)}{P'_{syn}(y)}$ can be written as $t = \theta L + (1-\theta)U$ for $\theta = \frac{U-t}{U-L} \in [0,1]$.

To satisfy (2), we need:

$$\mathbb{P}\left(\sum_{i=1}^{m} \log \frac{P_{syn}(y_i)}{P'_{syn}(y_i)} > \varepsilon\right) \leq \delta \implies \mathbb{P}\left(e^{\lambda \sum_{i=1}^{m} \log \frac{P_{syn}(y_i)}{P'_{syn}(y_i)}} > e^{\lambda\varepsilon}\right) \leq e^{-\lambda\varepsilon}\mathbb{E}\left[e^{\lambda \sum_{i=1}^{m} \log \frac{P_{syn}(y_i)}{P'_{syn}(y_i)}}\right] \leq \delta \tag{244}$$

Now, let $\lambda = \alpha - 1$ and consider,

$$\mathbb{E}\left[e^{\lambda \sum_{i=1}^{m} \log \frac{P_{syn}(y_i)}{P'_{syn}(y_i)}}\right] = \sum_{y_1,\ldots,y_m} \prod_{i=1}^{m} P_{syn}(y_i) e^{(\alpha-1)\sum_{i=1}^{m} \log \frac{P_{syn}(y_i)}{P'_{syn}(y_i)}} \tag{245}$$

$$= \sum_{y_1,\ldots,y_m} \prod_{i=1}^{m} P_{syn}(y_i)\left(\prod_{i=1}^{m} \frac{P_{syn}(y_i)}{P'_{syn}(y_i)}\right)^{\alpha-1} \tag{246}$$

$$= \prod_{i=1}^{m} \sum_{y_i} P_{syn}(y_i)^{\alpha} P'_{syn}(y_i)^{1-\alpha} \tag{247}$$

$$= \prod_{i=1}^{m} e^{\log \sum_{y_i} P_{syn}(y_i)^{\alpha} P'_{syn}(y_i)^{1-\alpha}} \tag{248}$$

$$= \prod_{i=1}^{m} e^{(\alpha-1)D_{\alpha}(P_{syn}||P'_{syn})} \tag{249}$$

$$= e^{(\alpha-1)mD_{\alpha}(P_{syn}||P'_{syn})} \tag{250}$$

From (244), we have the privacy constraint simplified to,

$$m \log\left(\sum_{y} P'_{syn}(y)\left(\frac{P_{syn}(y)}{P'_{syn}(y)}\right)^{\alpha}\right) - (\alpha-1)\varepsilon \leq \log \delta \tag{251}$$

Note that $\mathbb{E}_{P'_{syn}}[t^{\alpha}] = \sum_{y} P'_{syn}(y)\left(\frac{P_{syn}(y)}{P'_{syn}(y)}\right)^{\alpha}$ and $h(t) = t^{\alpha}$ is convex. Therefore,

$$t^{\alpha} = (\theta L + (1-\theta)U)^{\alpha} \leq \theta L^{\alpha} + (1-\theta)U^{\alpha} \tag{252}$$

$$\implies \mathbb{E}_{P'_{syn}}[t^{\alpha}] \leq \mathbb{E}_{P'_{syn}}[\theta]L^{\alpha} + \mathbb{E}_{P'_{syn}}[1-\theta]U^{\alpha} \tag{253}$$

$$= \frac{U-1}{U-L}L^{\alpha} + \frac{1-L}{U-L}U^{\alpha} \tag{254}$$

$$= \frac{\eta+1}{\eta+2}\frac{1}{(\eta+1)^{\alpha}} + \frac{1}{\eta+2}(1+\eta)^{\alpha} \tag{255}$$

where $\eta = \frac{(1-\beta)s}{\beta p}$. Then, from (251), the privacy constraint simplifies to:

$$m \log\left(\frac{\eta+1}{\eta+2}\frac{1}{(\eta+1)^{\alpha}} + \frac{1}{\eta+2}(1+\eta)^{\alpha}\right) - (\alpha-1)\varepsilon \leq \log \delta \tag{256}$$

Next, we find the optimum $\alpha^* > 1$ that minimizes the LHS of (256) with other parameters fixed. Let $f(\alpha) = m \log\left(\frac{\eta+1}{\eta+2}\frac{1}{(\eta+1)^\alpha} + \frac{1}{\eta+2}(1+\eta)^\alpha\right) - (\alpha-1)\varepsilon$ and $h(\alpha) = \frac{\eta+1}{\eta+2}\frac{1}{(\eta+1)^\alpha} + \frac{1}{\eta+2}(1+\eta)^\alpha$. Then,

$$f'(\alpha) = m\frac{h'(\alpha)}{h(\alpha)} - \varepsilon \tag{257}$$

$$h'(\alpha) = \frac{(\eta+1)^\alpha}{\eta+2}\log(\eta+1) - \frac{\eta+1}{\eta+2}(\eta+1)^{-\alpha}\log(\eta+1) = \frac{\log(\eta+1)}{\eta+2}((\eta+1)^\alpha - (\eta+1)^{1-\alpha}) \tag{258}$$

$$h''(\alpha) = \frac{\log^2(\eta+1)}{\eta+2}((\eta+1)^\alpha + (\eta+1)^{1-\alpha}) \tag{259}$$

$$f''(\alpha) = \frac{2m(h''(\alpha)h(\alpha) - h'(\alpha)^2)}{h^2(\alpha)} = \frac{m}{h^2(\alpha)}\frac{\log^2(\eta+1)}{(\eta+2)^2}(\eta+1) > 0 \tag{260}$$

Thus, $f(\alpha)$ has a unique minimum at $m\frac{h'(\alpha)}{h(\alpha)} - \varepsilon = 0$, which simplifies to:

$$\alpha^* = \frac{\log\left(\frac{(\eta+1)(m\log(\eta+1)+\varepsilon)}{m\log(\eta+1)-\varepsilon}\right)}{2\log(\eta+1)} \tag{261}$$

which requires $\varepsilon < m\log(\eta+1)$. For $\varepsilon \geq m\log(\eta+1)$, we can easily show that $f'(\alpha) < 0$ and the optimum $\alpha$ is given by $\alpha^* \to \infty$, at which $f(\alpha) = -\infty$, satisfying (251) for any $\beta$.

Note that (261) needs to satisfy $\alpha^* > 1$. For this, we need:

$$\frac{(1-\beta)s}{\beta p}\left(\log\left(1 + \frac{(1-\beta)s}{\beta p}\right) - \frac{\varepsilon}{m}\right) \leq \frac{2\varepsilon}{m} \tag{262}$$

Since the LHS of (262) is decreasing in $\beta$, any $\beta \geq \tilde{\beta}_1$ satisfies (262), where $\tilde{\beta}_1$ is the solution to $\beta$ in (262) with equality.

Now, plugging $\alpha^*$ back in the privacy constraint in (256) gives:

$$m\log\left(\frac{2m\log(\eta+1)}{\eta+2}\sqrt{\frac{\eta+1}{m^2\log^2(\eta+1)-\varepsilon^2}}\right) - \frac{\varepsilon}{2\log(\eta+1)}\left(\log\left(\frac{m\log(\eta+1)+\varepsilon}{m\log(\eta+1)-\varepsilon}\right) - \log(\eta+1)\right) \leq \log\delta \tag{263}$$

with $\eta = \frac{(1-\beta)s}{\beta p}$. Since the LHS of (263) is decreasing in $\beta$, any $\beta \geq \tilde{\beta}_2$ satisfies (263), where $\tilde{\beta}_2$ is the solution to $\beta$ in (263) with equality. Thus, we have,

$$\beta_{min} = \max\{\tilde{\beta}_1, \tilde{\beta}_2\} \tag{264}$$

where $\tilde{\beta}_1, \tilde{\beta}_2$ are functions of $\varepsilon, \delta, m, n, s$, and $p = \min_{y\in\{1,...,d\}} P_{nse}(y)$.

$\square$

# H. Implementation Details

## H.1. Implementation Details for the Experiment on Person Activity

We evaluate tabular DP synthetic data generation on the Person Activity dataset under a subject-based split to ensure there is no user overlap between private and public data. The dataset contains 164,860 total samples with 5 numerical features and 1 categorical feature, and the prediction target consists of 11 activity classes. We construct the private dataset from users with IDs $\{1, 2, 4\}$ (107,237 samples) and the public dataset from users with IDs $\{3, 5\}$ (57,623 samples).

Across methods, we consider privacy budgets $\varepsilon \in \{1.0, 2.0, 4.0\}$. We set $\delta = 1/(n\log n)$, where $n$ is the private dataset size. For the PE-based synthesis pipeline, we use 15 epochs with 3 sampling epochs, generate 5000 samples per sampling stage, and do 3 variations for candidate generation. For the PE+PubMIX variant, we keep the same base configuration and additionally set $\gamma$ with 0.9. For graphical-model baselines AIM, GSD, and GEM, we use the default configuration with degree=2. For PrivSyn, we use threshold with 20000 and 50 iterations as the default setting.

*Table 3.* Utility on Yelp Reviews under varying privacy budgets (public dataset = Google reviews from MA). We report mean $\pm$ std over multiple runs. Category is classification accuracy (%, higher is better) and Rating is RMSE (lower is better).

| Metric | Method | $\varepsilon = 1$ | $\varepsilon = 2$ | $\varepsilon = 4$ |
|---|---|---|---|---|
| **Category** ($\uparrow$) | PE (Xie et al., 2024) | **71.67**$\pm$**0.21** | 70.69$\pm$0.41 | 69.93$\pm$0.38 |
| | PUBMIX | 71.03$\pm$0.57 | **72.02**$\pm$**0.42** | **71.63**$\pm$**0.06** |
| **Rating** ($\downarrow$) | PE (Xie et al., 2024) | 0.898$\pm$0.025 | 0.905$\pm$0.009 | 0.889$\pm$0.021 |
| | PUBMIX | **0.874**$\pm$**0.011** | **0.887**$\pm$**0.027** | **0.869**$\pm$**0.010** |

## H.2. Implementation Details for the Experiment on Yelp

For text synthesis, we use the Yelp reviews corpus as the private training set (approximately 1.9M samples) with the standard dev/test splits. As the generator LLM, we use Llama-2-7B (Touvron et al., 2023) and enable stochastic decoding. Using PE, we run the synthesis procedure for 10 epochs. At each epoch, we generate 5,000 synthetic samples and set the variation degree to 0.5. To support selection via semantic similarity, we embed sentences using Sentence-BERT (Reimers & Gurevych, 2019) and perform nearest-neighbor matching with L2 distance. For privacy accounting, we consider $\varepsilon \in \{1, 2, 4\}$ and use the corresponding noise multipliers $\{15.34, 8.03, 4.24\}$, respectively. In addition to PE-style synthesis, we include PE+PubMix configured with $\delta = 10^{-5}$ and $\gamma = 0.1$.

For downstream utility evaluation, we fine-tune a RoBERTa-based classifier/regressor (Liu et al., 2019) on the generated synthetic reviews for two tasks: (i) business category prediction (10-class classification) and (ii) review star prediction (regression over 1–5). We train for 3 epochs with batch size 64, learning rate $3 \times 10^{-5}$, and maximum sequence length 64. We select the best checkpoint based on validation performance. We report accuracy for the category task and MAE for the star regression task.

The results for the same experiment as in Table 2, with a smaller public dataset (reviews from only MA as opposed to 10 states) is given in Table 3.

