# OpenReview forum: "Optimal Domain-Aware Privacy Mechanisms for Synthetic Data Generation"
_ICML.cc/2026/Conference — ICML 2026 regular_

### Official Review · Reviewer_sxdm · 2026-03-05

**Soundness:** 3
**Presentation:** 3
**Significance:** 3
**Originality:** 4
**Overall Recommendation:** 5
**Confidence:** 4

**Summary:**

The authors explore domain-aware noise distributions to achieve differential privacy (DP) when public data is available. The goal is to improve the privacy-utility trade-offs in DP synthetic data generation. The optimal noise distribution is theoretically characterized, leading to the PUBMIX framework, a domain-aware DP mechanism for histogram sampling that can be used in place of any DP histogram subroutine. Experiments on downstream tasks verify the improved privacy-utilty tradeoffs.

While prior synthetic data methods used public data via foundation model API access or public pre-training, this is the first work to directly incorporate public data into the DP noise mechanism design.

**Compliance With Llm Reviewing Policy:**

Affirmed.

**Final Justification:**

Cleverly leveraging public data is important to achieving good privacy-utility tradeoffs in modern machine learning applications.
The idea of using public data to calibrate the noise mechanism seems novel and broadly applicable.
I had slight reservations about the drop-in application to PE but the authors addressed this with additional experiments.
For these reasons, I recommend acceptance.

**Key Questions For Authors:**

1. Are there any practical rules-of-thumb for picking the TV radius $\gamma$? Or a way for estimate the optimal $\gamma$ using the privacy budget?

**Limitations:**

yes

**Strengths And Weaknesses:**

## Soundness

The theoretical claims are supported by rigorous proofs, which seem non-trivial and interesting, although I have not checked the appendix in detail. Experiments in a simplified setting confirm the improved utility compared to baseline techniques such as the Gaussian mechanism. The authors address the limitation that if the ball parameter is mispecified, the utility guarantee weakens.

## Presentation

I found the paper easy to follow since the high-level intuition is emphasized before the dense technical sections.

## Significance

Private synthetic data is an increasingly popular topic. The proposed PUBMIX mechanism offers good practical utility as it can be used as a drop-in replacement for DP histogram algorithms whenever public data is available. The experiments on PUMS seem quite convincing. I found the drop-in experiments on private evolution less convincing since some of the improvements seem smaller than the reported error margins.

## Originality

The perspective of designing the optimal noise mechanism in the presence of public data seems novel to me. The characterization in this setting as a “floor-raised” version of the public distribution is nice and clean.

---

> ### Author Rebuttal · Authors · 2026-03-31
>
> Thank you for the positive feedback and thoughtful comments!
>
> **W1:** *I found the drop-in experiments on private evolution less convincing since some of the improvements seem smaller than the reported error margins.*
>
> The private evolution experiment on tabular data (Table 1) shows that PUBMIX outperforms existing methods by 2.6%-5.6% in classification accuracy with error margins between 0.08%-0.49%.
>
> For the Yelp - rating prediction experiment, where the error margin issue exists, we provide additional results by evaluating PUBMIX under different public data settings. In particular, the original setup in the paper uses data from a single-state (MA) as public data. Here, we additionally construct a larger multi-state public dataset (from 10 states).
>
> The results are as follows:
>
> **Rating MSE (↓)**
>
> | Method                     | ε=1         | ε=2         | ε=4         |
> | -------------------------- | ----------- | ----------- | ----------- |
> | PE (Xie et al., 2024)      | 0.898±0.025 | 0.905±0.009 | 0.889±0.021 |
> | PUBMIX (MA public, paper)  | 0.874±0.011 | 0.887±0.027 | 0.869±0.010 |
> | PUBMIX (multi-state, new) | 0.810±0.016 | 0.814±0.022 | 0.800±0.039 |
>
> We observe that the performance gains of PUBMIX are not always within error margins. In particular, using a larger multi-state public dataset leads to improved results compared to both the original setup and the PE baseline (e.g., at $\epsilon=4$, MSE decreases from 0.889 to 0.800).
>
> We will include these additional results in the revised version to clarify the empirical impact of the proposed method.
>
>
>
> **Q1:** *Are there any practical rules-of-thumb for picking the TV radius gamma? Or a way for estimate the optimal gamma  using the privacy budget?*
>
> Thank you for raising this important point. $\gamma$ captures the natural variation of the domain data: if distributions estimated from different public datasets in the same domain vary substantially, $\gamma$ should be larger. If they are similar, a smaller $\gamma$ is appropriate. In our experiments, we estimate $\gamma$ heuristically. We randomly partition the public dataset into $k$ folds, compute pairwise TV distances between the fold distributions, and use the distribution of these distances to estimate $\gamma$ (see also footnote 3).
>
> We will include this discussion in the experimental section of the revised version.

---

> > ### Author Rebuttal · Reviewer_sxdm · 2026-03-31
> >
> > Thanks for the additional PE experiments. I am happy to maintain my positive score.

---

### Official Review · Reviewer_hYTf · 2026-03-06

**Soundness:** 2
**Presentation:** 2
**Significance:** 2
**Originality:** 2
**Overall Recommendation:** 4
**Confidence:** 3

**Summary:**

The paper studies how public data can be incorporated directly into DP mechanisms for synthetic data generation, rather than only through preprocessing or post-processing as in prior work. Focusing on the canonical setting of histogram-based distribution estimation, the authors develop a theoretical framework for domain-aware DP noise mechanisms and characterize the optimal noise distribution within a class of mixing-based mechanisms. Their analysis shows that the optimal mechanism depends on the public data distribution through a floor-raised transformation. Based on these insights, they propose PUBMIX, a practical domain-aware DP mechanism that can replace standard Gaussian or Laplace noise in histogram-based synthetic data pipelines.

**Compliance With Llm Reviewing Policy:**

Affirmed.

**Final Justification:**

I have revised my score positively following the satisfactory answers provided by the authors.

**Key Questions For Authors:**

Please refer to the previous section on Weaknesses.

**Limitations:**

Unfortunately, the limitations of the approach were not discussed in the paper.

**Strengths And Weaknesses:**

**Strengths**

- The paper proposes a novel paradigm for incorporating public data directly into DP mechanisms.
- The paper addresses a relevant and well-established problem, which is improving the utility of differentially private synthetic data generation while maintaining privacy guarantees.
- The paper is generally well-written and structured.

**Weaknesses**

- A major flaw is that the optimization of the two key parameters of PUBMIX, $\beta$ and $P_{nse}$, is performed using the private dataset. The paper does not describe any method to compute these parameters in a DP-compliant manner. This means the claimed differential privacy guarantees are not valid. The mechanism, as described, directly leaks private data during parameter optimization, which contradicts the core DP claim. This undermines both the theoretical and empirical contributions because the reported improvements in utility cannot be safely attributed to a privacy-preserving method.
- Like most approaches that rely on public datasets, PUBMIX assumes that such a dataset is very similar to the private data in its statistical properties. In practice, finding a suitable public dataset that closely matches the private data is often difficult, which limits the practical applicability of the method.

---

> ### Author Rebuttal · Authors · 2026-03-31
>
> Thank you for the feedback!
>
> **W1:** *A major flaw is that the optimization of the two key parameters of PUBMIX, beta  and P_nse, is performed using the private dataset. The paper does not describe any method to compute these parameters in a DP-compliant manner. This means the claimed differential privacy guarantees are not valid. The mechanism, as described, directly leaks private data during parameter optimization, which contradicts the core DP claim. This undermines both the theoretical and empirical contributions because the reported improvements in utility cannot be safely attributed to a privacy-preserving method.*
>
> We would like to clarify that the two parameters  $\beta$ and $P_{nse}$ in PUBMIX are **not** derived from the given private dataset. They are pre-computed parameters that are independent of the given private dataset. As formulated in Eq. (3), the optimization is performed over the worst-case private distribution  within a $\gamma$-TV ball around the public distribution $P_{pub}$. The inner max of the optimization in (3) explains this (see also lines 143-145). This worst-case distribution is entirely determined by $P_{pub}$ and $\gamma$, neither of which is private. In other words, this worst-case distribution is independent of the given private distribution $P_{prv}$. Once these minmax $\beta$ and $P_{nse}$ are obtained, they are used to mix with the given private distribution $P_{prv}$ to draw DP samples, at which point the standard DP guarantees apply.
>
> We will clarify this in the problem formulation section of the revised version to avoid confusion.
>
>
> **W2:** *Like most approaches that rely on public datasets, PUBMIX assumes that such a dataset is very similar to the private data in its statistical properties. In practice, finding a suitable public dataset that closely matches the private data is often difficult, which limits the practical applicability of the method.*
>
> We agree that this is a challenge all methods that use public data face. That said, PUBMIX does not assume that the public data is nearly identical to the private data. It only requires that the public data provide useful structure that can be used to inform the design of a domain-aware noise distribution. Moreover, if the public data is less reliable or exhibits greater variation, one can choose a larger $\gamma$ to reduce its dependency. In practice, we set $\gamma$ based on the observed variation within the public dataset.
>
> **L1:** *Unfortunately, the limitations of the approach were not discussed in the paper*
>
> Thank you for raising this important point. The main limitations of our approach are: (1) restricting to linear mixing mechanisms, (2) requiring public data from the same domain, and (3) focusing the complete theoretical treatment only on histogram-based methods.
>
> These are briefly mentioned in sections 2 and 6, but we will add a separate limitations section to the revision to make it clear.

---

> > ### Author Rebuttal · Reviewer_hYTf · 2026-04-03
> >
> > I would like to thank the authors for their responses, which addressed my concerns. Accordingly, I will raise my score.

---

### Official Review · Reviewer_WfCZ · 2026-03-13

**Soundness:** 4
**Presentation:** 4
**Significance:** 3
**Originality:** 3
**Overall Recommendation:** 4
**Confidence:** 4

**Summary:**

This work introduces a novel paradigm for differentially private (DP) synthetic data generation by directly embedding domain-specific public data into the noise mechanism itself, rather than relying on it merely for pre-processing or post-processing. The authors theoretically characterize the optimal domain-aware noise distribution for normalized histograms and propose PUBMIX, a plug-and-play DP sampling mechanism. Empirical evaluations demonstrate that integrating PUBMIX into existing synthesis pipelines significantly improves the privacy-utility trade-offs across both tabular and text modalities.

**Compliance With Llm Reviewing Policy:**

Affirmed.

**Final Justification:**

I think all my concerns have been addressed now. I will keep my positive score and raise my confidence accordingly.

**Key Questions For Authors:**

See weaknesses.

**Limitations:**

yes

**Strengths And Weaknesses:**

## Strengths

- The paper addresses a well-motivated problem. The idea of encoding public information inside the mechanism itself is novel and relevant for practical histogram-based DP data synthesis.

- The theory is one of the stronger parts of the paper. The optimization problem is clearly formulated, and the resulting structure of the optimal noise distribution is nontrivial and technically interesting.

- The method is easy to plug into existing pipelines. Replacing the histogram sampling step in PE and TABPE makes the practical value of the proposal easy to understand.

## Weaknesses

- The empirical section may not fully support the breadth of the claims in the introduction. The evidence is still concentrated on a fairly narrow set of histogram-based settings.

- Since the main claim is about the value of public data, the paper should compare more carefully against simpler and stronger public-data-aware baselines, not only against methods that do not use public information in the same way.

- The choice of the domain-mismatch parameter $\gamma$ is not well justified. As the guarantees and expected gains depend heavily on this quantity, the current choices feel somewhat ad hoc without a stronger selection rule or sensitivity analysis.

---

> ### Author Rebuttal · Authors · 2026-03-31
>
> Thank you for the positive feedback and thoughtful comments!
>
> **W1:** *The empirical section may not fully support the breadth of the claims in the introduction. The evidence is still concentrated on a fairly narrow set of histogram-based settings.*
>
> We agree that the empirical results are concentrated on histogram-based settings.  While our problem formulation can be extended more broadly, our theoretical analysis focuses on DP histogram sampling in the context of DP synthetic data generation. Our experiments are designed to validate these theoretical claims. We will make this clear in the revision by stating explicitly that our empirical claims concern the effectiveness of PUBMIX in histogram-based pipelines.
>
>
>
> **W2:** *Since the main claim is about the value of public data, the paper should compare more carefully against simpler and stronger public-data-aware baselines, not only against methods that do not use public information in the same way.*
>
>
> Thank you for this suggestion. We implemented JAM-PGM [6], which is a tabular DP synthetic data generation method that leverages public data and reports stronger performance among public-data-aware methods. We evaluate this method on the tabular benchmark, which reports downstream classification accuracy (corresponding to Table 1).
>
> | Method                    | ε=1.0             | ε=2.0             | ε=4.0             |
> | ------------------------- | ----------------- | ----------------- | ----------------- |
> | JAM-PGM (public)          |   55.96    |   56.51    |   57.37    |
> | PUBMIX (ours)        |   66.40         |   66.47          |   66.85         |
>
> We observe that JAM-PGM does not outperform PUBMIX under the same evaluation setting. We will include this additional comparison and more baselines in the revised version.
>
>
>
> [6] https://proceedings.mlr.press/v238/fuentes24a/fuentes24a.pdf
>
>
>
> **W3:** *The choice of the domain-mismatch parameter gamma is not well justified. As the guarantees and expected gains depend heavily on this quantity, the current choices feel somewhat ad hoc without a stronger selection rule or sensitivity analysis.*
>
> $\gamma$ is a domain-dependent hyperparameter that does not affect privacy, but does influence utility (see footnote 1). It captures the natural variation of the domain data: if distributions estimated from different public datasets in the same domain vary substantially, $\gamma$ should be larger; if they are similar, a smaller $\gamma$ is appropriate. In our experiments, we estimate $\gamma$ heuristically. We randomly partition the public dataset into $k$ folds, compute pairwise TV distances between the fold distributions, and use the distribution of these distances to estimate $\gamma$ (see also footnote 3).
>
> We will include this discussion in the experimental section of the revised version.

---

> > ### Author Rebuttal · Reviewer_WfCZ · 2026-04-04
> >
> > Thank you for your response. I think all my concerns have been addressed now. Good paper. I will keep my positive score and raise my confidence accordingly.

---

### Official Review · Reviewer_sA2d · 2026-03-19

**Soundness:** 3
**Presentation:** 4
**Significance:** 4
**Originality:** 3
**Overall Recommendation:** 4
**Confidence:** 3

**Summary:**

The present paper examines the problem of generating samples from a discrete distribution with differential privacy. The go-to approach a practitioner might reach for is (1) computing the empirical histogram of private data; (2) adding gaussian noise; (3) post processing the noisy histogram into a probability distribution and then sampling from it.

The present paper points out that such a process is generally stronger than what is needed; and as such, makes some reasonable relaxations in pursuit of better utility; (1) the practioner only needs m samples, not the entire distribution; (2) the practictioner may have access to some public distribution to anchor results to.

For the class of mechanisms that are defined by sampling from $(1-\beta)P_{priv} + \beta P_{noise}$, they derive an procedure that yields an optimal solution.

**Compliance With Llm Reviewing Policy:**

Affirmed.

**Final Justification:**

The rebuttal partly resolved my concerns regarding discussing prior works that looked at the same problem.

The new empirical result they presented helped, but I was puzzled as to why it does worse than the baseline with not public information, and believe there needs to more investigation there, which is beyond the scope of the present paper.

As such I maintain my positive assessment of the paper. The paper is nice, but I would have liked to see more from the empirical side (comparing to other reasonable baselines that use public data / other algorithms proposed for the same task with a different theoretical framing),

**Key Questions For Authors:**

Have authors considered possible applications to private prediction? Authors attribute improvements to the fact that the output is simply a sample instead of a full distribution. Similar empricial improvements have been seen (Privatizing the histogram: https://arxiv.org/abs/2309.11765, privatizating the sample: https://arxiv.org/abs/2407.12108).

Why does LinMix error blow up beyond Gaussian noise histogram as the number of samples increase? In theory, we could re-estimate the distribution with the samples from LinMix, and then sample from that. Does such a problem point to a limitation in the problem formulation (3)?

Where does the algorithm implied by Gaussian Mechanism on the histogram fit into the problem formulation and optimality claims? It does better than LinMix for sufficiently large synthetic data samples.

A large body of work has discussed the issues of typical gaussian histograms in the sparse setting. How does sparsity affect the performance of the present algorithm? Are PE problems typically sparse?

**Limitations:**

yes

**Strengths And Weaknesses:**

Strengths
- The paper is very well motivated and written extremely clearly. The well-laid out motivation and formulation of the problem is already a useful contribution. Plenty of anticipated reader questions are answered in the writing.
- The final algorithm is simple and easy to implement.
- Empirical improvements over TabPE and AIM, which are both very strong baselines.

Weaknesses
- The related work focuses on high-level connections to DP synthetic data and (as pointed out by the authors) a similar but subtly different problem in DP histogram estimation; but it seems to miss some important comparisons to the most technically similar work on private sampling or private prediction which this makes it difficult to evaluate the contribution. In fact, I believe the problem at hand has been studied in the literature under the name "differentially private multi-sampling" theoretically (https://arxiv.org/abs/2412.10512); and mixing the sampling distribution with a public one has also been studied in the context of LLM token sampling (https://arxiv.org/pdf/2201.00971, https://arxiv.org/abs/2403.15638).
- The title says optimal, but as pointed out, they only produce optimality for a class of methods -- not necessarily general mechanisms. Do the authors make any claims to towards optimality in general? Or do they believe this class of methods is near optimal (thus motivating studying this class) possibly due to evidence from other problems (e.g. mean estimation). I believe this discussion is missing and would be helpful to understand the results.
- Empirical improvements are strong but there are some concerns regarding soundness. There seems to be a mismatch of the setting, as the other methods do not make use of public data at all. To be fair, we should compare to the baseline of PE with a straightforward mixing of the public histogram with the private histogram, using the same choice of $\gamma$ as PubMix: $\gamma P_{DPHistogram} + (1-\gamma)P_{public}$ (although really, this $\gamma$ is a free parameter that should searched over for the best result).
- There is little explanation for the choice of $\gamma$ in the paper. $\gamma$ is taken to be 0.1 in the tabular experiments (suggesting we are almost entirely using the public distribution) and 0.9 for Yelp (recovering almost LinMix). For tabular experiments, if we are heavily leaning into the public distribution, then the other baselines should be able to share the advantage. For Yelp, Some explanation on why a low choice of $\gamma$ compared to the other experiment was used would be helpful.


Overall I am excited about the paper. Some further clarification on experimental details, and elaborated connections to prior work are sufficient for me to raise my score.

---

> ### Author Rebuttal · Authors · 2026-03-31
>
> Thank you for the positive feedback and insightful comments!
>
> **W1:** PUBMIX extends the DP multi-sampling setting by incorporating public data into the problem formulation. In contrast to DP multi-sampling [1], which gives mechanism-agnostic sample-complexity bounds over broader distribution families, we study a more structured setting to make public-data integration tractable. Specifically, we focus on normalized histograms as distribution estimators and consider the class of linear mixing mechanisms, to analyze the optimal privacy mechanism within this class.
>
> Private prediction [2,3] and PUBMIX have different goals and formulations. Their only similarity is using a linear mixing step to achieve DP. In [2,3], a private token distribution is mixed with a fixed public distribution, and only the mixing weight is optimized. In contrast, PUBMIX mixes a private normalized histogram with a noise distribution and jointly optimizes both the mixing weight and the noise distribution, allowing improved flexibility to the privacy mechanism.
>
> We will include this discussion under related work in the revised version.
>
> [1] https://arxiv.org/abs/2412.10512
> [2] https://arxiv.org/pdf/2201.00971
> [3] https://arxiv.org/abs/2403.15638
>
> **W2:** In this work, we only consider the class of linear mixing mechanisms. Finding the optimal general mechanism remains an open problem, which our framework helps motivate. We will make this precise in our revision.
>
> **W3:** We implemented the proposed baseline (call it DirectMix) in the tabular PE experiment. The downstream classification accuracy is provided below. DirectMix does not improve over PUBMIX.
>
> |Method|eps=1|eps=2|eps=4|
> |-|-|-|-|
> |TabPE|60.75|63.06|64.00|
> |DirectMix|60.60|62.06|62.46|
> |PUBMIX|66.40|66.46|66.85|
>
> **W4:** $\gamma$ captures the variation of the domain data: if distributions estimated from different public datasets in the same domain vary substantially, $\gamma$ should be larger. If they are similar, a smaller $\gamma$ is appropriate. In our experiments, we randomly partition the public dataset into k folds, compute pairwise TV distances between the fold distributions, and use the distribution of these distances to estimate $\gamma$. We will state this in the revision.
>
> **Q1:** Yes, this is on our radar. Private prediction is a natural application of PUBMIX, via its extension to general discrete distributions in Appendix C. This setting is promising because private prediction operates in the low-sample regime (sample only one token), where PUBMIX is most effective. We leave this as an interesting future direction.
>
> On related work: [4] privatizes the entire averaged private token distribution, whereas PUBMIX privatizes only the released sample. Both [5] and PUBMIX release private samples, but the mechanisms are different: [5] uses the exponential mechanism, while PUBMIX constructs a domain-aware noise distribution and releases samples through optimized linear mixing.
>
> [4] https://arxiv.org/abs/2309.11765
> [5] https://arxiv.org/abs/2407.12108
>
> **Q2: Gaussian-LINMIX comparison:** As shown in Fig. 1(b), the LINMIX synthetic distribution $P_{syn}$ moves along the line connecting $P_{prv}$ and the uniform distribution $P_{unif}$ as $m$ increases: close to $P_{prv}$ for small m, and close to $P_{unif}$ as $m \to \infty$. The Gaussian mechanism on the other hand, as shown in fig. 1c, lies at a fixed distance from $P_{prv}$ irrespective of $m$. Thus, for smaller to moderate $m$, LINMIX outperforms Gaussian. As $m$ grows large, in cases where $P_{prv}$ is far from $P_{unif}$, Gaussian performs better.
>
> **Re-estimating $\hat{P}_{syn}$  from LINMIX samples:** This is a very interesting question! There is a fundamental trade-off here. Accurately re-estimating the synthetic distribution to draw more (free) samples requires a larger number of LINMIX samples. At the same time, releasing a larger number of LINMIX samples moves $P_{syn}$ away from the private distribution $P_{prv}$. Thus, as the estimate $\hat{P}_{syn}$ improves, LINMIX pays an increased utility cost. Analyzing this trade-off is an interesting future direction.
>
> **Q3:** The Gaussian mechanism does not directly fit into the formulation in (3). The purpose of the formulation in (3) is to find a synthetic distribution $P_{syn}$ as close as possible to the private distribution $P_{prv}$, considering the **finite sample requirement m**. The Gaussian mechanism is sample-agnostic: it always samples from the same noise-added distribution regardless of m. As shown in Fig. 1c, this places Gaussian $P_{syn}$ at a TV distance from $P_{prv}$ that scales with dimension d, in an arbitrary direction within the simplex. This is outside the sample-aware linear mixing class we study.
>
> **Q4:** Most of the challenges with sparse DP histograms arise when the goal is to release the entire privatized histogram. PUBMIX avoids this by releasing only samples. In our PE experiments, the private histograms had sparsity levels 20%-40%.

---

> > ### Author Rebuttal · Reviewer_sA2d · 2026-04-03
> >
> > My concerns regarding related work are resolved. I trust the authors will perform a close examination of aforementioned cited works, which will help readers understand and engage with their contributions. I understand it is difficult to compare the works due to the different choices in the theoretical design and notions of optimality, but it should be recognized that all these works are attempting to solve the same problem (getting samples from a histogram privately), and could potentially be compared empirically, although that is likely out of scope for this work.
> >
> > Thank you for implementing the baseline. Since the choice of $\gamma=0.1$ is used in tabular experiments, directmix should be only $<=0.2$ TV away from Pubmix, so I am surprised it performs so much worse, and even worse than Gaussian noise DP histogram with no public data. However I recognize a full sweep of the hyperparameters and detailed analysis is out of scope for the rebuttal.

---

### Decision · Program_Chairs · 2026-04-30

**Decision:**

Accept (regular)

**Comment:**

This paper makes a clear and useful contribution to differentially private synthetic data generation. Reviewers found the problem well motivated and the main idea novel: rather than using public data only in preprocessing or postprocessing, the paper incorporates public information directly into the DP mechanism itself. The theoretical characterization of the optimal domain-aware mechanism within the studied mixing class is interesting, the resulting PubMix method is simple easy to use as a olug-and-play in common pipelines, and the empirical results are strong.

The main concerns were about positioning and evaluation. In particular, the paper needed to be more explicit that its optimality claim is within a specific mechanism class (histogram based), to engage more carefully with closely related work on private sampling and private prediction, to justify the choice of the domain-mismatch parameter, and to compare against stronger public-data-aware baselines. The rebuttal addressed these concerns well by clarifying the theoretical scope, explaining how the parameter gamma is estimated, and adding additional baseline comparisons and experiments. Reviewers remained positive after the rebuttal. What remains still somewhat unsatisfying is that gamma, an important parameter for utility, is assumed to be known or publically estimated.

I recommend acceptance, if space permits. The final version should make the scope of the optimality claim especially explicit in the main text and keep the additional related-work and baseline discussion promised in the rebuttal.